# Time-dependent memory transformation in hippocampus and neocortex is semantic in nature

Valentina Krenz [1], Arjen Alink[2,3], Tobias Sommer [3], Benno Roozendaal [4,5] & Lars Schwabe [1] ✉

Memories undergo a time-dependent neural reorganization, which is assumed to be accompanied by a transformation from detailed to more gist-like memory. However, the nature of this transformation and its underlying neural mechanisms are largely unknown. Here, we report that the time-dependent transformation of memory is semantic in nature, while we find no credible evidence for a perceptual transformation. Model-based MRI analyses reveal time-dependent increases in semantically transformed representations of events in prefrontal and parietal cortices, while specific pattern representations in the anterior hippocampus decline over time. Posterior hippocampal memory reinstatement, in turn, increases over time and is linked to the semantic gist of the original memory, without a statistically significant link to perceptual details. These findings indicate that qualitative changes in memory over time, associated with distinct representational changes in the neocortex and within the hippocampus, reflect a semantic transformation, which may promote the integration of memories into abstract knowledge structures.

Episodic memory changes over time. Converging lines of evidence from lesion studies in rodents[1,2], human neuroimaging studies[3–5] or studies in amnesic patients[6,7] indicate that episodic memories undergo a time-dependent neural reorganization. While memories are initially dependent on the hippocampus, they become more dependent on neocortical structures, such as the ventromedial prefrontal cortex (vmPFC)[8–10], inferior frontal gyrus (IFG)[4,5], anterior cingulate cortex (aCC)[2,11–13], angular gyrus and precuneus[14,15], as time after encoding proceeds. Whether remote memories become entirely independent of the hippocampus is still debated[16–18] and, intriguingly, initial evidence points to the possibility of a time-dependent reorganization of memories within the hippocampus, from anterior to parietal parts[19,20]. Critically, the neural reorganization of memory is thought to be accompanied by a transformation from a detailed episodic memory

trace to a more gist-like representation[16,17]. Such qualitative changes over time are a fundamental aspect of memory and may promote the building of abstract knowledge networks[4]. Moreover, they have highly relevant implications, for instance, for eyewitness testimony or the generalized memory for aversive events in mental disorders.

The nature of these qualitative changes of memories over time remains, however, elusive. One possible mechanism is a perceptual transformation, in which a detailed, perceptually rich episodic trace evolves over time into a less specific trace that contains knowledge of general perceptual features of the original event (e.g. 'I remember the painting contained a lot of red and brown'). Indeed, the hippocampus is critically implicated in remembering perceptual details[21] and the perceptual transformation perspective may be close to the common view that memories fade away and simply lose (perceptual) detail over

[1]Department of Cognitive Psychology, Institute of Psychology, Universität Hamburg, Von-Melle-Park 5, 20146 Hamburg, Germany. [2]Department of General Psychology, Institute of Psychology, Universität Hamburg, Von-Melle-Park 11, 20146 Hamburg, Germany. [3]Department of Systems Neuroscience, University Medical Centre Hamburg-Eppendorf, Martinistraße 52, 20246 Hamburg, Germany. [4]Department of Cognitive Neuroscience, Radboud University Medical Center, 6500 HB Nijmegen, The Netherlands. [5]Donders Institute for Brain, Cognition and Behaviour, Radboud University, Kapittelweg 29, 6525 EN Nijmegen, The Netherlands. ✉e-mail: lars.schwabe@uni-hamburg.de

time[22]. Alternatively, with time, memories may not be just a perceptually degraded version of the original trace but become semantically transformed into representations that carry the semantic gist, with only minimal (detailed or generalized) perceptual information (e.g. 'I remember the painting showed an apple on a table'). This semantization of memories over time may provide a better explanation of how episodic experiences are integrated into abstract knowledge structures than a mere decay of (perceptual) features of a memory trace. While prominent theoretical accounts appear to favor the semantic transformation view[16,17], there is a lack of clear empirical evidence for a semantic transformation of memory over time. Paradigms used in previous studies on time-dependent memory transformation in humans or rodents involved tests of transformation that were both semantically and perceptually similar to the original event and could thus not distinguish between different mechanisms of transformation. Thus, whether the transformation of memory over time is perceptual or semantic in nature (or both) remains unclear.

In the present experiment, we aimed at elucidating the nature and neural signature of time-dependent memory transformation. Specifically, we sought to determine whether there is a semantic or a perceptual transformation of the original memory over time. Moreover, because emotional arousal has been shown, on the one hand, to enhance memory for the gist of the event at the cost of reduced memory for peripheral features[23–25] but, on the other hand, to increase memory specificity in the long-run[2,26], we further tested whether the nature of memory transformation over time, as well as its neural underpinnings, would differ depending on the level of emotionality of the encoded material. To this end, we tested participants' recognition memory for emotionally neutral and negative pictures either 1d or 28d after encoding. As the neural reorganization of memories can be expected to be much further progressed 28d compared to 1d after encoding[2,19], varying the delay between encoding and recognition testing allowed probing time-dependent memory transformation. Critically, this recognition test included, in addition to initially encoded and entirely new pictures, also lures that were either perceptually or semantically related to the original stimuli. Encoding as well as memory testing took place in an MRI scanner, enabling us to analyze time-dependent changes in the reinstatement of encoding patterns and the specificity of memory representations during memory testing by leveraging multivariate fMRI-analysis approaches. A perceptual transformation would be indicated if, with increasing delay after encoding, perceptually related, but not semantically related, items are endorsed as 'old'. Conversely, a semantic transformation would be indicated if participants endorse semantically related, but not perceptually related, items as 'old'.

Here, we show that episodic memories are semantically transformed over time, while we obtain no credible evidence for a perceptual transformation. This time-dependent semantization of memories was further enhanced for emotionally negative compared to neutral stimuli. At the neural level, the time-dependent transformation of memories was reflected in semantic, gist-like representations of remote memories in prefrontal as well as parietal neocortical storage sites. The anterior hippocampus was associated with distinct representations of encoded events that declined with increasing delay after encoding. Posterior hippocampal memory reinstatement increased over time and was associated with less specific memory representations that were linked to the semantic gist of the original memory, again without evidence for a reliable effect of the perceptual gist.

## Results

To elucidate whether episodic memories are semantically or perceptually transformed over time and whether this process is equally evident for emotionally neutral compared to negative pictures, we performed a 3-day study: Day 1—encoding of emotionally neutral or negative pictures in the MRI scanner; Day 2 (either 1d or 28d after

Day 1)—recognition testing in the MRI scanner; Day 3—individual assessment of the semantic and perceptual relatedness of the stimulus material. In order to dissociate semantic and perceptual mechanisms of time-dependent memory transformation, the recognition test included, in addition to original and entirely novel items, items that were either perceptually or semantically related to the original pictures. Each originally encoded picture corresponded precisely to one semantically related, one perceptually related and one unrelated picture, matching the original picture in terms of the level of emotionality and other relevant features (see methods section). The semantic and perceptual relatedness of each originally encoded item to their corresponding semantically related, perceptually related, or unrelated lure was tested in an independent behavioral pilot study ($n = 32$ participants), which confirmed that semantically related items were rated as significantly more semantically related but significantly less perceptually related to the original items than perceptually related items (see Supplementary Fig. 1).

On the first experimental day, 52 healthy, right-handed young adults (26 females, 26 males, age: $M = 24.29$ years, SEM = 0.55 years) encoded 60 pictures (30 emotionally neutral, 30 emotionally negative) in an MRI scanner, each presented for 3 s in each of three consecutive runs (see Fig. 1). To control for alertness during encoding, participants were instructed to respond with a button press as soon as a fixation cross appeared between trials. On average, participants missed only 1.48 (SEM = 0.43) responses across all trials and runs, indicating that participants were attentive during encoding, without statistically significant differences between 1d- and 28d-groups (main effect delay: $F(1, 50) = 1.46$, $p = 0.233$, $\eta_p^2 = 0.03$, 95% Confidence Interval: [9e−05, 0.18]; delay × run: $F(1.87, 93.71) = 0.84$, $p = 0.429$, $\eta_p^2 = 0.02$, 95% Confidence Interval: [0.001, 0.12]; mixed ANOVA). To ensure that the 1d- and 28d-groups did not differ in initial encoding, we asked participants to recall as many of the pictures as possible immediately after the encoding session. In this immediate free recall test, participants recalled on average 50.99% (SEM = 2.21%) of the 60 previously encoded items. A mixed ANOVA with the between-subjects factor delay (1d vs. 28d) and the within-subject factor emotion (neutral vs. negative) did not indicate a statistically significant difference between delay groups in immediate memory performance (main effect delay: $F(1, 50) = 0.17$, $p = 0.678$, $\eta_p^2 = 0.003$, 95% Confidence Interval: [2e−05, 0.11]; delay × emotion: $F(1, 50) = 1.13$, $p = 0.293$, $\eta_p^2 = 0.02$, 95% Confidence Interval: [6e−05, 0.16]). As expected, participants recalled significantly more negative ($M = 58.78\%$, SEM = 2.30%) than neutral pictures ($M = 43.21\%$, SEM = 2.49%; main effect emotion: $F(1, 50) = 69.33$, $p = 5e−11$, $\eta_p^2 = 0.58$, 95% Confidence Interval: [0.42, 0.72]; Supplementary Fig. 2), indicating an enhancement of immediate memory performance due to the emotionality of the encoded material, in line with previous reports[27,28].

### Memories are semantically transformed over time

On experimental Day 2 (either 1d or 28d after initial encoding), participants underwent a recognition test in which they were instructed to indicate for each of the presented pictures, whether the picture had been presented on Day 1 ('old') or not ('new'). Critically, this recognition test included, in addition to original and entirely novel, unrelated items, lures that were either semantically or perceptually related to the old items, thus enabling us to examine the nature of time-dependent memory transformation. As expected, the hit rate was significantly higher in the 1d-group ($M = 91.86\%$, SEM = 1.12%) than in the 28d-group ($M = 75.58\%$, SEM = 2.45%; main effect delay: $F(1, 50) = 20.72$, $p = 3e−05$, $\eta_p^2 = 0.29$, 95% Confidence Interval: [0.11, 0.49]; Fig. 2a and Supplementary Table 1). Notably, this delay-dependent decrease in memory performance was dependent on the emotionality of the stimuli (emotion × delay: $F(1, 50) = 9.23$, $p = 0.004$, $\eta_p^2 = 0.16$, 95% Confidence Interval: [0.02, 0.36]; main effect emotion: $F(1, 50) = 4.52$, $p = 0.038$, $\eta_p^2 = 0.08$, 95% Confidence

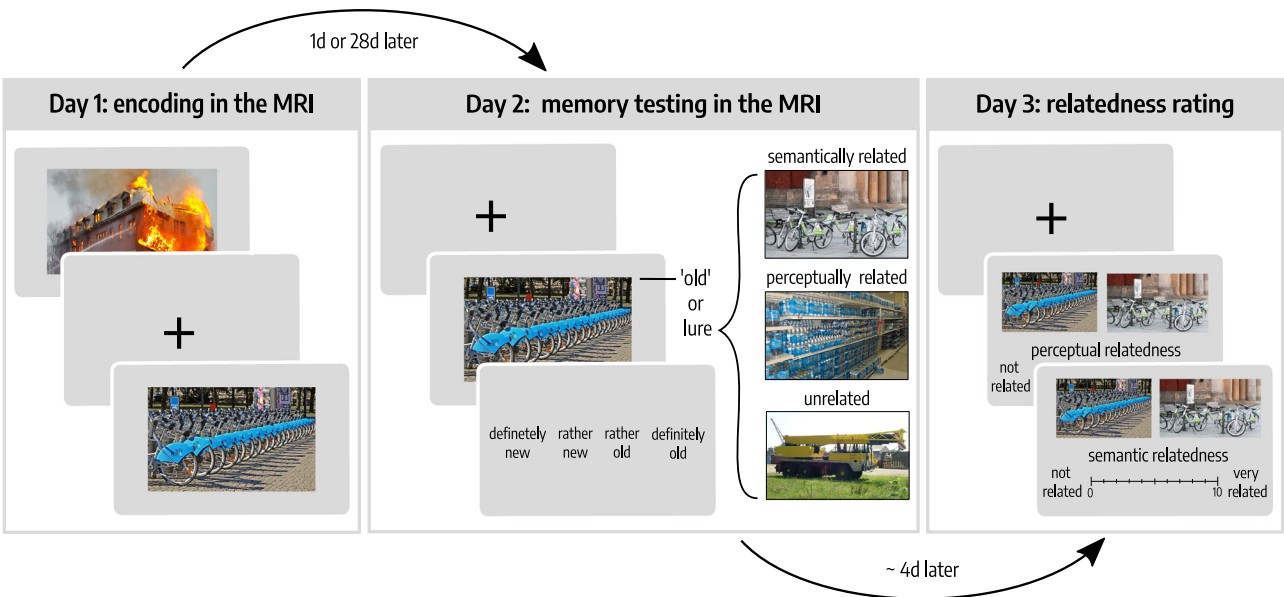

**Fig. 1 | Experimental paradigm.** On the first experimental day (Day 1), participants encoded 30 emotionally neutral and 30 negative pictures, each presented once in each of three consecutive runs. After a delay of 1d or 28d (Day 2), participants were presented with the encoded pictures, lures that were perceptually or semantically related to the old pictures or entirely novel, unrelated material in a recognition test. Both encoding and memory testing were conducted in an MRI scanner. On the third experimental day (Day 3), participants rated the individually perceived semantic and perceptual relatedness between each old image and their corresponding semantically related, perceptually related or unrelated lure. All depicted images are licensed under Creative Commons BY-SA License: image representing emotionally negative item at encoding (fire) is courtesy of Sylvain Pedneault ([https://commons.](https://commons.) wikimedia.org/wiki/File:Fire_inside_an_abandoned_convent_in_Massueville,_ Quebec,_Canada.jpg; edited), image representing 'old' item is courtesy of W. Bulach ([https://commons.wikimedia.org/wiki/File:00_2141_Bicycle-sharing_systems_-_](https://commons.wikimedia.org/wiki/File:00_2141_Bicycle-sharing_systems_-_) Sweden.jpg; edited), image representing 'semantically related' item is courtesy of Matti Blume ([https://commons.wikimedia.org/wiki/File:Bike_share_2019_Berlin_](https://commons.wikimedia.org/wiki/File:Bike_share_2019_Berlin_) (P1080139).jpg; edited), image representing 'perceptually related' item is courtesy of Ivy Main ([https://fi.m.wikipedia.org/wiki/Tiedosto:Bottled_water_in_](https://fi.m.wikipedia.org/wiki/Tiedosto:Bottled_water_in_) supermarket.JPG; edited), image representing 'unrelated' item is courtesy of Hannes Drexl ([https://commons.wikimedia.org/wiki/File:Autokran_Seite.jpg?](https://commons.wikimedia.org/wiki/File:Autokran_Seite.jpg?) uselang=de; unchanged).

Interval: [0.002, 0.27]): the decrease in hits for the 28d- compared to the 1d-group was significantly lower for emotionally negative compared to emotionally neutral pictures (interaction contrast: $t(50) = 3.04$, $p = 0.004$, $d = 0.40$, 95% Confidence Interval = [0.14, 0.66]). Accordingly, the hit rate for negative pictures after 28d was significantly higher than for neutral pictures (paired $t$-test: $t(50) = -3.65$, $p = 6e-04$, $d = -0.66$, 95% Confidence Interval = [−1.01, −0.31]), while there was no statistically significant difference in the hit rate for emotionally negative and neutral pictures when tested 1d after encoding (paired $t$-test: $t(50) = 0.64$, $p = 0.522$, $d = 0.14$, 95% Confidence Interval = [−0.28, 0.56]). The latter finding may be owing to the overall very high memory performance on the recognition test 1d after encoding.

To assess the nature of memory transformation over time, the key question of this study, we analyzed participants' false alarms (FAs) to unrelated (i.e., entirely novel), semantically related and perceptually related lures by means of a mixed ANOVA with the between-subjects factor delay (1d vs. 28d) and the within-subject factors emotion (neutral vs. negative) and lure type (unrelated vs. semantically related vs. perceptually related). This analysis showed a time-dependent increase in FA rates depending on the lure type (delay × lure type: $F(1.55, 77.43) = 9.33$, $p = 7e-04$, $\eta_p^2 = 0.16$, 95% Confidence Interval: [0.05, 0.32]; main effect lure type: $F(1.55, 77.43) = 42.90$, $p = 2e-11$, $\eta_p^2 = 0.46$, 95% Confidence Interval: [0.32, 0.60]; main effect delay: $F(1, 50) = 6.79$, $p = 0.012$, $\eta_p^2 = 0.12$, 95% Confidence Interval: [6e-03, 0.29]). As shown in Fig. 2a, a striking increase in the FA rate for the 28d- compared to the 1d-group was observed selectively for semantically related lures (two-sample $t$-test: $t(50) = -3.32$, $p = 0.002$, $d = -1.09$, 95% Confidence Interval = [−1.73, −0.45]), which was significantly higher than for perceptually related (interaction contrast: $t(50) = -4.29$, $p = 2e-04$, $d = -0.58$, 95% Confidence Interval = [−0.85, −0.32]; two-sample $t$-test: $t(50) = -1.22$, $p = 0.226$, $d = -0.26$, 95% Confidence

Interval = [−0.69, 0.16]) or entirely novel, unrelated lures (interaction contrast: $t(50) = -2.68$, $p = 0.030$, $d = -0.47$, 95% Confidence Interval = [−0.82, −0.13]; two-sample $t$-test: $t(50) = -3.32$, $p = 0.002$, $d = -1.09$, 95% Confidence Interval = [−1.73, −0.45]). Thus, after a delay of 28d, 52.78% of all new pictures which were incorrectly endorsed as 'old' were semantically related, while only 23.14% and 24.08% were perceptually related or unrelated to the encoded pictures, respectively. This pattern of results suggests a semantic memory transformation over time. Our results did not suggest a statistically significant difference in FAs for perceptually related items compared to unrelated items at both 1d (paired $t$-test: $t(50) = -2.31$, $p = 0.073$, $d = -0.34$, 95% Confidence Interval = [−0.62, −0.05]) and 28d after encoding (paired $t$-test: $t(50) = -0.88$, $p = 0.767$, $d = -0.11$, 95% Confidence Interval = [−0.37, 0.14]).

Interestingly, this semantization over time was significantly more pronounced for emotionally negative compared to neutral pictures (delay × emotion × lure type: $F(1.96, 97.98) = 4.27$, $p = 0.017$, $\eta_p^2 = 0.08$, 95% Confidence Interval: [0.01, 0.21]), resulting in a significantly higher difference in FAs between emotionally negative and neutral semantically related lures at 28d (paired $t$-test: $t(50) = -2.72$, $p = 0.009$, $d = -0.58$, 95% Confidence Interval = [−1.00, −0.16]), compared to 1d (interaction contrast: $t(50) = 2.88$, $p = 0.006$, $d = 0.52$, 95% Confidence Interval = [0.17, 0.88]; paired $t$-test: $t(50) = 1.36$, $p = 0.181$, $d = 0.25$, 95% Confidence Interval = [−0.11, 0.6]). To follow up on this three-way interaction, we further analyzed the FAs by a separate ANOVA per lure type, each with the factors delay and emotion. These analyses confirmed a significant emotionality-dependent increase in the FA rate in the 28d-group compared to the 1d-group selectively for semantically related lures (delay × emotion: $F(1, 50) = 8.30$, $p = 0.006$, $\eta_p^2 = 0.14$, 95% Confidence Interval: [0.02, 0.34]) and did not indicate a statistically significant interaction effect for unrelated lures (delay × emotion:

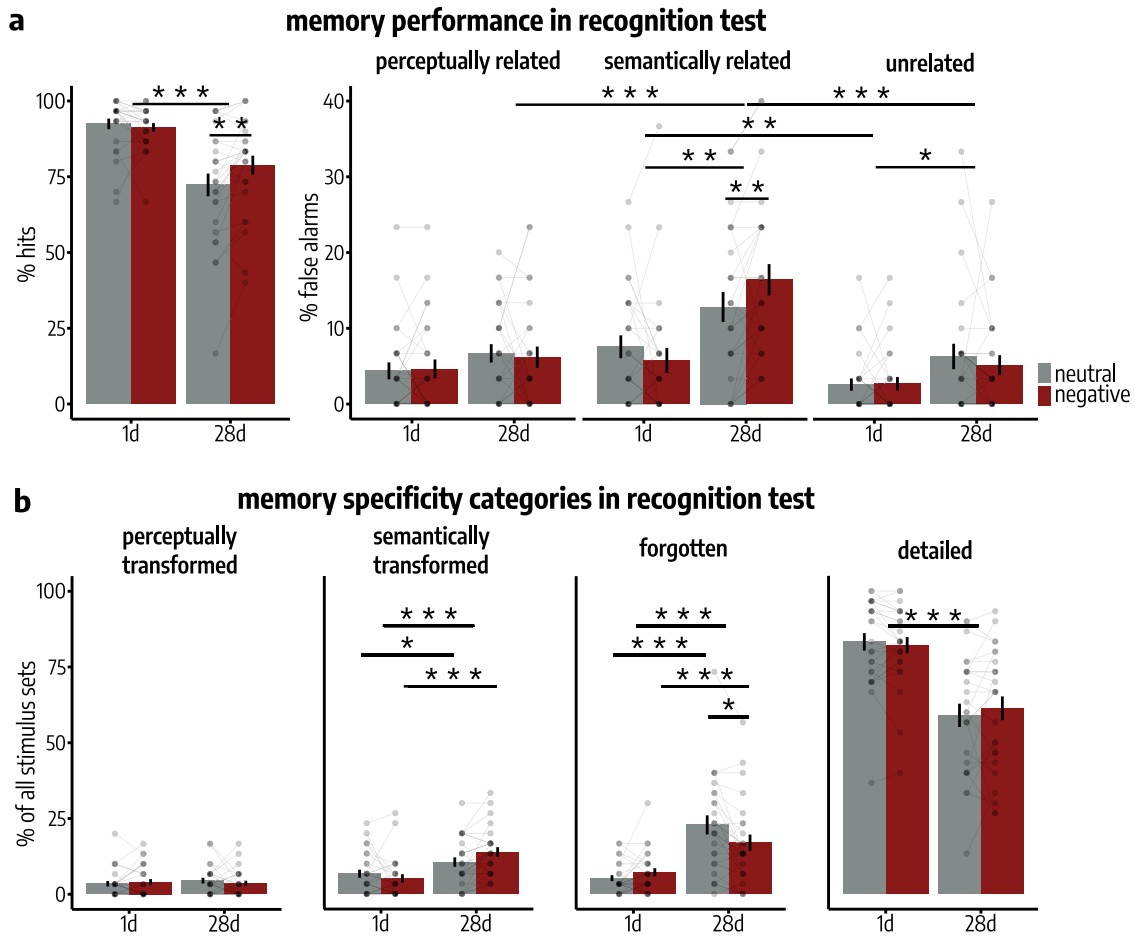

**Fig. 2 | Memory performance during recognition testing based on stimulus categories. a** Left: The decrease in hits from 1d to 28d after encoding (main effect delay: $p = 0.003$) was significantly higher for emotionally neutral than negative items (delay × emotion: $p = 0.004$; mixed ANOVA). Right: The increase in false alarms (FAs) from 1d to 28d after encoding (main effect delay: $p = 0.012$) was significantly higher for lures that were semantically related to the encoded pictures, compared to perceptually related (interaction contrast: $p = 2e{-}04$) or unrelated lures (interaction contrast: $p = 0.030$; delay × lure type: $p = 7e{-}04$). This semantization of memories over time was significantly higher for emotionally negative compared to neutral items (interaction contrast: $p = 0.006$; delay × lure type × emotion: $p = 0.017$; mixed ANOVA). All $n = 52$ participants. Bars represent mean ± SEM. Individual data points indicate the percentage of the 30 items per participant, emotion and item type which were correctly (left) or incorrectly (right) endorsed as

'old'. **b** Individual items were significantly more likely to be semantically transformed (main effect delay: $p = 0.030$), but not significantly more likely to be perceptually transformed in the 28d- compared to the 1d-group (all $p > 0.293$). Accordingly, detailed memory decreased with increasing delay after encoding (main effect delay: $p = 1e{-}07$). Moreover, emotionally negative memories were more robust against forgetting over time (delay × emotion: $p = 0.003$), but, again, more often semantically transformed than neutral ones (delay × emotion: $p = 0.014$; binomial generalized linear mixed models; all $n = 52$ participants). Bars represent mean ± SEM. Connected dots represent individual data points. All post-hoc tests were applied on estimated marginal means with Šidák correction for multiple comparisons. All reported $p$-values are two-tailed. Source data are provided as Source Data file. *$p < 0.050$; **$p < 0.010$; ***$p < 0.001$.

$F_{(1, 50)} = 0.54$, $p = 0.467$, $\eta_p^2 = 0.01$, 95% Confidence Interval: [3e−05, 0.13]) or perceptually related lures (delay × emotion: $F_{(1, 50)} = 0.23$, $p = 0.637$, $\eta_p^2 = 0.003$, 95% Confidence Interval: [2e−05, 0.11]).

Weighting the FAs by level of confidence (×1 = 'rather old', ×2 = 'definitely old') before analyzing them by means of a mixed ANOVA with the factors delay (1d vs. 28d), lure type (1d vs. 28d) and emotion (neutral vs. negative), did not change our pattern of results regarding delay-dependent effects on memory specificity (delay × lure type × emotion: $F_{(1.96, 98.12)} = 5.57$, $p = 0.005$, $\eta_p^2 = 0.10$, 95% Confidence Interval: [0.02, 0.24]; delay × lure type: $F_{(1.50, 75.19)} = 8.83$, $p = 0.001$, $\eta_p^2 = 0.15$, 95% Confidence Interval: [0.04, 0.32]; main effect lure type: $F_{(1.50, 75.19)} = 37.45$, $p = 3e{-}10$, $\eta_p^2 = 0.43$, 95% Confidence Interval: [0.28, 0.58]; main effect delay: $F_{(1, 50)} = 5.45$, $p = 0.024$, $\eta_p^2 = 0.10$, 95% Confidence Interval: [0.004, 0.29]; see Supplementary Fig. 3), indicating that our finding of an emotionally enhanced memory semantization in the course of time-dependent memory transformation was not significantly influenced by the confidence of FAs. Moreover, analyzing the confidence associated with FAs by means of binomial generalized linear

mixed models (LMMs) did not reveal any significant main effect or interaction of the predictors delay and emotion, neither for semantically related (all $p > 0.455$), perceptually related (all $p > 0.131$) nor for unrelated lures (all $p > 0.448$; see Supplementary Table 2).

While the previous analyses showed a time-dependent increase in FAs depending on the lure type, the correspondence of each originally encoded picture to precisely one perceptually related and one semantically related lure during memory testing furthermore allowed us to analyze the response pattern at the level of each individual set of related stimuli to assess the extent of detailed, semantically transformed, perceptually transformed or entirely forgotten memories[19]. For this, we categorized the responses for each of the 60 related stimulus sets as either detailed, semantically transformed, perceptually transformed, or forgotten and analyzed the occurrence of each specificity category by means of binomial generalized LMMs with delay (1d vs. 28d), emotion (neutral vs. negative) and their interactions as fixed effects and the random intercept of participants and stimulus sets. Memories were classified as detailed when participants endorsed

## a   relatedness between encoded items and lures

## b   individually perceived relatedness and false alarms

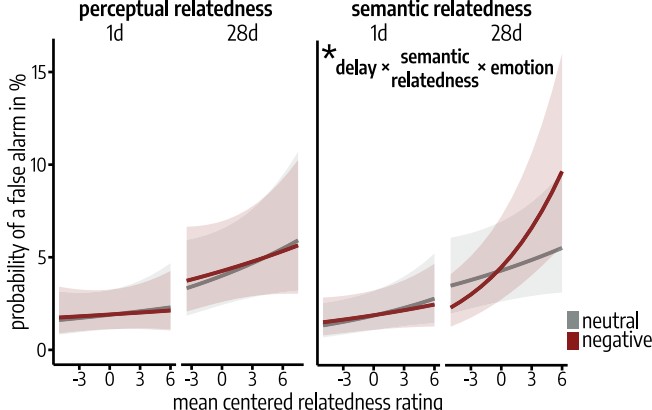

**Fig. 3 | Individually perceived relatedness and memory specificity.**
**a** Participant's relatedness ratings confirmed that semantically related items were perceived as significantly more semantically related to the corresponding old picture than perceptually related (paired $t$-test: $p < 9e{-}99$) and unrelated lures (paired $t$-test: $p < 9e{-}99$; main effect lure type on semantic relatedness: $p = 1e{-}43$) and that perceptually related lures were perceived as significantly more perceptually related to their corresponding old picture than unrelated lures (paired $t$-test: $p < 9e{-}99$; main effect lure type on perceptual relatedness: $p = 6e{-}32$; mixed ANOVAs; all $n = 52$ participants). Bars represent mean ± SEM. Connected dots represent individual data points. **b** Taking these individual relatedness ratings into

account when analyzing false alarms (FAs) by means of a binomial generalized linear mixed model (gLMM), confirmed that the delay-dependent increase in FAs (main effect delay: $p = 0.016$) was primarily driven by the semantic relatedness, specifically for emotionally negative stimuli (delay × semantic relatedness × emotion: $p = 0.018$). $N = 52$ participants. Lines represent predicted probabilities for FAs as estimated by the binomial gLMM, with error bands indicating the 95% Confidence Interval for these predicted probabilities. All post-hoc tests were applied on estimated marginal means with Šidák correction for multiple comparisons. All reported $p$-values are two-tailed. Source data are provided as Source Data file. $*p < 0.050$; $***p < 0.001$.

solely the originally encoded pictures as 'old' but not the semantically or perceptually related lures. If participants endorsed the semantically related lures but not the perceptually related lures, the respective memories were classified as being semantically transformed. Conversely, if participants endorsed the perceptually related lures but not the semantically related lures, the memories were classified as perceptually transformed. If participants endorsed neither the old nor the semantically or perceptually related items, the respective memories were classified as 'forgotten'. Thus, all 60 items per specificity category and participant are included in each analysis except of trials in which participants missed to indicate their memory for the previously presented item (missed responses), which on average led to only 0.95% (SEM = 0.44%) of missing data points per participant (no significant difference between delay groups; two-sample $t$-test: $t(31.20) = -1.07$, $p = 0.294$, $d = -0.30$, 95% Confidence Interval = $[-0.86, 0.26]$; see Supplementary Table 3 for an overview of the number of stimulus sets per category). Compared to the 1d-group, participants of the 28d-group had significantly fewer detailed (main effect delay: $z = -5.29$, $p = 1e{-}07$, $\beta = -1.51$, 95% Confidence Interval: $[-2.07, -0.95]$) and more forgotten memories (main effect delay: $z = 5.75$, $p = 9e{-}09$, $\beta = 1.79$, 95% Confidence Interval: $[1.18, 2.41]$; see Fig. 2b). Importantly, the 28d-group showed also significantly more semantically transformed memories than the 1d-group (main effect delay: $z = 2.17$, $p = 0.030$, $\beta = 0.64$, 95% Confidence Interval: $[0.06, 1.22]$) without a statistically significant increase in perceptually transformed memories (all $p > 0.293$; see Supplementary Table 4). Again, the nature of the time-dependent changes in memory was critically dependent on the emotionality of the items: Over time, significantly fewer emotionally negative pictures were forgotten than neutral ones (delay × emotion: $z = -3.00$, $p = 0.003$, $\beta = -0.75$, 95% Confidence Interval: $[-1.25, -0.26]$). Even more importantly, emotionally negative pictures were significantly more often semantically transformed over time ($z$-test: $z = -4.31$, $p = 2e{-}05$, $d = -1.31$, 95% Confidence Interval: $[-1.90, -0.71]$) than neutral ones ($z$-test: $z = -2.17$, $p = 0.030$, $d = -0.64$, 95% Confidence Interval: $[-1.22, -0.06]$; delay × emotion: $z = 2.46$, $p = 0.014$, $\beta = 0.66$, 95% Confidence Interval: $[0.14, 1.19]$), in line with findings suggesting that superior memory for emotional material, indicated

here by a slower forgetting rate, may come at the cost of reduced memory specificity[19,23-25].

Participants' relatedness ratings on Day 3 confirmed the results of our behavioral pilot study (see methods section and Supplementary Fig. 1) that semantically related lures were perceived as being significantly more semantically related ($M = 9.20$, SEM = 0.10) to the corresponding old picture than perceptually related ($M = 2.30$, SEM = 0.16; paired $t$-test: $t(50) = -33.41$, $p < 9e{-}99$, $d = -4.59$, 95% Confidence Interval = $[-4.86, -4.32]$) and unrelated lures ($M = 1.72$, SEM = 0.16; paired $t$-test: $t(50) = -36.87$, $p < 9e{-}99$, $d = -5.02$, 95% Confidence Interval = $[-5.28, -4.75]$; main effect lure type on semantic relatedness: $F(1.22, 60.96) = 1157.08$, $p = 1e{-}43$, $\eta_p^2 = 0.96$, 95% Confidence Interval: $[0.94, 0.97]$; see Fig. 3a, Supplementary Fig. 4 and Supplementary Table 5). Perceptually related lures were perceived as being significantly more perceptually related (M = 6.09, SEM = 0.21) to their corresponding old picture than unrelated lures ($M = 1.74$, SEM = 0.17; paired $t$-test: $t(50) = -22.4$, $p < 9e{-}99$, $d = -3.05$, 95% Confidence Interval = $[-3.32, -2.78]$; main effect lure type on perceptual relatedness: $F(1.45, 72.71) = 201.81$, $p = 6e{-}32$, $\eta_p^2 = 0.80$, 95% Confidence Interval: $[0.74, 0.85]$). As expected, semantically related lures were also rated higher in perceptual relatedness to their corresponding old picture ($M = 5.50$, SEM = 0.20) compared to unrelated lures (paired $t$-test: $t(50) = -16.00$, $p < 9e{-}99$, $d = -2.18$, 95% Confidence Interval = $[-2.44, -1.91]$). Importantly, perceptually related lures were rated as significantly higher in perceptual than in semantic relatedness to their corresponding old image (paired $t$-test: $t(51) = 16.67$, $p = 3e{-}22$, $d = 3.25$, 95% Confidence Interval = $[2.66, 3.83]$) while semantically related items were rated as significantly more semantically than perceptually related to their corresponding old image (paired $t$-test: $t(51) = -16.38$, $p = 6e{-}22$, $d = -2.83$, 95% Confidence Interval = $[-3.37, -2.28]$).

The individual stimulus relatedness ratings on Day 3 further allowed us to analyze FAs by means of a binomial generalized LMM with the factors delay (1d vs. 28d), emotion (neutral vs. negative), semantic relatedness rating, perceptual relatedness rating and their interactions as fixed effects and the random intercept of participants and stimuli. This analysis showed, in line with the categorical analyses

above, a time-dependent increase in FAs that was primarily driven by the semantic relatedness, which affected the probability of a FA in particular for emotionally negative stimuli (delay × semantic relatedness × emotion: $z = 2.36$, $p = 0.018$, $\beta = 0.12$, 95% Confidence Interval = [0.02, 0.21]; main effect delay: $z = 2.40$, $p = 0.016$, $\beta = 0.85$, 95% Confidence Interval = [0.16, 1.55]; main effect semantic relatedness: $z = 2.04$, $p = 0.041$, $\beta = 0.07$, 95% Confidence Interval = [0.003, 0.13]; Fig. 3b). We obtained no statistically significant effect of the individual perceptual relatedness ratings on FAs and their increase over time (all $p > 0.127$; see Supplementary Table 6).

As semantically related items are usually also high in perceptual relatedness to original stimuli, we additionally analyzed whether the delay-dependent increase in FAs for semantically related items was equally evident in semantically related lures low ($\leq 5$) vs. high ($> 5$) in perceptual relatedness. A generalized LMM with the factors perceptual relatedness level (low vs. high), delay (1d vs. 28d) and emotion (neutral vs. negative) and the random intercept of participants and stimuli confirmed our previous finding of an emotionally enhanced increase in the probability for a FA for semantically related lures over time (delay × emotion: $\beta = 0.93$, $p = 0.029$, $z = 2.19$, 95% Confidence Interval = [0.09, 1.85]). This analysis did not indicate any influence of the level of perceptual relatedness of a semantically related stimulus to its corresponding original item on FAs (all $p > 0.215$; see Supplementary Table 7).

In sum, our behavioral data demonstrate that memories are semantically transformed over time while we found no statistically significant evidence for a perceptual memory transformation. This time-dependent semantization of memories was further consistently more pronounced for emotionally negative than for neutral stimuli.

## Distinct pattern representations of encoded events in the anterior hippocampus decrease over time

In order to examine the neural mechanisms involved in the semantic transformation of memories over time, we leveraged model-based Representational Similarity Analyses (RSAs)[19,29,30] assessing how the similarity between activation patterns of encoded items and different lure types (semantically related vs. perceptually related vs. unrelated) at memory testing changes in the course of memory transformation. Here, neural representational similarity matrices (RSMs) were compared to three conceptual model RSMs (see Fig. 2a), each predicting different similarity patterns between old items and the different lure types at memory testing: (i) similar representations for old pictures that are distinct from patterns for all novel stimuli (model 1: 'old items are distinct from all lures'), (ii) similar representations between old items and semantically related lures which are distinct from perceptually related and unrelated lures (model 2: 'old and semantically related items are similar') and (iii) similar representations between old items and perceptually related lures, which are distinct from semantically related and unrelated lures (model 3: 'old and perceptually related items are similar'). Note that for all models we expected old items to be represented more similarly, as they should equally initiate recognition processes in neural areas relevant for memory representations that, in case of recent, specific memory, should be distinct from all lures (model 1), or, in case of transformed memory representations, similar to either semantically (model 2), or perceptually (model 3) related lures. Based on recent evidence[19], we hypothesized that the anterior hippocampus is particularly relevant for the specificity of recent memories while the posterior hippocampus represents remote, semantically transformed memories. Accordingly, the anterior hippocampus should reflect distinct representations (model 1) at a short delay, but this representation should decrease over time, while we expected the posterior hippocampus to represent semantically transformed memory that should increase over time (model 2). A mixed ANOVA with the factors delay (1d vs. 28d), emotion (neutral vs. negative), model (1: 'old items are distinct from all lures' vs. 2: 'old and

semantically related items are similar' vs. 3: 'old and perceptually related items are similar') and hippocampal long axis (anterior vs. posterior) revealed a significant delay × model × long axis interaction ($F(1.55, 75.88) = 5.36$, $p_{corr} = 0.024$, $\eta_p^2 = 0.10$, 95% Confidence Interval: [0.02, 0.25]) and a delay × model × long axis × emotion interaction ($F(1.68, 82.19) = 4.64$, $p_{corr} = 0.034$, $\eta_p^2 = 0.09$, 95% Confidence Interval: [0.01, 0.23]; see Fig. 4b). Note that one extreme outlier (28d group) was excluded from this analysis. Post-hoc tests confirmed a significant decrease in recognition processes for the encoded material (model 1) over time in the anterior hippocampus (two-sample $t$-test: $t(49) = 2.42$, $p = 0.020$, $d = 0.36$, 95% Confidence Interval = [0.07, 0.64]), which was significant for emotionally negative items (two-sample $t$-test: $t(49) = 2.40$, $p = 0.021$, $d = 0.46$, 95% Confidence Interval = [0.08, 0.83]), while emotionally neutral items did not show a statistically significant decrease in model fit over time (two-sample $t$-test: $t(49) = 1.05$, $p = 0.297$, $d = 0.25$, 95% Confidence Interval = [−0.22, 0.73]). Interestingly, the anterior hippocampus also showed a delay-dependent decrease in perceptually similar memory representations for neutral items (model 3, two-sample $t$-test: $t(49) = 2.40$, $p = 0.003$, 0.94, 95% Confidence Interval = [0.34, 1.54]) indicating a time-dependent decrease in the representation of perceptual details in the anterior hippocampus for those items. Neither the anterior hippocampus (model 2: $t(49) = −0.01$, $p = 0.989$, $d = −3e−03$, 95% Confidence Interval = [−0.39, 0.38]) nor the posterior hippocampus ($t(49) = 1.11$, $p = 0.271$, $d = 0.23$, 95% Confidence Interval = [−0.17, 0.62]) showed a statistically significant delay-dependent change in the fit to the model reflecting semantically transformed pattern representations.

Together, these data indicate that the anterior hippocampus represents recently encoded events in a detailed manner, including perceptual features, and that these anterior hippocampal representations decrease over time, while our results did not yield reliable evidence for a more gist-like, transformed pattern representation in the anterior hippocampus, neither at the 1d- nor at the 28d-delay.

## Semantically transformed representations of encoded events increase in prefrontal and parietal cortices over time

While the hippocampus has been implicated to be particularly important for recently encoded and specific memories in previous studies[16,18], neocortical regions are assumed to become more relevant for remote memory[9,16,18,31]. Specifically, the vmPFC[8–10], IFG[4,5], aCC[2,11–13], angular gyrus and the precuneus[14,15] have been associated with the formation of long-term memories. Thus, we analyzed time-dependent memory transformation processes in these neocortical long-term memory regions. We first performed a delay (1d vs. 28d) × model (1: 'old items are distinct from all lures' vs. 2: 'old and semantically related items are similar' vs. 3: 'old and perceptually related items are similar') × emotion (neutral vs. negative) ANOVA using a combined mask, including the vmPFC, IFG, aCC, angular gyrus and precuneus, as we expected a similar increase in transformed memory representations over time in all of those neocortical regions. This analysis showed a significant increase in representational similarity between old items and semantically related lures in the 28d- compared to the 1d-group in the neocortex (model 2; two-sample $t$-test: $t(50) = −2.04$, $p = 0.047$, $d = −0.62$, 95% Confidence Interval = [−1.21, −0.02]), and no statistically significant delay-dependent change in the fits to models reflecting distinct (model 1; two-sample $t$-test: $t(50) = −1.62$, $p = 0.111$, $d = −0.38$, 95% Confidence Interval = [−0.84, 0.08]) or perceptually similar memory representations (model 3; two-sample $t$-test: $t(50) = 0.83$, $p = 0.412$, $d = 0.13$, 95% Confidence Interval = [−0.18, 0.45]; delay × model: $F(1.48, 74.03) = 7.1$, $p = 0.004$, $\eta_p^2 = 0.12$, 95% Confidence Interval: [0.03, 0.29]; main effect model: $F(1.48, 74.03) = 19.11$, $p = 3e−06$, $\eta_p^2 = 0.28$, 95% Confidence Interval: [0.13, 0.44]; see Fig. 5). Accordingly, neocortical activity patterns during memory testing showed a

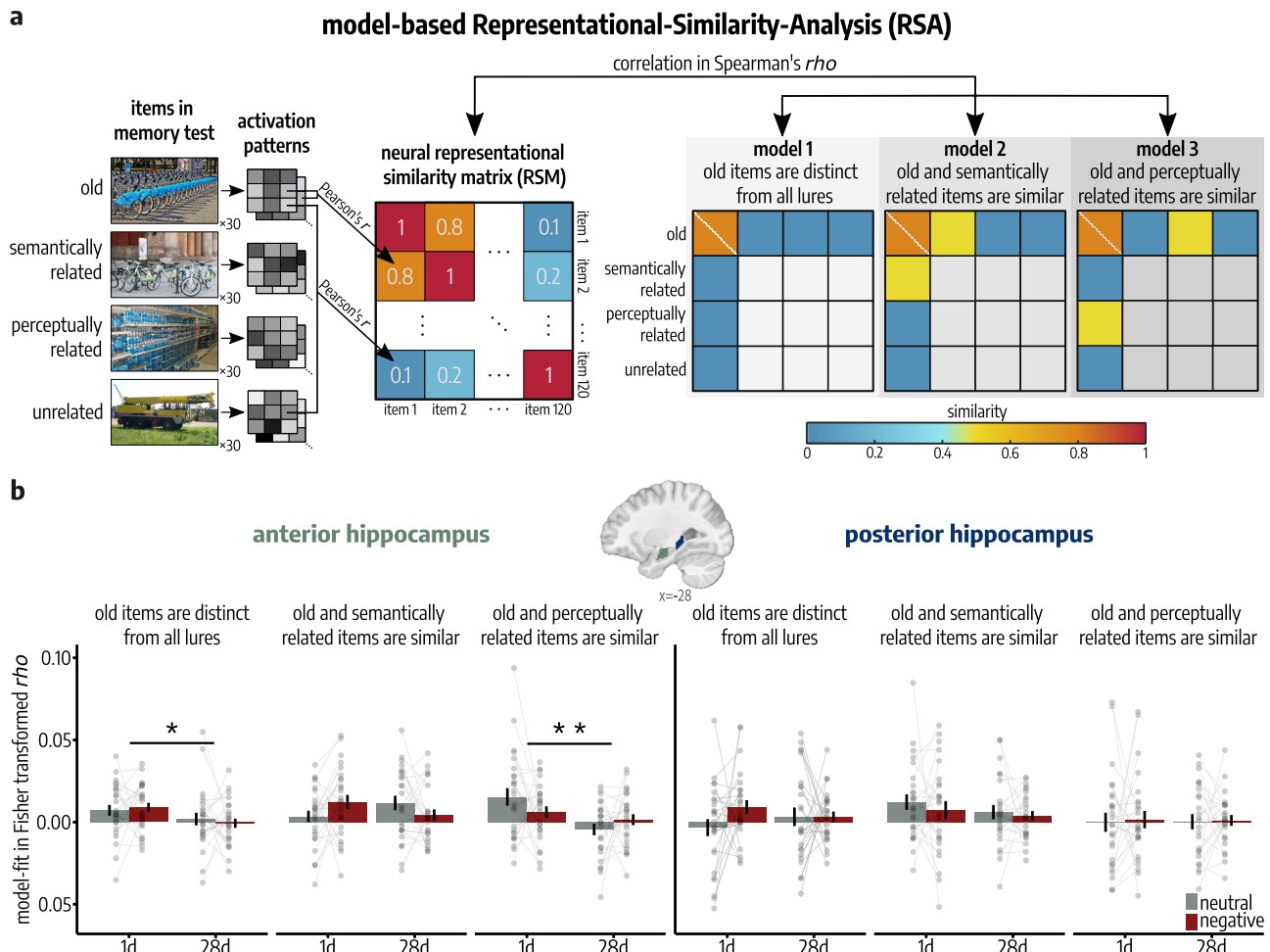

**Fig. 4 | Computational approach for model-based RSA analyses and results along the hippocampal anterior-posterior axis. a** Schematic overview over the creation of a neural RSM for emotionally neutral items with exemplary correlation values. Each neural RSM per region of interest (ROI), emotion category and subject was compared to three conceptual models. All depicted images are licensed under Creative Commons BY-SA License: image representing 'old' item is courtesy of W. Bulach (https://commons.wikimedia.org/wiki/File:00_2141_Bicycle-sharing_systems_-_Sweden.jpg; edited), image representing 'semantically related' item is courtesy of Matti Blume (https://commons.wikimedia.org/wiki/File:Bike_share_2019,_Berlin_(P1080139).jpg; edited), image representing 'perceptually related' item is courtesy of Ivy Main (https://fi.m.wikipedia.org/wiki/Tiedosto:Bottled_water_in_supermarket.JPG; edited), image representing 'unrelated' item is courtesy of Hannes Drexl (https://commons.wikimedia.org/wiki/File:Autokran_Seite.jpg?

uselang=de; unchanged). **b** In the left anterior hippocampus, specifically for negative items (model 1; two-sample *t*-test: *p* = 0.021), distinct representations of encoded pictures (model 1; two-sample *t*-test: *p* = 0.019) and, specifically for emotionally neutral items (model 3; two-sample *t*-test: *p* = 0.003), perceptually similar representations (model 3; two-sample *t*-test: *p* = 0.002) decreased with increasing delay after encoding (delay × long axis × model: $p_{corr}$ = 0.024; delay × emotion × long axis × model: $p_{corr}$ = 0.034; mixed ANOVA; *n* = 51 participants). Bars represent mean ± SEM. If analyses were repeated for both hemispheres, Bonferroni-corrected *p*-values ($p_{corr}$) are reported. All reported *p*-values are two-tailed. All post-hoc tests were applied on estimated marginal means with Šidák correction for multiple comparisons. Regions of interest are visualized on a sagittal section of a T1-weighted template[82] in MNI-152 space. Source data are provided as Source Data file. **p* < 0.050; ***p* < 0.010.

significantly higher fit to model 2 ('old and semantically related items are similar') than to both other models in the 28d-group (paired *t*-tests; model 1: *t*(50) = −4.02, *p* = 6*e*−04, *d* = −0.61, 95% Confidence Interval = [−0.91, −0.31]; model 3: *t*(50) = 5.50, *p* = 4*e*−06, *d* = 0.93, 95% Confidence Interval = [0.60, 1.26]) while there was no statistically significant difference in fit to either model in the 1d-group (paired *t*-tests; model 1: *t*(50) = −2.15, *p* = 0.106, *d* = −0.27, 95% Confidence Interval = [−0.52, −0.02]; model 3: *t*(50) = 1.36, *p* = 0.446, *d* = 0.19, 95% Confidence Interval = [−0.08, 0.46]). Thus, this analysis indicates the formation of semantically transformed representations of encoded events in the neocortex over time.

To investigate whether this time-dependent memory semantization was equally evident in all individual neocortical regions, we analyzed the fit of the RSM for each individual neocortical ROI to the model reflecting semantically transformed pattern representations (model 2) by means of mixed ANOVAs with the factors delay and emotion (see Fig. 5). This analysis confirmed a delay-dependent

increase in representational similarity between old items and semantically related lures in the vmPFC (main effect delay: *F*(1, 50) = 4.19, *p* = 0.046, $\eta_p^2$ = 0.08, 95% Confidence Interval: [0.001, 0.26]) and right angular gyrus (main effect delay: *F*(1, 50) = 8.34, $p_{corr}$ = 0.011, $\eta_p^2$ = 0.14, 95% Confidence Interval: [0.02, 0.34]). This analysis did not indicate a statistically significant delay-dependent change in similarity between model 2 and pattern representations in the precuneus (main effect delay: *F*(1, 50) = 3.93, *p* = 0.053, $\eta_p^2$ = 0.07, 95% Confidence Interval: [0.00, 0.25]; *F*(1,50) = 0.01, *p* = 0.943, $\eta_p^2$ = 1*e*−04, 95% Confidence Interval: [2*e*−05, 0.10]), IFG (main effect delay: *F*(1,50) = 2.25, *p* = 0.140, $\eta_p^2$ = 0.04, 95% Confidence Interval: [2*e*−04, 0.21]; delay × emotion: *F*(1,50) = 1.8, *p* = 0.186, $\eta_p^2$ = 0.03, 95% Confidence Interval: [1*e*−04, 0.19]), and aCC (main effect delay: *F*(1, 50) = 1.15, *p* = 0.289, $\eta_p^2$ = 0.02, 95% Confidence Interval: [6*e*−05, 0.16]; delay × emotion: *F*(1,50) = 1.41, *p* = 0.240, $\eta_p^2$ = 0.03, 95% Confidence Interval: [8*e*−05, 0.18]).

Furthermore, we repeated this model-based RSA in the bilateral occipital pole and Heschl's gyrus as neocortical control regions for

## model-based Representation-Similarity-Analysis (RSA) in neocortical long-term memory storage sites

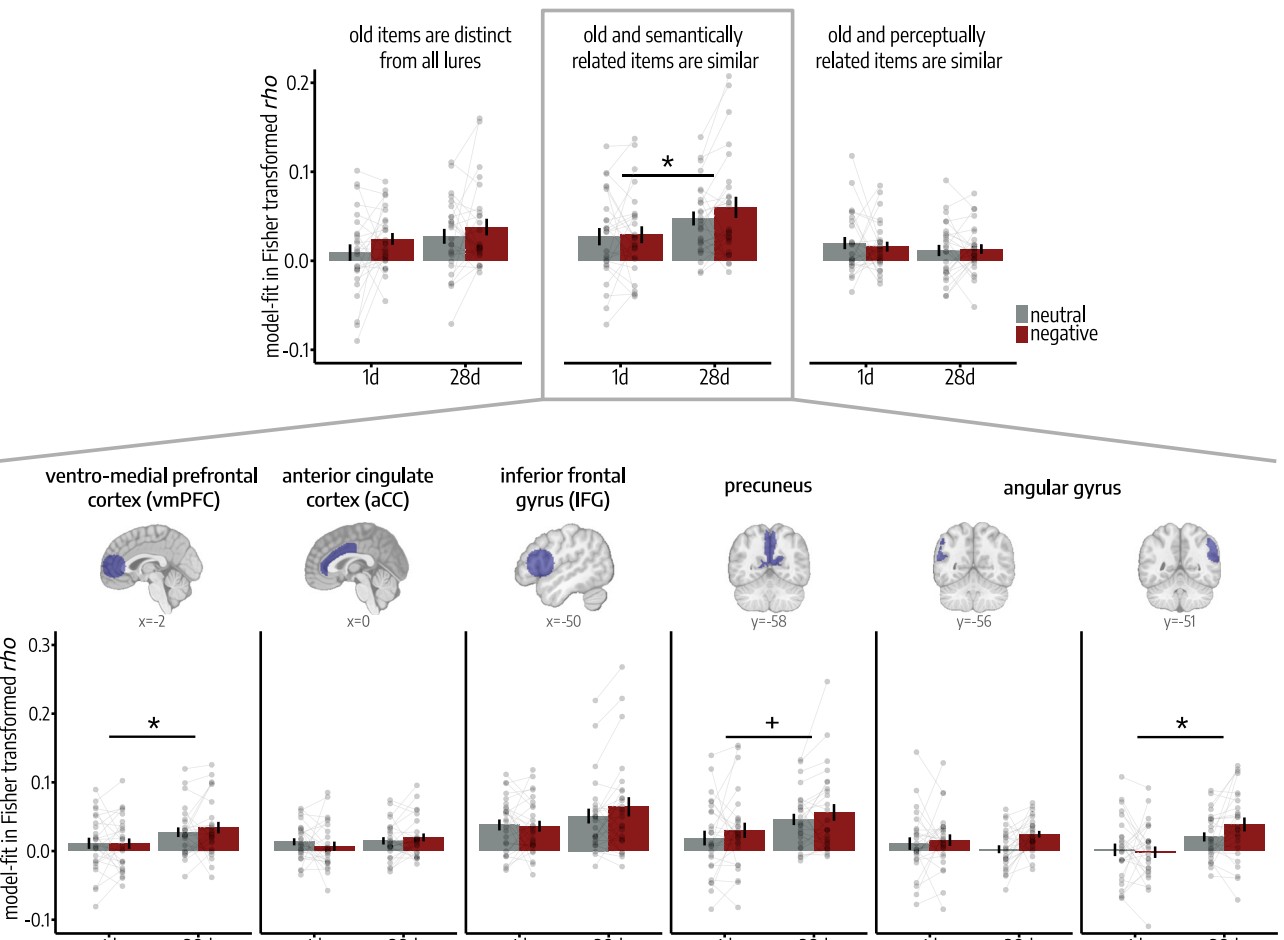

**Fig. 5 | Model-based RSA results in neocortical long-term memory storage sites.** *Upper panel:* Pattern representations in a combined ROI including long-term memory cortices (vmPFC, IFG, aCC, angular gyrus and precuneus) were semantically (model 2; two-sample *t*-test: *p* = 0.047) transformed over time, while there was no statistically significant effect for the model testing for perceptually transformed representation patterns (model 3; two-sample *t*-test: *p* = 0.412; delay × model: *p* = 0.004; mixed ANOVA). *Lower panel:* Post-hoc testing revealed that this time-dependent semantization of pattern representations (model 2) was specific to the vmPFC (main effect delay: *p* = 0.046) and right angular gyrus (main effect delay: $p_{corr}$ = 0.010; mixed ANOVAs; *n* = 52 participants). Bars represent mean ± SEM. If analyses were repeated for both hemispheres, Bonferroni-corrected *p*-values ($p_{corr}$) are reported. All reported *p*-values are two-tailed. All post-hoc tests were applied on estimated marginal means with Šidák correction for multiple comparisons. Regions of interest (ROIs) are visualized on sagittal (prefrontal ROIs) and axial (parietal ROIs) sections of a T1-weighted template[82] in MNI-152 space. Source data are provided as Source Data file. ⁺*p* < 0.060; *p* < 0.050.

which we did not expect any statistically significant increase in transformed memory representations over time. Analyzing activation patterns in those regions by means of delay (1d vs. 28d) × model (1: 'old items are distinct from all lures' vs. 2: 'old and semantically related items are similar' vs. 3: 'old and perceptually related items are similar') × emotion (neutral vs. negative) ANOVAs did not indicate a statistically significant time-dependent change in fit of pattern representations, neither in the occipital pole (delay × model: *F*(1.83, 91.39) = 0.87, *p* = 0.415, $\eta_p^2$ = 0.02, 95% Confidence Interval: [0.001, 0.12]; delay × emotion × model: *F* (2.00, 99.82) = 0.47, *p* = 0.624, $\eta_p^2$ = 9*e*−03, 95% Confidence Interval: [0.001, 0.10]) nor in Heschl's gyrus (delay × model: *F*(1.38, 68.96) = 0.32, *p* = 0.645, $\eta_p^2$ = 6*e*−03, 95% Confidence Interval: [2*e*−04, 0.08]; delay × emotion × model: *F*(1.69, 84.5) = 0.48, *p* = 0.587, $\eta_p^2$ = 1*e*−02, 95% Confidence Interval: [5*e*−04, 0.10]). Interestingly, activation patterns in the occipital pole showed an overall higher fit to model 3 ('old and perceptually related items are similar') compared to model 1 (paired *t*-test: *t*(50) = −4.87, *p* = 3*e*−05, *d* = −0.5, 95% Confidence Interval = [−0.70, −0.30]) as well as model 2 (paired *t*-test: *t*(50) = −3.85, *p* = 0.001, *d* = −0.49, 95%

Confidence Interval = [−0.73, −0.24]) without a statistically significant effect of temporal delay (main effect model: *F*(1.83, 91.39) = 13.26, *p* = 2*e*−05, $\eta_p^2$ = 0.21, 95% Confidence Interval: [0.09, 0.36]). This finding most likely reflects the processing of overlapping visual features in old and perceptually related images in this region.

Our model-based analyses thus indicate that semantically transformed representations of previously encoded events emerge in prefrontal and posterior parietal cortices in the course of memory transformation while we did not observe any credible evidence for a perceptual transformation in these regions.

### Posterior hippocampal memory reinstatement increases over time

While our model-based approach assessed the time-dependent change in representational similarity between encoded and new item categories at memory testing, we further analyzed the reactivation of individual items during memory test, i.e., Encoding-Retrieval Similarity (ERS), as a measure of trial-specific memory reinstatement[20,32–37]. For this, we computed the similarity (Pearson's *r*) between activation

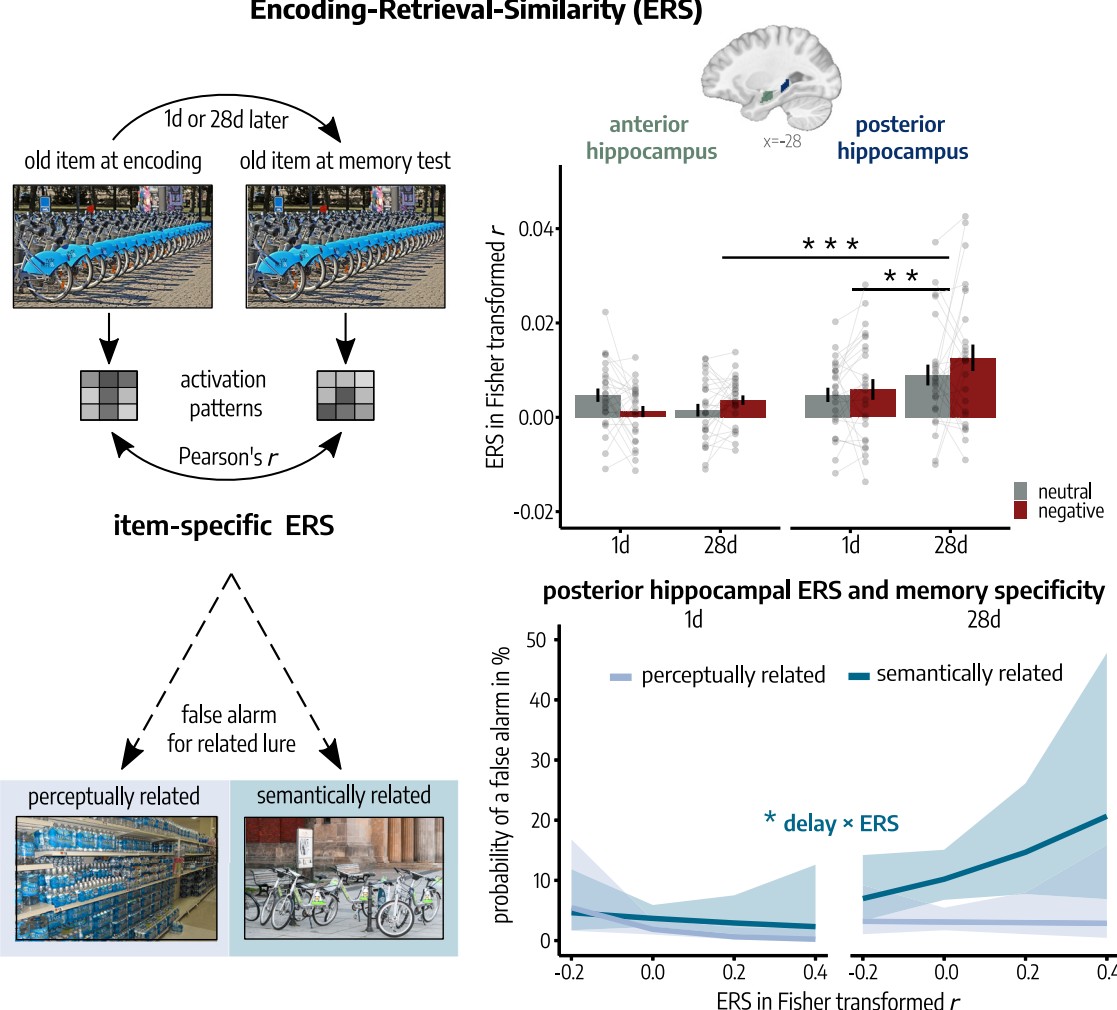

**Fig. 6 | Time-dependent changes in memory reinstatement along the hippo-campal anterior-posterior axis.** *Upper Panel:* Posterior hippocampal memory reinstatement increased over time (paired *t*-test: *p* = 0.004; linear mixed model, LMM: delay × long axis: $p_{corr}$ = 0.038). Bars represent mean ± SEM. Connected dots represent individual data points. *Lower Panel:* Posterior hippocampal ERS showed a significantly higher positive association with false alarms (FAs) for corresponding semantically related lures at a delay of 28d compared to 1d after encoding (delay × ERS: *p* = 0.048) but no statistically significant association with FAs for perceptually related lures (all *p* > 0.184, generalized LMMs). These results indicate that posterior hippocampal reinstatement of remote memories is associated with the semantic gist of the original memory. Lines represent predicted probabilities for FAs as estimated by the binomial generalized LMM, with error bands indicating the 95% Confidence Interval for these predicted probabilities. All *n* = 52 participants. If analyses were repeated for both hemispheres, Bonferroni-corrected *p*-values ($p_{corr}$) are reported. All reported *p*-values are two-tailed. All post-hoc tests were applied on estimated marginal means with Šidák correction for multiple comparisons. Regions of interest are visualized on a sagittal section of a T1-weighted template[82] in MNI-152 space. All depicted photographs are licensed under Creative Commons BY-SA License: image representing 'old' item is courtesy of W. Bulach (https://commons.wikimedia.org/wiki/File:00_2141_Bicycle-sharing_systems_-_Sweden.jpg; edited), image representing 'semantically related' item is courtesy of Matti Blume (https://commons.wikimedia.org/wiki/File:Bike_share_2019,_Berlin_(P1080139).jpg; edited), image representing 'perceptually related' item is courtesy of Ivy Main (https://fi.m.wikipedia.org/wiki/Tiedosto:Bottled_water_in_supermarket.JPG; edited). Source data are provided as Source Data file. \**p* < 0.050; \*\**p* < 0.010; \*\*\**p* < 0.001.

patterns during encoding on Day 1 and activation patterns of the same item during memory testing either 1d or 28d after encoding (Day 2; see Fig. 6). Due to the crucial role of the hippocampus in the reinstatement of episodic details[38,39], we focused specifically on the hippocampus and the differentiation along its anterior-posterior axis in this ERS analysis. Based on recent data suggesting that recent, specific memories are represented by the anterior hippocampus while more gist-like memories are associated with the posterior hippocampus[19,20], we predicted that, after a longer delay after encoding, memory reinstatement should rely more on the posterior hippocampus, while the anterior hippocampus should reinstate recent, specific memories.

Time dependent changes in item-specific ERS were analyzed by means of trial-wise LMMs with the factors delay (1d vs. 28d), emotion (neutral vs. negative), long axis (anterior vs. posterior) and their interactions as fixed effects and the random intercept of participants

and stimuli. This analysis showed that the ERS changed significantly along the left hippocampal long axis depending on the delay after encoding (delay × long axis: *t*(6124) = 2.35, $p_{corr}$ = 0.038, β = 0.01, 95% Confidence Interval = [0.001, 0.01]) with a significantly greater ERS in the posterior compared to the anterior hippocampus in the 28d-group (paired *t*-test: *t*(6124) = −5.25, *p* = 2e−07, *d* = −0.19, 95% Confidence Interval = [−0.26, −0.11]), while there was no statistical difference in the 1d-group (*t*(6124) = −1.53, *p* = 0.126, *d* = 0.06, 95% Confidence Interval = [−0.13, 0.02]). No other effects approached statistical significance in this analysis (all $p_{corr}$ > 0.148; see Supplementary Table 8). Note that repeating this analysis after excluding items that were not correctly recognized (misses) did not change the result of a significant increase in posterior hippocampal ERS from 1d to 28d (delay × long axis: β = 0.01, *t*(5107.81) = 3.13, $p_{corr}$ = 0.004, 95% Confidence Interval = [0.001, 0.01]; see Supplementary Table 9).

Next, we investigated the relationship of left posterior hippocampal memory reinstatement and behavioral memory indicators. Analyzing the probability of a hit by means of a generalized LMM with ERS, emotion (neutral vs. negative), delay (1d vs. 28d) and their interaction as fixed effects and the random intercepts of participants and stimulus sets indicated that with increasing delay, posterior hippocampal memory reinstatement was significantly positively associated with the correct endorsement of an old item as 'old' (delay × ERS: $z = 2.17$, $p = 0.030$, $\beta = 7.31$, 95% Confidence Interval = [0.73, 13.89]; see Supplementary Table 10). However, correct memory could be supported by specific, detailed memory representations but also by more abstract, gist-like representations. We therefore further analyzed the specificity of the reinstated memories by taking into account the responses for corresponding related lures. Analyzing the probability of a detailed recognition (correct response for old items without FAs for related lures) by means of a binomial generalized LMM with the factors ERS, delay and emotion did not indicate a statistically significant association of ERS with detailed memory (all $p > 0.688$; see Supplementary Fig. 5 and Supplementary Table 11). However, analyzing the probability of a FA for the corresponding semantically or perceptually related lure of each old item, by means of binomial generalized LMMs with ERS, emotion, delay and their interaction as fixed effects and the random intercepts of participants and stimulus sets, indicated a significantly higher positive association of ERS with the probability of a FA for semantically related lures in the 28d- compared to the 1d-group (delay × ERS: $z = 1.98$, $p = 0.048$, $\beta = 7.16$, 95% Confidence Interval = [0.02, 14.29]; see Fig. 3). No effects including the factor ERS were statistically significant when analyzing FAs for perceptually related lures (all $p > 0.184$; see Supplementary Table 12). Furthermore, analyzing response times during memory testing, as an indicator of the attentiveness in the respective trial, by means of an LMM with delay, emotion, posterior hippocampal ERS and their interaction as fixed effects and the random intercept of participants and stimuli did not show any statistically significant main effect ($p > 0.421$) nor interaction (all $p > 0.186$) of the included factors (see Supplementary Table 13). Thus, we found no statistically reliable evidence that the delay-dependent increase in posterior hippocampal ERS might be related to attentional differences during memory testing between groups.

While ERS is computed by correlating pattern representations of individual items during encoding and memory test, i.e. old items, we furthermore explored the similarity elicited by perceptually or semantically related items at memory test and corresponding old items during encoding as a possible indicator for a reinstatement of the perceptual or semantic gist of the original memory. Analyzing the anterior-posterior hippocampal representational similarity between items at encoding and their corresponding semantically related lures by means of an LMM, the posterior hippocampus tended to show a higher reinstatement of the semantic gist of the original memory compared to the anterior hippocampus, which, however, failed to reach statistical significance (main effect long axis: $t(6124) = 1.94$, $p = 0.052$, $\beta = 0.004$, 95% Confidence Interval = [−4e−05, 0.01]; see Supplementary Fig. 6). No effect approached statistical significance when analyzing hippocampal reinstatement of the perceptual gist (all $p > 0.235$; see Supplementary Table 14). To assess whether the results of our memory reinstatement analyses were indeed specific to the hippocampus, we explored delay-dependent changes in memory reinstatement in neocortical areas that have been previously implicated in long-term memory (IFG, vmPFC, aCC, precuneus, angular gyrus) as well as sensory control ROIs (occipital pole, Heschl's gyrus) by means of LMMs with the factors delay, emotion and their interactions as fixed effects. These analyses did not indicate any statistically significant delay-dependent variations in pattern reinstatement, neither by old items (i.e. ERS; all $p > 0.164$; see Supplementary Table 15) nor by semantically (all $p > 0.243$; see Supplementary Table 16) or perceptually (all $p > 0.201$; see Supplementary Table 17) related lures.

Taken together, our memory reinstatement analyses indicated that the posterior hippocampus is associated with the reinstatement of remote memories that may be rather unspecific in nature representing the semantic gist of the original memory, while we found no statistically significant link between hippocampal ERS and a perceptual memory transformation.

## Discussion

Memories are thought to undergo a transformation over time. Here, we aimed at elucidating the nature and neural signature of the proposed time-dependent memory transformation. Specifically, we determined whether memories are semantically or perceptually transformed, which neural mechanisms are involved in this process, and whether memories for emotionally neutral and negative material are transformed in a comparable manner over time. We show that episodic memories are semantically transformed over time, while we did not obtain any credible evidence for a perceptual transformation. Our results further show that this time-dependent memory semantization is more pronounced for emotionally negative compared to neutral information. At the neural level, the transformation of memories over time was linked to a time-dependent increase in semantically transformed memory representations in prefrontal and parietal cortices. Beyond these time-dependent changes in neocortical areas, we also report significant representational changes within the hippocampus, along its anterior-posterior axis. Activation patterns that were specific to previously encoded events were represented in the anterior hippocampus, while the posterior hippocampus was associated with the reinstatement of remote memories that were rather unspecific in nature and likely to be confused with the semantic gist of the original memory, without reliable evidence for links to the perceptual gist.

Although prominent theoretical accounts of the temporal dynamics of memory postulate a transformation of memory over time[16,17], the nature of this time-dependent memory transformation remained elusive. In particular, it has been unclear whether the generalization of memories over time is semantic in nature or due to a perceptual transformation, with the latter being more in line with the common view that memories fade over time[22]. Previous studies could not distinguish between these alternatives as test materials and contexts were typically both perceptually and semantically related to the original episode. Here, we aimed at overcoming this issue by using a recognition test that included lures carrying either the semantic or the perceptual gist of the original material. Participants showed a significant time-dependent increase in the endorsement of semantically related lures over time indicating that remote memories represented the semantic gist of the original memory. For the endorsement of perceptually related lures, however, we found no credible evidence for an increase over time. Even more strikingly, when we analyzed participants' individual perceptual and semantic relatedness ratings for each of the lures, we observed that participants' subjectively perceived semantic relatedness between lure and original stimulus predicted the time-dependent increase in FAs on a trial-by-trial basis, demonstrating that remote memories were semantically transformed, while there was no statistically significant effect of the perceptual relatedness. These findings are generally consistent with core tenets of the Multiple-Trace-Theory[18] and Trace-Transformation-Theory[16,17], which suggest a semantic transformation over time. Alternatively, however, it could also be argued that previously available perceptual detail, which prevented the FAs at 1d, has been lost over time, while coarse semantic information was still available at 28d. Instead of a semantic transformation, our findings would then rather suggest forgetting of identifiable detail. This detail could pertain to the perceptual domain or to the semantic domain. In other words, both semantic and perceptual information could be encoded during initial encoding but then being forgotten at different rates over time. Interestingly, however, our finding that semantically related items induce significantly higher FAs

compared to both unrelated lures and perceptually related lures indicates that memories are, regardless of temporal delay, mostly stored in a semantic rather than in a perceptual form.

The proposed memory transformation over time has been linked to a time-dependent neural reorganization of memories. According to the classic systems consolidation theory, the hippocampal involvement in memory should decrease as memories become more and more reliant on neocortical areas over time[12,40]. While the Trace-Transformation-Theory does also assume an increased involvement of neocortical areas, in particular for transformed memories, specific memories are thought to remain hippocampus dependent[16,17]. In line with both of these theories, we obtained pattern representations that were highly specific to encoded events in the (anterior) hippocampus when the retention interval was short (i.e. 1d) and that these specific representations decreased over time, as did participants' memory specificity. In parallel, neocortical patterns emerged as the time interval after encoding increased, in particular in the vmPFC, angular gyrus and precuneus, which have been previously associated with long-term memory storage[2,4,5,8,11–15]. Notably, while we show here a time-dependent increase in the involvement of these areas, there is also recent evidence that the recruitment of parietal storage sites may be accelerated as a function of the number of retrieval attempts[14,15]. Most importantly, however, our model-based RSA data revealed that the neocortical representations that emerged over time, coded the semantic gist of the originally encoded event, again, in line with the Trace-Transformation-Theory[16,17], while we found no credible evidence for a coding of the perceptual gist.

Whereas it is commonly assumed that the time-dependent neural reorganization of memory involves a reduced hippocampal and increased neocortical contribution[14,15], there is initial evidence that there may be also a time-dependent reorganization within the hippocampus, along its anterior-posterior axis[19,33,36,41,42]. In line with this view, we report here that while pattern representations that were specific to encoded events in the anterior hippocampus decreased over time, as indicated by a time-dependent decrease in the anterior hippocampal fit to the 'old items are distinct from all lures' model, posterior hippocampal memory reinstatement (i.e. ERS) increased with time. The exact functional differentiation of the anterior and posterior portions of the hippocampus is still a matter of debate. For example, a recent theoretical account suggests the exact opposite course of memory transformation along the hippocampal long axis[16,43–45]. This account was originally based on rodent data[46] showing that firing fields of place cells in the ventral hippocampus, corresponding to the human anterior hippocampus[47], are larger than those in the dorsal hippocampus, which might translate into more abstract, large-scale anterior hippocampal representations. It is further argued that through an increased connectivity of the anterior hippocampus to prefrontal areas and of the posterior hippocampus to the posterior neocortex, both hippocampal poles might be specifically prone to represent semantic or perceptually detailed memories, respectively. However, it has been shown that rodents' ventral hippocampal cell population allows decoding the precise location, despite each individual cell only representing a larger area of the environment[48], which points to a mnemonic specificity of anterior hippocampal representations. Moreover, recent research[14,15] has revealed that the role of posterior neocortical areas connected to the posterior hippocampus, such as the precuneus and angular gyrus[49], in memory goes far beyond the mere processing of perceptual information and, instead, represent long-term memory storage sites. This is also in line with the present model-based analyses indicating that these parietal areas might represent the semantic gist of a memory, while we obtained no credible evidence for the representation of perceptual details. Moreover, our results suggest that perceptual memory features are represented in the anterior hippocampus and that those representations decline over time. Our finding that the anterior hippocampus represents specific memories is

further consistent with research implicating the anterior hippocampus with the recollection of contextual details[36], novelty detection[50], source memory specificity[42], constructing autobiographical memories[51] and detailed future event representations[52].

Although the increase in posterior hippocampal ERS over time and its direct association with our behavioral indicator of semantic transformation (i.e. FAs to semantically related lures) and the decrease in distinct representations of encoded stimuli in the anterior hippocampus over time supports the idea of a time-dependent transformation along the hippocampal anterior-posterior axis with detailed memory representations in the anterior hippocampus and remote, gist-like representations in the posterior hippocampus, it is important to note that we did not find reliable evidence for a decrease in the anterior hippocampal memory reinstatement over time. Moreover, our model-based RSA did not provide credible evidence that posterior hippocampal representations of encoded events increase in similarity to semantically related material. The absence of reliable evidence for an anterior hippocampal decrease over time or a time-dependent increase of a posterior hippocampal fit to the 'old and semantically related items are similar' model might be taken as support against the suggested differential memory transformation over time in anterior and posterior hippocampal areas. It is to be noted, however, that these seemingly discrepant findings may be owing to the different methodological approaches. Whereas the ERS measures a change in reinstatement of an individual memory at test, the model-based analysis is directed at representational changes for a specific item category at test, i.e. recognition processes that are either specific to old items (model 1), shared by semantically related (model 2) or perceptually related (model 3) lures. Thus, our pattern of results might point to distinct patterns of changes in anterior vs. posterior hippocampus. Elucidating the distinct contributions of anterior vs. posterior regions of the hippocampus to recent and remote memories remains a challenge for future research. Furthermore, it has to be noted that memory performance was overall high in the present study, in particular for the 1d-group, which did not allow an analysis of neural activity associated with FAs to specific types of lures. To enable an analysis focussed on incorrectly endorsed related material, future studies should thus consider increasing task difficulty, for instance by increasing the number of encoded items or extending the retention interval.

Notably, the time-dependent transformation of memories into semantically generalized representations was significantly impacted by the emotionality of the encoded material. Although emotionally negative items were more robust against forgetting over time compared to neutral memories—corroborating the well-known memory enhancement for emotionally arousing information[53,54]—there was also an increased FA rate to emotionally negative, semantically related lures, suggesting an increased semantic transformation over time for negative material (for perceptually related lures we did not find credible evidence for a similar effect). This pattern of results is generally well in line with previous research indicating that the memory-enhancing effect of emotional arousal is specific to central aspects of a memory and comes at the cost of its peripheral, emotionally less salient features[23,55]. In other words, emotional arousal may prioritize the storage of the most salient aspects of an experience, which are then particularly well retained in the long run. This process might reflect a 'better-safe-than-sorry' mechanism that is highly adaptive for emotionally arousing, potentially threatening experiences. At first glance, this increased memory semantization for emotional relative to neutral items might seem in conflict with recent rodent and human data showing that noradrenergic arousal after encoding may reverse the systems consolidation process and hence result in more specific memories in the long run[2,56]. These studies, however, increased noradrenergic arousal pharmacologically after encoding, whereas the arousal boosts in the present study were rather transient and occurred during the encoding of individual stimuli. Thus, in the present study,

arousal did not selectively affect memory consolidation but primarily encoding processes, including the attentional focus when processing stimuli. On the neural level, this increased semantization for emotional events over time were associated with a specific decrease in distinct representations of encoded events in the anterior hippocampus.

To conclude, our findings show that the transformation of memory over time is semantic in nature and that this time-dependent memory semantization is enhanced for emotionally negative events. For a potential perceptual transformation over time, we did not find any credible evidence. In the brain, this semantic transformation was not only linked to the emergence of semantically transformed representations in neocortical areas over time but also to time-dependent changes within the hippocampus, with highly specific pattern representations for encoded events in the anterior hippocampus that decreased over time while posterior hippocampal reinstatements were linked to the extent to which remote memories were semantically transformed. Those findings provide insights into a key aspect of memory, its evolution over time, and how episodic experiences may be abstracted into semantic knowledge structures.

## Methods

### Behavioral pilot study

To validate the semantic and perceptual relatedness of the stimulus set, we conducted a behavioral pilot study in a sample of 33 undergraduate students (24 females, 9 males; age: $M = 22.48$ years, SEM = 0.60 years). All participants gave informed consent and received course credit for participation. One participant did not finish the task due to discomfort during viewing the emotionally negative stimuli, resulting in a final sample of 32 participants (23 females, 9 males; age: $M = 22.53$ years, SEM = 0.62 years).

In this pilot study, participants were presented with 280 pictures of scenes, taken from the International Affective Picture System[57] and open internet platforms. Half of the pictures contained emotionally negative scenes or objects while the other half contained neutral contents. The pictures were divided into 70 sets of four stimuli each: (1) the original picture (i.e. the old item in the main study), (2) one picture containing the semantic gist of this original picture, (3) one picture containing a different gist, while being perceptually related to the original picture; and (4) one unrelated picture, i.e. neither perceptually nor semantically related to the original item. The four pictures belonging to a set were matched to a respective old item in terms of subjectively perceived visual complexity, the depiction of people or animals by the first author and another independent rater. All unrelated (and perceptually related) images carried a different semantic gist than all other images, i.e. if one original image carried the semantic gist 'rental bikes' no other lure (or old item) besides the corresponding semantically related lure depicted rental bikes.

During the pilot study, each original picture was presented once next to its corresponding semantically related, perceptually related or unrelated counterpart using PsychoPy2 (v1.90.1)[58]. Participants rated the semantic and perceptual similarity of each picture pair via mouse-click on a 10-point Likert-Scale from 0 ('not related') to 10 ('very related'). Participants either rated first the semantic and subsequently the perceptual relatedness of a picture pair or vice versa, with the order of rating scales being counterbalanced across participants. Which side of the screen the comparison picture was presented on as well as the presentation order of image pairs, was randomized. Prior to the task, participants conducted two practice trials: one with a semantically related picture pair and one with a perceptually related pair. Participants were instructed to focus exclusively on visual features, e.g. shapes and colors of the pictures, when rating the perceptual relatedness of a picture pair. Accordingly, they were asked to consider only content-related aspects when rating the semantic relatedness of a picture pair and were further informed that it might help to think of a short title representing the gist of each picture. Participants were

instructed to look thoroughly at each picture before responding. The duration of each of the 210 trials was self-paced.

For the main study, we aimed at a final sample of 30 stimulus sets per emotionality category (neutral vs. negative). Based on the results of the pilot study, we therefore excluded 10 stimulus sets for which participants' ratings indicated that semantically and perceptually related pictures were not sufficiently distinct on the respective relatedness dimensions. In the resulting final stimulus sets, semantically related pictures were rated as being significantly more semantically related to the original picture (M = 9.38, SEM = 0.08) than both perceptually related (M = 2.38, SEM = 0.18; paired $t$-test $t(31) = 33.24$, $p = 8e–14$, $d = 5.64$, 95% Confidence Interval = [5.31, 5.97]) and unrelated pictures (M = 1.97, SEM = 0.22; paired $t$-test: $t(31) = −29.63$, $p = 8e–14$, $d = −4.84$, 95% Confidence Interval = [−5.16, −4.52]; main effect lure type for semantic relatedness: $F(1.59, 49.35) = 794.01$, $p = 3e–36$, $\eta_p^2 = 0.96$, 95% Confidence Interval: [0.95, 0.98]). Moreover, perceptually related items were rated as being significantly more perceptually related (M = 7.22, SEM = 0.14) to original pictures than both semantically related items (M = 5.66, SEM = 0.24; paired $t$-test: $t(31) = −6.32$, $p = 1e–06$, $d = −1.10$, 95% Confidence Interval = [−1.44, −0.76]) and unrelated items (M = 2.05, SEM = 0.20; paired $t$-test: $t(31) = −14.91$, $p = 3e–15$, $d = −2.57$, 95% Confidence Interval = [−2.91, −2.23]; main effect lure type for perceptual relatedness: $F(1.70, 52.68) = 284.85$, $p = 2e–27$, $\eta_p^2 = 0.90$, 95% Confidence Interval: [0.86, 0.94]). See Supplementary Fig. 1 for an overview of the relatedness ratings for the different stimulus categories in this pilot study.

### Main study

**Participants and design.** Fifty-five healthy volunteers (28 males, 27 females, age: $M = 24.22$ years, SEM = 0.54 years) participated in this experiment. Exclusion criteria were checked in a standardized interview and comprised a history of any psychiatric or neurological diseases, medication intake or drug abuse, as well as any contraindications for MRI measurements. All participants provided informed consent before taking part in the experiment and received a monetary compensation for participation (70€ or 75€, depending on whether fMRI measurements were conducted within a 1d or a 28d time frame). This study is part of a larger project investigating modulators of time-dependent systems consolidation and memory-transformation processes. The study protocol was approved by the ethics committee of the Medical Chamber Hamburg (PV5480) and was in accordance with the declaration of Helsinki. Three participants had to be excluded from the analysis because of technical failure ($n = 1$ participant) or falling asleep during at least one of the MRI sessions ($n = 2$ participants), resulting in a final sample of 52 right-handed young adults (26 females, 26 males, age: $M = 24.29$ years, SEM = 0.55 years). Participants were pseudo-randomly assigned to the 1d or 28d group (13 females and 13 males per group). The investigators were not blinded to allocation during experiments and outcome assessment. The final sample size is in line with previous fMRI studies on time-dependent memory-transformation processes[4,19] and a sensitivity analysis using MorePower 6.0.4[59] confirmed that this sample size is sufficient to detect a medium-sized effect ($\eta_p^2 > 0.09$) for our primary behavioral effect of interest reflected in a 2 (delay) × 3 (lure type) × 2 (emotion) mixed ANOVA with a power of 0.80 (α = 0.05).

**Experimental procedure.** Testing took place on three days: Day 1–encoding, Day 2–recognition testing, and Day 3–relatedness rating. We collected MRI data during experimental Day 1 and Day 2. Critically, in order to assess time-dependent changes in memory, the delay between encoding and memory testing was either 1d or 28d. All testing took place in the afternoon (between 1 and 6 pm).

*Memory encoding (Day 1)*: After providing informed consent, participants completed the Trier Inventory for the Assessment of Chronic Stress (TICS)[60], the Beck Depression Inventory (BDI-II)[61] and

the State-Trait Anxiety Inventory (STAI)[62]. At the beginning of the second experimental day (either 1d or 28d after Day 1), participants also filled out the Pittsburgh Sleep Quality Index (PSQI)[63] extended by questions regarding the duration and quality of sleep in the last 24 h. We obtained no statistically significant difference between groups in any of these parameters (all $p > 0.180$; see Supplementary Table 18). Afterwards, participants performed three encoding runs in the MRI scanner. In each run, participants encoded the same 60 pictures (30 emotionally neutral, 30 negative) presented in random order using MATLAB (The Mathworks, Inc, Natick, US) Version 2016b with the Psychophysics Toolbox 3 extensions[64], i.e. each picture was presented once in each of the three encoding sessions. On each trial, a picture was presented for 3 s followed by a jittered fixation period of $4 \pm 1$ s. Participants were instructed to memorize the presented pictures and informed that there will be a subsequent memory test immediately afterwards. To make sure that participants remained fully attentive throughout the encoding task, they were instructed to press a button as soon as the fixation-cross appeared on the screen. Immediately after the encoding task, participants completed a free recall task outside the MRI scanner. Here, participants had 15 min to recall as many stimuli in as much detail as possible, while an experimenter ticked off the correct stimuli from a list.

*Memory testing (Day 2)*: Depending on the experimental condition, participants returned to the lab either 1d or 28d after Day 1. On this second experimental day, participants performed a recognition task in the MRI scanner, which was separated into three consecutive runs. During the recognition test, participants saw the 60 pictures that were presented on Day 1 ('old') and 60 pictures that were new but semantically related to the old pictures, 60 pictures that were perceptually related to the old pictures and 60 pictures that were neither perceptually nor semantically related to the old pictures. Immediately after a picture was presented for 3 s, participants were requested to indicate via button press whether the shown picture had been presented on Day 1 or not, using a four-point scale ('definitely new', 'rather new', 'rather old', 'definitely old'). Between trials, a jittered fixation cross was presented for $4$ s $\pm 1$ s. Finally, the participants rated all pictures with respect to picture-valence and -arousal on a scale from 0 ('very negative'/'not arousing') to 10 ('very positive'/'very arousing'). In retrospect, these data confirmed that negative pictures ($M = 2.56$, SEM $= 0.09$) were perceived as significantly more negative than neutral ones ($M = 5.65$, SEM $= 0.14$; paired $t$-test: $t(51) = 14.94$, $p = 3e{-}20$, $d = -3.65$, 95% Confidence Interval $= [-4.28, -3.02]$). Furthermore, negative pictures ($M = 5.37$, SEM $= 0.17$) were associated with significantly higher subjective arousal than neutral ones ($M = 2.59$, SEM $= 0.21$; paired $t$-test: $t(51) = -15.55$, $p = 5e{-}21$, $d = 2.03$, 95% Confidence Interval $= [1.55, 2.50]$).

*Relatedness Rating (Day 3):* Participants returned to the lab for a last, behavioral task after at least three and a maximum of eight days after experimental Day 2 ($M = 4.17$d, SEM $= 0.18$d; without a statistically significant difference between groups regarding the delay between experimental Day 2 and Day 3; two-sample $t$-test: $t(43.13) = 0.99$, $p = 0.329$, $d = -0.28$, 95% Confidence Interval $= [-0.82, 0.27]$). In this final task, participants rated the semantic and perceptual relatedness of the 60 encoded pictures to each of its perceptually related, semantically related or unrelated lure on a scale reaching from 0 ('not related') to 10 ('very related'). This task was identical to the behavioral validation task (see the pilot study above), comprising the 240 pictures of the recognition task presented using MATLAB (The Mathworks, Inc, Natick, US) Version 2016b with the Psychophysics Toolbox 3 extensions[64].

**Behavioral data analysis.** To control for attentiveness during encoding on Day 1, the number of missed responses to the fixation cross was analyzed by means of a mixed ANOVA with the between-subjects factor delay (1d vs. 28d) and the within-subject factor run (run 1 vs. run 2

vs. run 3). To control for potential group differences in immediate memory, free recall performance right after encoding was analyzed by means of a mixed ANOVA with the between-subjects factor delay (1d vs. 28d) and the within-subject factor emotion (neutral vs. negative).

To assess the overall performance in the recognition test, we subjected the percentage of hits to a mixed ANOVA with delay (1d vs. 28d) as between-subjects factor and the within-subject factor emotion (neutral vs. negative). In order to assess the specificity of memory, the key question of this study, we further analyzed the percentages of FAs for each lure type by means of a mixed ANOVA with the between-subjects factor delay (1d vs. 28d) and the within-subject factors lure type (semantically related vs. perceptually related vs. unrelated) and emotion (neutral vs. negative). To further test for potential differences in the confidence in those FAs, we multiplied each FA by its level of confidence (1 = 'rather old', 2 = 'definitely old') before subjecting the FA rate to another mixed ANOVA with the factors delay, lure type and emotion. Moreover, we analyzed the confidence in FAs for each lure type by means of binomial generalized LMMs with delay (1d vs. 28d), emotion (neutral vs. negative) and their interaction as fixed effects and the random intercept of participants and stimuli.

We further assessed changes in memory quality for each encoded item by considering the response pattern over each related stimulus set, i.e. containing the original stimulus and its corresponding perceptually related and semantically related lure[19]. To this end, we assigned memories to one of four categories: (1) detailed memories, for which participants rated old pictures as 'old' and all other pictures of a set as 'new', (2) semantically transformed memories, for which participants endorsed the semantically related picture, but not the perceptually related picture, as 'old' (irrespective of the response to the old picture), (3) perceptually transformed memories, for which participants endorsed the perceptually related lure as 'old' while classifying the semantically related lure as 'new', and (4) forgotten sets, for which participants missed the old picture and correctly rejected both semantically and perceptually related pictures. The occurrence of each specificity category was analyzed by means of binomial generalized LMMs with a logit function, i.e. logistic mixed models, with delay (1d vs. 28d), emotion (neutral vs. negative) and their interaction as fixed effects and the random intercept of participants and stimulus sets.

The individual stimulus relatedness ratings on Day 3 further allowed us to analyze FAs by means of a binomial generalized LMM with a logit function and the factor delay (1d vs. 28d), emotion (neutral vs. negative), semantic relatedness, perceptual relatedness and their interactions as fixed effects and the random intercept of participants and stimuli. As our main effect of interest contained a cross-level interaction requiring unbiased estimates of the Level-1 association[65], our continuous level-1 predictors (semantic and perceptual relatedness ratings) were group mean-centered prior to fitting the generalized LMM. Note that results did not change when these predictors were grand mean-centered.

All statistical analyses were performed with R Version 4.0.2 (https://www.r-project.org/). All reported $p$-values are two-tailed. In case of violated sphericity, as indicated by Mauchly's test, results of ANOVA-models are reported with Greenhouse-Geisser corrected degrees of freedom and $p$-values. Results of all main analyses were tested on distortions due to extreme outliers, defined as data points with a standard deviation $\pm 3$ SD of the mean of the interesting condition. Note that if not stated otherwise, results did not change after excluding outliers. Post-hoc tests were conducted using $t$-tests, $z$-tests and interaction contrasts, i.e. contrasts between contrasts, by comparing estimated marginal means of each ANOVA-model or (generalized) LMM, with Šidák correction for multiple comparisons, using the R-package emmeans Version 1.7.2[66]. For ANOVAs and LMMs, Satterthwaite's approximation method was applied to calculate degrees of freedom for post-hoc $t$-tests. For all generalized LMMs and

corresponding post-hoc z-tests, p-values were computed using Wald z-distribution approximation, which does not rely on the specification of degrees of freedom. LMMs were fitted with Restricted Maximum Likelihood and the 'nloptwrap' optimizer. Generalized LMMs were fitted with Maximum Likelihood and the 'BOBYQA' optimizer. All (generalized) LMMs were estimated using the package lme4[67] Version 1.1. Results were visualized by utilizing bar plots and individual data points with the package ggplot2[68] Version 3.4.2 and plotting marginal effects of generalized LMMs with the package sjPlot[69] Version 2.8.12.

**MRI acquisition.** MRI data were acquired using a 3 T Prisma Scanner (Siemens, Germany) with a 64-channel head coil. Each MRI session consisted of three functional runs and a magnetic (B0) field map to unwarp the functional images (TR = 634 ms, $TE_1$ = 4.92 ms, $TE_2$ = 7.38 ms, 40 slices, voxel size = $2.9 \times 2.9 \times 3.0 mm^3$, FOV = 224 mm). For the functional scans, T2*-weighted echo planar imaging sequences were used to obtain 2 mm thick transversal slices (TR = 2000ms, TE = 30 ms, flip angle = 60°, FOV = 224). Additionally, a high-resolution T1 weighted anatomical image (TR = 2500 ms, TE = 2.12 ms, 256 slices, voxel size = $0.8 \times 0.8 \times 0.9 mm^3$) was collected at the end of the MRI session of Day 2.

**Preprocessing.** All scans underwent the same preprocessing steps using SPM12 (Wellcome Trust Centre for Neuroimaging, London, UK). To allow for magnetic field (T1) equilibration, the first three functional scans were discarded. The images were first realigned and unwarped using the field maps, then coregistered to the structural image followed by a normalization to Montreal Neurological Institute (MNI) space, as implemented in SPM12 (IXI549Space). No smoothing was performed on the echoplanar imaging data that entered the GLM.

**ROI definition.** Anatomical masks for the aCC, precuneus, angular gyrus (left and right), the occipital pole and Heschl's gyrus (left and right) were derived from the Harvard-Oxford atlas using a probability threshold of 50%. For the IFG and vmPFC, a sphere with 20 mm radius was used that was centered on the peak voxel ($x = -50$, $y = 16$, $z = 12$) derived from 386 imaging studies reporting 'IFG' and on the peak voxel ($x = -2$, $y = 46$, $z = -8$) derived from 199 imaging studies reporting 'vmPFC', respectively, as determined by meta-analyses conducted on the neurosynth.org platform (status 02/06/2022). As we expected a time-dependent representational change along the hippocampal long axis, we used anatomical masks of the anterior and posterior hippocampus (left and right), which were derived using the WFU pick-atlas[70,71].

**Quantification and statistical analysis.** For our MRI data analysis, each trial of the encoding and recognition task was modeled as an individual regressor convolved with a hemodynamic response function along with six session-constants in one GLM per subject using SPM12. To increase the reliability by normalizing for noise[72], the resulting beta-values were transformed into t-statistics. Data were further subjected to RSAs[29] using custom scripts in MATLAB Version 2020b (The Mathworks, Inc, Natick, USA). Note that for our neural analyses, activation patterns of all trials of relevant item types were included. We opted for an analysis at the category level instead of relying on participants' correct or incorrect responses because (i) we were interested in how the encoding-retrieval delay and lure type affected the similarity between representational patterns as an indicator of the specificity of the neural representational patterns rather than the underlying neural patterns of a specific behavioral response; (ii) (multivariate) neural data are much more sensitive to fine-grained changes in memory representations compared to behavioral data that is merely based on dichotomous 'yes' vs. 'no' (i.e. 'old' vs. 'new') responses; (iii) reducing analyses on incorrectly endorsed lures (FAs)

would have resulted in an insufficient number of trials for the fMRI analyses while (iv) focusing solely on correctly endorsed items (hits) would exclude items that are particularly low in memory specificity, which are of particular interest when investigating the neural under-pinnings in memory transformation over time.

**Model-based retrieval-similarity analysis.** We analyzed time-dependent changes of representational similarities between the different stimulus-types at recognition testing by applying a model-comparison RSA[29,30,73]. This approach, i.e. comparing multivariate representational patterns of all experimental trials (irrespective of the correctness of the response) to conceptual models, allows inferences about the structure of neural representations[29,30,73] and has been successfully employed in previous studies to characterize memory representations, even at longer delays after encoding[19,74–77] and is thus highly suitable for investigating changes in memory quality over time.

Here, separately for both emotionality categories, each trial's activation pattern across voxels was correlated (Pearson's r) with the activation patterns of each other trial during memory testing. Next, we computed the mean pattern similarity for comparisons within each of the three runs and for each between-run combination (run 1 and run 2, run 2 and run 3 or run 3 and run 1). Those run-related pattern similarities where then subtracted from each correlation estimate of the corresponding run-combination to account for inflated correlations as a function of temporal proximity between scans[78,79]. In the resulting 120 × 120 RSMs, each combination of trials was placed in the respective cells, ordered by stimulus type (Fig. 4a, left panel). The resulting neural RSMs were compared to three theoretical model RSMs (Fig. 4a, right panel), each predicting different similarity patterns between the four stimulus categories at recognition testing: similar representations for old pictures that are distinct from patterns for all novel stimuli (model 1: 'old items are distinct from all lures'), similar representations between old items and semantically related lures which are distinct from perceptually related and unrelated lures (model 2: 'old and semantically related items are similar') and a model that expects similar representations between old items and perceptually related lures which are distinct from semantically related and unrelated lures (model 3: 'old and perceptually related items are similar'). Note that for all models we expected old items to be represented more similarly, as they should equally initiate recognition processes in neural areas relevant for specific (model 1) or transformed (model 2 and model 3) memory representations. We computed Spearman's rank correlation coefficient for each single-subject RSM and the conceptual models as we did not assume a direct linear match between the compared RSMs[29]. The resulting rho-values were further Fisher z-transformed and subjected to mixed ANOVAs with the factors delay (1d vs. 28d), emotion (neutral vs. negative) and a-priori model (1: 'old items are distinct from all lures' vs. 2: 'old and semantically related items are similar' and' vs. 3: 'old and perceptually related items are similar') in R. As we expected a time-dependent differentiation along the anterior-posterior hippocampal long axis, we additionally included the factor long axis (anterior vs. posterior) in the analysis regarding the hippo-campus. For the neocortex, we predicted a comparable increase in semantically transformed memory representations (model 2) with increasing delay in each of our prefrontal (aCC, IFG, vmPFC) and parietal (precuneus, angular gyrus) long-term memory ROIs. We therefore first performed a mixed ANOVA with the between-subjects factor delay (1d vs. 28d) and the within-subject factors emotion (neutral vs. negative) and model RSM (model 1 vs. model 2 vs. model 3) using a combined mask that included all of these prefrontal and parietal ROIs. To confirm whether the resulting effect in model 2 was equally evident in the individual neocortical storage sites, we repeated this delay × emotion ANOVA with the neural RSM of each neocortical ROI. In case analyses were repeated for both hemispheres, resulting p-values were Bonferroni corrected ($p_{corr}$) to account for multiple comparisons.

**Memory reinstatement analysis.** Additionally, we assessed ERS as a measure of trial-specific memory reinstatement[20,32–37]. Due to the important role of the hippocampus in the reinstatement of episodic memories[38,39], we focused specifically on the hippocampus and the differentiation along its anterior-posterior axis in the analyses of ERS. We computed the similarity (Pearson's $r$) between activation patterns across all encoding runs as a reliable indicator of encoding-related activation patterns on experimental Day 1 and activation patterns of the same item during memory testing at Day 2 (see also[20]). Note that contrasting this ERS measure with ERS measures based on each individual encoding run, i.e. run 1, run 2, run 3, on a trial-by-trial level yielded a very similar pattern of results and no differences in anterior (all $p > 0.333$) nor posterior hippocampal ERS (all $p > 0.165$) between different ERS measures. Resulting correlation estimates were Fisher $z$-transformed before statistical analyses in R were conducted. First, time-dependent changes in item-specific hippocampal ERS were analyzed by means of trial-wise LMMs with the factors delay (1d vs. 28d), emotion (neutral vs. negative), long axis (anterior vs. posterior) and their interactions as fixed effects and the random intercept of participants and stimuli. As this analysis was repeated for both hemispheres, resulting $p$-values were Bonferroni corrected ($p_{corr}$) to account for multiple comparisons. Further, we followed up whether the observed delay-dependent increase in left posterior hippocampal ERS was associated with a decrease in specificity of the reinstated memories. To this end, we analyzed the occurrence of a FA for a semantically related or perceptually related lure by means of binomial generalized LMMs with emotion (neutral vs. negative), delay (1d vs. 28d), ERS and their interaction as fixed effects and the random intercept of participants and stimuli.

While ERS is computed by correlating pattern representations of individual items during encoding and memory test, i.e. 'old' items, we furthermore assessed the similarity elicited by perceptually or semantically related items at memory test and corresponding old items during encoding as a possible indicator for a reinstatement of the perceptual or semantic gist of the original memory. The resulting Fisher transformed $r$-values were again subjected to LMMs with delay (1d vs. 2d), emotion (neutral vs. negative), long axis (anterior vs. posterior) and their interaction as fixed effects and the random effects of subjects and stimuli. Furthermore, we explored delay-dependent changes in memory reinstatement, i.e. ERS, and reinstatement by related material in our neocortical long-term memory as well as sensory control ROIs by means of LMMs with the fixed effects of delay (1d vs 28d), emotion (neutral vs. negative) and their interactions.

### Reporting summary

Further information on research design is available in the Nature Portfolio Reporting Summary linked to this article.

## Data availability

The behavioral and fMRI data generated in this study have been deposited in the Open Science Framework (OSF) (https://doi.org/10.17605/OSF.IO/W5MXR[80]). Raw and processed fMRI data are available at OSF. Raw behavioral data is available at OSF. The data that can be used to reproduce the figures and tables are provided in the Source Data file and at OSF. ROI masks used for fMRI analyses were derived from the Harvard-Oxford atlas as included in the FMRIB Software Library, (https://fsl.fmrib.ox.ac.uk/fsl/fslwiki/FSL) from the WFU pick-atlas[70,71] and from the neurosynth.org database. All ROIs adapted for this study are available at OSF. Source data are provided with this paper.

## Code availability

Custom code used to model and analyze the data is available at Zenodo: https://doi.org/10.5281/zenodo.8363230[81] and integrated in the study's repository at OSF[80].

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

## Acknowledgements

This study was funded by a grant from the German Research Foundation (DFG) to L.S. and B.R. (SCHW1357/19) and by a grant from the European Union to A.A. (ERC, 101044616). Views and opinions expressed are however those of the authors only and do not necessarily reflect those of the European Union or the European Research Council Executive Agency. Neither the European Union nor the granting authority can be held responsible for them. We further acknowledge financial support from the Open Access Publication Fund of Universität Hamburg. Finally, we gratefully acknowledge the assistance of Carlo Hiller with the programming of the task and of Fabian Schacht, Vincent Kühn, Roberta Souza-Lima, Miguel Bermudez Alcaide, Anne Tiefert and Anja Turlach during data collection.

## Author contributions

L.S. and B.R. designed research; V.K. performed research; V.K., A.A. and T.S. analyzed data; V.K. and L.S. drafted and revised the manuscript; B.R., TS. and A.A. revised the manuscript.

## Funding

## Competing interests

The authors declare no competing interests.
