## [Peer Review File · Nature Communications]

Unraveling the semantic nature of memory transformation over timeReviewers' comments:

Reviewer #1 (Remarks to the Author):

The present study explores how episodic memories are modified over time. In particular, the authors investigated how low- and higher-level features could be independently transformed over time, depending on their emotional value (neutral or negative) and how different brain areas (anterior/posterior hippocampus and neocortical structures) could play a role in such memory representation. For this, they used behavioural measurements and fMRI data combined with RSA. The questions investigated here are undoubtedly of great interest for the long-term memory community, since they are not only trying to test how memories become more gist-like or perceptual independent over time (see Lifanov et al., 2021), but illustrate how memory representations are transformed semantically but not perceptually. Understanding this memory reorganization is fundamental in order to have a clear picture of episodic memory processes. According to the authors, they found that episodic memories are semantically but not perceptually transformed and this semantic transformation increases in the neocortex over time. However, perceptual details in the hippocampus decline over time.

Besides the importance of the research questions, the elegant paradigm implemented and how well the paper is written, I found especially difficult to interpret some of the results given the amount of factors and levels. In particular, interactions with 3 or more factors. However, I believe that the authors decompose particularly well such complex interactions adding the right analyses to interpret the results. In general, I am very positive about the impact, the quality of analyses and some of the interpretations of this research. However, I do not recommend the manuscript for publication before addressing the next points; one of the points being of particular relevance to interpret the results.

1. I found that some of the claims regarding the role of the posterior hippocampus reflecting gist-like memory representations are not fully backed-up by the results.

First, the authors suggest that the posterior hippocampus may reinstate rather unspecific and semantically transformed episodic memories based on their ERS results. However, this claim is mainly based on a (very) weak interaction between ERS and the delay factor on the probability of false alarms (where they did not find a main effect of ERS).

Second, I find particularly challenging the lack of any model fit (regarding the model-based RSA) in the posterior hippocampus. More specifically, I find the absence of significant fits regarding the second model ('old and semantically related items are similar') problematic. In addition, I found surprising that, given some of the publications cited (see line 291-292) or their interpretation of the ERS results, the authors did not hypothesize that the posterior part is relevant for the gist of episodic memories but only focus on the anterior hippocampus being key to process specific information of recent memories (line 349).

Altogether, I find claims like "The involvement of the posterior hippocampus in memory reinstatement increased over time and was associated with less specific memory representations, reflecting the semantic, but not perceptual, gist of the original memory" (line 111-113) not fully supported by the

results.

The authors should address the limitations of the results regarding the role of the posterior hippocampus in the semantic transformation of episodic memories. This would change some of the main messages of the manuscript.

2. Given how important is the transparency of code and data for reproducibility, I would prefer to confirm that the authors properly share this information as they stand in the manuscript. At this moment, the Github repository indicated in the manuscript does not contain data or code. As a personal recommendation, I would suggest using an alternative platform for the data (especially for large files; see for example Zenodo or OSF).

Some minors points:

- In the caption of some Figures “*” and “***” seems to indicate the same p-value (see line 214 and 307-308).

- I would add a label of the color bar in Fig. 4a.

Reviewer #2 (Remarks to the Author):

In their manuscript entitled “Unraveling the semantic nature of memory transformation over time”, the authors examine the retrieval of memory representations either at one day or 28 days after encoding using fMRI. During the Day 2 memory test, Ps were presented with target images, but also lure images that were either perceptually or semantically related to each target image. They did so to be able to behaviorally infer whether these memories were being transformed into more semantic or perceptual engrams. Not surprisingly, memory forgetting is accompanied by an increase in participants embracing semantic, but not perceptual, lures. The authors also include both negative and neutral images, the motivation for which was not entirely clear. The design of the study is commendable with the addition of the different lures. The authors find greater forgetting for neutral as compared to negative images over time. They do a nice analysis of false alarm rates to the different kinds of lures (unrelated, semantically related and perceptually related). They find that there are significantly more semantic false alarms at 28 days compare to 1 day and this seems more elevated for emotional images. They then use the results from this behavioral test to examine the imaging data to better understand where these transformations may be taking place. They see an increase in encoding-retrieval similarity at the 28 day period in the posterior hippocampus and interpret this as a greater ‘reliance on’ posterior hippocampal processes over time. They also report in Fig 3 a relationship between posterior hippocampal pattern similarity and false alarms for the semantic lures at 28 days and not for the perceptual lures. They then perform further analyses using a model based approach to determine more about the memory specificity.

I may be missing it here but the authors should report the hits and false alarm rates for the memory tests. They report hit rate (and we can infer misses from that) but we do not get a sense of the total # of false alarms that are made to the different kinds of lures. They do show the % of false alarms and how they differ between types of lures and how they increase over time but we have no sense of the total # of lures that are embraced as old. This is very important for all of the imaging analyses which examine activation patterns during these lure trials.

A major concern of mine is that the main imaging data analyses rely on very few trials. Even at 28 days when there are more false alarms, the total number of trials going into these analyses is very low. If there are 60 semantically related items (30 negative and 30 neutral), and 10% of those are false alarms in any category (the semantic one is the most relevant so I use that one as an example), that means an average of 3 or 4 trials are used per person for this computation. The other lures contain even fewer trials (1 or 2). I cannot endorse the inclusion of imaging data where only a handful (not even a handful!) of trials are used in the MAIN condition of interest.

At certain points, the inferences the authors make are not justified by the data. For example for the data show in Fig 3 which shows greater encoding-retrieval pattern similarity in posterior hippocampus at 28 days compared to 1, the authors write 'Memory reinstatement became reliant on' posterior hippocampus with time. There is not evidence for reliance here. These data are not causal. I suggest they simply report the results and say that pattern reinstatement is significantly higher at 28 days than it is at 1 day. Maybe they can see if pattern reinstatement in posterior hipp differs for correct and incorrect trials? Or to see if it is predictive of overall memory performance. That would bring them close to a potential mechanism, but still a correlational one.

It would be informative to know how reinstatement differs for targets and false alarms.

Other minor points

1. Because there were three encoding trials during learning, it is important to consider that memory representations may change with repetition. Since the main data in the imaging paper involve an analysis of encoding and retrieval pattern overlap, it would be important to know how the reported effects change when using the different repetition trials.
2. It is curious that the posterior hippocampal patterns did not fit better with a semantic transformation model in their analyses. They should then not be suggesting that semantic representations are developed in this region, leading to the increased semantic false alarm rate.

Reviewer #3 (Remarks to the Author):

The authors of the manuscript “Unravelling the semantic nature of memory transformation over time” investigate how the representations of human declarative memories change over time. This topic is very timely and of highest interest. The study uses a creative and well executed experimental design as well as a clever methodological approach of multivariate imaging to investigate the change of neural representations of memories over time. While it is clear that not all specific details of memories can be preserved indefinitely, it is still under debate in which way memories are transformed. The authors try to answer the question whether memories are perceptually or semantically transformed meaning that either the perceptual essence is preserved over time or the semantic gist of the memory. Overall, this is a very nice study with interesting results. For publication in Nature Communications, I would still be missing some additional analyses, and some statements should be softened.

Major concerns:

1) From figure 2C, supplementary table 1 and the pilot data, it becomes clear that semantically related (SR) pictures are not much less perceptually related than perceptually related (PR) items (both have a perceptual relatedness rating of ~6 while unrelated items have a rating of ~2). This perceptual relatedness in the semantic lures should be clearly mentioned, and the corresponding significance tests should be added.

Because SR items show high perceptual similarity to old items, only comparing SR items with PR items allows to draw conclusions about the effects of presence or absence of semantic relationships, and for any conclusions about the effects of presence or absence of a perceptual relationship, a comparison of PR items with unrelated items is necessary. This does not affect the majority of analyses, but the contrasts comparing SR-PR or PR-UR and results of the unrelated items should be included where relevant, and a discussion should be added.

A supplementary analysis within the SR lures comparing FAs with high and low perceptual relatedness ratings might further validate the argumentation.

2) It is highly interesting that perceptually related items do not seem to produce more false alarms than unrelated items. It is mainly semantically related false alarms that are produced. This speaks in favor of the idea that the memories are mostly stored in semantic (‘abstract’) form rather than in perceptual (‘photographic’) form. The lack of PR FAs should be discussed.

3) The authors interpret the increase in semantic FAs as a sign of a semantic transformation. It can, however, also mean that previously available perceptual detail, which prevented the false alarm at 1d, has been lost over time, while coarse semantic information was still available at 28d. Instead of a semantic transformation, the results would then rather describe forgetting of identifiable detail. That detail could pertain to the perceptual (photographic representation of blue bikes) domain or to the semantic domain (abstract representation of “blue bikes”).

In fact, the much simpler explanation is that both semantic and detailed perceptual information is encoded from the start and lost over time than that the brain is occupied during the 28 days after encoding random images with their re-coding and semanticization. In that view, the data would not be in line with MTT/TTT and repeated reactivation and re-encoding (line 473).

4) Similarly, the increased false alarm probability and higher ERS for SR items at 28d (figure 3 lower right) can be seen as simply a consequence of the better retention of semantic information compared to

perceptual information. Higher ERS would result from a better retention of at least some semantic memory, which is indicated by higher probability of FA.

A similar figure to figure 3 lower right with probability of correct recall as a function of ERS (perhaps separately for neutral and negative items) could bring clarity to this issue. If stronger ERS correlates with higher correct recall as well as more semantic false alarms, then both indicate better memory retention. Without showing that higher ERS explicitly does not correlate with correct ('detailed') recall, the conclusion that "representations are unspecific in nature, reflecting semantically transformed memories" (line 333) is not warranted.

5) I do not concur with the interpretation of the ERS analysis that there is a "change" or "reorganization" along the hippocampal long axis. While they find an increased reinstatement after 28d in the posterior hippocampus, the authors do not observe the (expected) higher reinstatement of recent memories in the anterior hippocampus. While it is an interesting finding to see a change in ERS in the posterior hippocampus from 1d to 28d, it does not permit any conclusions about representational changes along the anterior-posterior axis of the hippocampus. The authors could have used any other region that showed no change in ERS for the same argument. Conclusions of a change or reorganization along the hippocampal axis have to be removed.

6) The analysis of retrieval similarity between stimulus types is not entirely convincing for several reasons:

- First and foremost, the four predicted similarity matrices are not plausible. RSM model 1 assumes high correlations between response patterns of all old items. It thus seems to assume that all old items have been learned in the same session and should therefore have some similarity (the context during encoding). It therefore does not reflect item specificity (contrary to line 350), but rather the generalizable aspects of the memories. Model 2 assumes high similarities also between activity evoked by stimuli that are semantically related to the old stimuli. However, there is no similarity that I could think of between SR items, except their link to the old items which in turn have a common context. However, the model assumes the association between SR and old items to be weaker than the association between unrelated SR items. Because their only link is the association with items that were learned in the same context, SR items should be assumed to have a weaker relation between each other than with the associated old item. Model 3 is probably incorrectly depicted in figure 4. PR items should show similarity to old items, not to SR items. Again, as for SR items, I do not see why one could assume a high similarity between PR items, which are unrelated to each other.

- The main effects in the anterior hippocampus stem from a negative fit of the model at the 28d interval. A negative fit would indicate that the model is not well suited to describe the actual activation patterns. Specifically, a negative model fit for Model 1 at 28d could mean that items lose their common, generalizable response pattern (context) and evoke more dissimilar responses, which explicitly suppress each other. Results can therefore not be interpreted as a detailed representation in the anterior hippocampus that is lost over time.

7) Some other additional analyses would be helpful:

- Analyses on PR and SR items could be done depending on whether an image evoked a false alarm or not.

- The ERS analysis might provide more information, if the ERS of later remembered vs forgotten (or detailed vs transformed vs forgotten) items would be compared separately instead of only looking at group differences.

- The ERS analysis might also be done on a whole brain level. Although the hippocampus is central for episodic memory reinstatement, with the special interest of the present study in semantic generalization, it suggests itself to include a wider range of regions in the analysis.

- Calculating Encoding-Retrieval-Similarity only for the original stimuli misses out on the important point whether PR or SR items also increase or decrease in similarity with the original items over time. In particular, a “semantic transformation” should increase similarity of activation patterns from 1d to 28d for SR with old items. If similarity does not increase, results would rather speak for forgetting of detail and retaining of coarse semantic information than for a semantic transformation.

- The RSA analyses show increasing model fit in all ROIs. Were there more ROIs tested or did all selected ROIs show the increase? Are there any cortical regions that do not show this pattern?

Minor concerns:

1) Unrelated items were unrelated to the three others in a set, but were they also unrelated to the old images of the other 59 sets? How were relations between image sets assessed?

2) It is highly interesting that the activity pattern in posterior hippocampus becomes more similar to encoding after 28d. This is difficult to explain with any current model of memory consolidation. Is it possible that the higher difficulty of retrieval at 28d had participants focus longer and more attentively on the images?

3) On page 9, the authors describe a mixed model analysis investigating the occurrence of categorized stimulus sets regarding their memory (detailed, transformed, forgotten sets). The authors should state how many sets are included in each category for each group. In general, false alarm rates are very low in the 1d-group and might affect the interpretability of this analysis.

4) In figure 3C right, the number of stimuli for the extreme ends is also unclear. Here, it would be interesting to see the individual data points that underlie these predicted probabilities. It is also not entirely clear how this figure was generated. A gLMM is mentioned, which would imply linear predictors, but the relationship between relatedness and FA rate is depicted as a smooth curve.

5) If I understand the analyses correctly, the unpaired t-tests and the main effect of delay (1d/28d) in line 166 should be redundant.

6) In line 261 (page 10) the authors should probably refer to figure 2C instead of 2B. They should refer to figure 2B when reporting the results of the mixed model in lines 244-256.

7) In line 214 and 307 and 395 it should say $*p < .050$.

8) It is unclear why the authors state in line 252 that better memory for emotional items comes “at the cost of reduced memory for semantic details.” There are more items correctly remembered and there is memory for the semantic gist for additional items, indicating at least some residual semantic memory for those items. Both speak for better semantic memory for emotional material.

9) The authors should also state whether the “main effect delay” for the aCC and IFG is significant (line 431-436) and report statistical parameters.

10) What is meant by “we subtracted run-related pattern variances” in line 780?

11) There are a number of references that cite pre-prints without this being clear in the text. I do not believe that the non-peer reviewed literature should be put on the same footing as the peer-reviewed literature.

12) Is the 1997 version of the IAPS the same as the 1988 version? [Lang, P., Ohman, A., & Vaitl, D. (1988). The international affective picture system Gainesville, FL: University of Florida, Centre for Research in Psychophysiology]

13) For the first supplementary result, please provide a figure (or at least means), not just significance tests.

14) Supplementary Table 1: There might be a copy error for perceptual relatedness of semantically related negative and unrelated negative values, as both are identical.

Responses to Reviewers

Reviewer #1

We thank the reviewer for the positive and helpful comments. We are glad that he or she considers our study to be of great interest for the long-term memory community, the implemented paradigm to be elegant, the qualitative and complex analyses to be well decomposed and the manuscript to be well written.

1. I found that some of the claims regarding the role of the posterior hippocampus reflecting gist-like memory representations are not fully backed-up by the results.

RESPONSE: First, we would like to emphasize that the focus of the present manuscript was on whether the memory transformation over time is perceptual or semantic in nature and which neural mechanisms are involved herein. Time-dependent changes along the hippocampal long axis are a very exciting and timely topic (e.g. Sekeres et al., 2018; Zeidman & Maguire, 2016), which is why we included these findings in our manuscript, yet these were not the focus of our study.

Our conclusion that the posterior hippocampus reflects more gist-like representations is based on our finding that there was a significant, time-dependent increase in posterior hippocampal ERS, which was directly correlated with the probability of a false alarm (FA) for corresponding semantically related (but not perceptually related) lures, our main behavioral readout of memory transformation. Furthermore, the interpretation of this specific ERS pattern as reflecting a more gist-like posterior hippocampal representation was at least partly motivated by initial previous evidence suggesting such time-dependent changes along the hippocampal long axis (Bonnici et al., 2012; Dandolo & Schwabe, 2018; Tomparý & Davachi, 2017).

However, we make the basis of our conclusion clearer now, emphasize which evidence speaks for and which against this interpretation, and soften our statements wherever required (see below). Moreover, we have restructured the results section to make the focus of the present manuscript clearer.

First, the authors suggest that the posterior hippocampus may reinstate rather unspecific and semantically transformed episodic memories based on their ERS results. However, this claim is mainly based on a (very) weak interaction between ERS and the delay factor on the probability of false alarms (where they did not find a main effect of ERS).

RESPONSE: Please note that our main analysis of time-dependent changes in ERS along the anterior-posterior hippocampal axis indicating that the posterior hippocampus specifically reinstates remote memories was very robust as it remained significant even after Bonferroni correction (corrected for both hemispheres). Additionally, we investigated specifically whether the observed increase in posterior hippocampal memory reinstatement may explain the semantization in memory over time. Thus, the interaction of ERS and delay was crucial in this post-hoc analysis and indeed significant. A main effect of ERS would have not explained the semantization of memory over time and was thus less relevant in this analysis. We agree, however, that we should be more careful with how to interpret this ERS effect in the posterior hippocampus in relation to memory transformation along the hippocampal long axis and now modified our wording related to this aspect throughout the manuscript (please see also below).

Second, I find particularly challenging the lack of any model fit (regarding the model-based RSA) in the posterior hippocampus. More specifically, I find the absence of significant fits regarding the second model ('old and semantically related items are similar') problematic.

RESPONSE: Please note that we did see significant increases in model fit to model 2 ('old and semantically related are similar') in neocortical areas. However, while we did see an increase in memory reinstatement (i.e. ERS) in the posterior hippocampus over time that was positively associated with semantically related FAs indicative of an unspecific memory reinstatement, we did not observe a significant model fit to the 'old and semantically related items are similar' model in the posterior hippocampus. These discrepant findings might be owing to the different methodological approach. Whereas the ERS measures a change in reinstatement of an individual memory at test, the model-based analysis is directed at representational changes for a specific item category at test, i.e. recognition processes that are either specific to old items (model 1), shared by semantically related (model 2) or perceptually related (model 1) lures. We make the absence of a significant model fit to model 2 in the posterior hippocampus and its implications for the idea of gist-like representations in the hippocampus more explicit now and discuss in this context also the methodological differences between the ERS and model-based analyses. Please see pages 23-24, lines 598-616:

'Although the specific increase in posterior hippocampal ERS over time and its direct association with our behavioral indicator of semantic transformation (i.e. FAs to semantically related lures) and the decrease in distinct memory representations in the anterior hippocampus over time supports the idea of a time-dependent transformation along the hippocampal anterior-posterior axis with detailed memory representations in the anterior hippocampus and remote, gist-like representations in the posterior hippocampus, it is important to note that we did not observe a decrease in the anterior hippocampal memory reinstatement over time. Moreover, posterior hippocampal memory representations did not increase in similarity to semantically related material as indicated by our model-RSA analyses. The absence of an anterior hippocampal decrease over time and a time-dependent increase of a posterior hippocampal fit to the 'old and semantically related items are similar' model might be taken as support against the suggested differential memory transformation over time in anterior and posterior hippocampal areas. It is to be noted, however, that these seemingly discrepant findings may be owing to the different methodological approaches. Whereas the ERS measures a change in reinstatement of an individual memory at test, the model-based analysis is directed at representational changes for a specific item category at test, i.e. recognition processes that are either specific to old items (model 1), shared by semantically related (model 2) or perceptually related (model 3) lures. Thus, our pattern of results might point to distinct patterns of changes in anterior vs. posterior hippocampus. Elucidating the distinct contributions of anterior vs. posterior regions of the hippocampus to recent and remote memories remains a challenge for future research.'

In addition, I found surprising that, given some of the publications cited (see line 291-292) or their interpretation of the ERS results, the authors did not hypothesize that the posterior part is relevant for the gist of episodic memories but only focus on the anterior hippocampus being key to process specific information of recent memories (line 349).

RESPONSE: We completely agree and this hypothesis was implied in our initial version as we expected memories to be transformed 'from anterior to posterior parts' (see page 3, line 63). We have now made the hypothesis regarding the posterior hippocampus more explicit, please see page 12, lines 329-335:

'Based on recent evidence¹⁹, we hypothesized that the anterior hippocampus is particularly relevant for the specificity of recent memories while the posterior hippocampus represents remote, semantically transformed memories. Accordingly, the anterior hippocampus should reflect distinct representations (model 1) at a short delay, but this representation should decrease over time, while we expected the posterior hippocampus to represent semantically transformed memory that should increase over time (model 2).'

In addition, please note that we did include the posterior hippocampal pattern similarity in our model-based RSA-analysis.

Altogether, I find claims like “The involvement of the posterior hippocampus in memory reinstatement increased over time and was associated with less specific memory representations, reflecting the semantic, but not perceptual, gist of the original memory” (line 111-113) not fully supported by the results.

RESPONSE: We agree that this statement has been a bit too far-reaching and rephrased it accordingly, please see pages 4-5, lines 112-114:

‘Posterior hippocampal memory reinstatement increased over time and was associated with less specific memory representations that were linked to the semantic, but not perceptual, gist of the original memory.’

Moreover, we have revised the manuscript to soften our statements regarding the hippocampal long axis, wherever required (see comment above), and based these statements directly on the observed data. For instance, we changed our subtitle ‘Memory reinstatement becomes reliant on the posterior hippocampus over time’, please see page 17, line 421:

‘Posterior hippocampal memory reinstatement increases over time’

We also changed the statement ‘Memory reinstatement became reliant on the posterior hippocampus with increasing delay after encoding’ in the legend of Figure 4, please see page 18, line 436:

‘Posterior hippocampal memory reinstatement increased over time’

Moreover, we changed the statement ‘indicating that the posterior hippocampus is specifically associated with the reinstatement of the semantic, but not net perceptual gist of a remote memory’, please see page 18, lines 441-443:

‘indicating that posterior hippocampal reinstatement of remote memories is associated with the semantic, but not perceptual gist of the original memory’

We changed the statement ‘Thus, these findings indicate that, over time, memory reinstatement becomes increasingly reliant on the posterior hippocampus, and that those reinstated, remote memory representations are unspecific in nature reflecting semantically, but not a perceptually transformed memories.’, please see page 20, lines 506-508:

‘Taken together, our memory reinstatement analyses indicated that the posterior hippocampus is associated with the reinstatement of remote memories that may be rather unspecific in nature representing the semantic, but not perceptual, gist of the original memory.’

We also we changed the statement ‘memory reinstatement became increasingly reliant on the posterior hippocampus over time, which represented the semantic, but not perceptual, gist of the memory’, please see page 21, lines 523-525:

‘the posterior hippocampus was associated with the reinstatement of remote memories that were rather unspecific in nature and likely to be confused with the semantic, but not perceptual gist of the original memory’

The authors should address the limitations of the results regarding the role of the posterior hippocampus in the semantic transformation of episodic memories. This would change some of the main messages of the manuscript.

RESPONSE: We completely agree and explicitly mention now in the discussion that we did not see a delay-dependent increase in similarity of posterior hippocampal patterns to the model ‘old and semantically related items are similar’ (model 2; pages 23-24, lines 598-617, see comment above).

Please note, however, that – as indicated above – the main focus of the present study was not the differentiation along the hippocampal longitudinal axis but whether qualitative changes over time are characterized by a semantic or perceptual transformation and the neural underpinnings of this process.

2. Given how important is the transparency of code and data for reproducibility, I would prefer to confirm that the authors properly share this information as they stand in the manuscript. At this moment, the Github repository indicated in the manuscript does not contain data or code. As a personal recommendation, I would suggest using an alternative platform for the data (especially for large files; see for example Zenodo or OSF).

RESPONSE: We apologize for the delay in uploading our code and data. We are strongly committed to principles of open data and materials. The code to analyze the behavioral and neuronal data is now publicly available on the github-repository indicated in the manuscript on page 34, lines 934-936. As suggested by the reviewer, we now uploaded our data on OSF, please see page 34, lines 931-932:

'The behavioral and fMRI data generated in this study are provided at OSF:
https://osf.io/w5mxr/?view_only=34bbe9cd98fd4701b744b069477d5ad7'

Some minors points:

- In the caption of some Figures “**” and “***” seems to indicate the same p-value (see line 214 and 307-308).

RESPONSE: We thank the reviewer for noticing this mistake, which has now been corrected, please see page 18, line 446.

- I would add a label of the color bar in Fig. 4a.

RESPONSE: We have now added a label to the color bar Figure 3A (previously Figure 4A), as suggested.

Reviewer #2

We thank the reviewer for the positive and very constructive comments. We are happy that he or she commends our study design and our analysis of false alarm rates to the different kind of lures (unrelated, semantically related and perceptually related).

I may be missing it here but the authors should report the hits and false alarm rates for the memory tests. They report hit rate (and we can infer misses from that) but we do not get a sense of the total # of false alarms that are made to the different kinds of lures. They do show the % of false alarms and how they differ between types of lures and how they increase over time but we have no sense of the total # of lures that are embraced as old. This is very important for all of the imaging analyses which examine activation patterns during these lure trials.

RESPONSE: We agree that an understanding of the number of hits and false alarms (FAs) is relevant. We report the percentage of hits and FAs for each lure type in Figure 2A and these were calculated relative to the total number of old items and different lures, respectively (i.e. 30 old images, and 30 stimuli for each of the three lure types per level of emotion). We have made this clearer now in the Figure legend on page 8, lines 207-208:

'Individual data points indicate the mean percentage of the 30 items per emotion and item type correctly (left) or incorrectly (right) endorsed as 'old'.'

Additionally, we now list the mean raw data for FAs (per lure type), correct rejections (per lure type), hits and misses for the recognition task on Day 2 in Supplementary Table 2.

Please note, however, that the imaging data were based on all lure trials, not only on those that were incorrectly endorsed as 'old' (i.e. FAs; please see also our response to the next comment of this reviewer).

A major concern of mine is that the main imaging data analyses rely on very few trials. Even at 28

days when there are more false alarms, the total number of trials going into these analyses is very low. If there are 60 semantically related items (30 negative and 30 neutral), and 10% of those are false alarms in any category (the semantic one is the most relevant so I use that one as an example), that means an average of 3 or 4 trials are used per person for this computation. The other lures contain even fewer trials (1 or 2). I cannot endorse the inclusion of imaging data where only a handful (not even a handful!) of trials are used in the MAIN condition of interest.

RESPONSE: This is a misunderstanding and we apologize for not having been clearer on this in our original manuscript. Both, the Encoding-Retrieval-Similarity (ERS) and the model-based Representational Similarity Analyses (RSAs) are highly powered as they are based on pattern responses to 120 and 240 trials, respectively. Thus, the neural analyses included all trials per item type (old, semantically related, perceptually related, and unrelated; $n = 60$ per item type). We decided for an analysis at the category level and against an analysis of items depending on participants correct or incorrect responses because (i) we were interested in how the encoding-retrieval delay and lure type affected the similarity between representational patterns as an indicator of the specificity of the neural representational patterns rather than the underlying neural patterns of semantical FAs, (ii) (multivariate) neural data are much more sensitive to fine-grained changes in memory representations compared to behavioral data that is merely based on dichotomous ‘yes’ vs. ‘no’ (i.e. ‘old’ vs. ‘new’) responses, and (iii) We link these neural data to the behavioral level through correlational analyses and by using theoretical representational models.

In order to avoid any misunderstanding, we clarified now for all neuroimaging analyses that these were based on all trials (and we specify the respective number of trials included) and added also our rationale for this approach. Please see on page 31-32, lines 853-861:

‘Note that for our neural analyses, activation patterns of all trials of relevant item types were included. We decided for an analysis at the category level and against an analysis of items depending on participants correct or incorrect responses because (i) we were interested in how the encoding-retrieval delay and lure type affected the similarity between representational patterns as an indicator of the specificity of the neural representational patterns rather than the underlying neural patterns of a specific behavioral response (ii) (multivariate) neural data are much more sensitive to fine-grained changes in memory representations compared to behavioral data that is merely based on dichotomous ‘yes’ vs. ‘no’ (i.e. ‘old’ vs. ‘new’) responses and (iii) reducing analyses on incorrectly endorsed lures would have resulted in an insufficient number of trials for the fMRI analyses.’

At certain points, the inferences the authors make are not justified by the data. For example for the data show in Fig 3 which shows greater encoding-retrieval pattern similarity in posterior hippocampus at 28 days compared to 1, the authors write ‘Memory reinstatement became reliant on’ posterior hippocampus with time. There is not evidence for reliance here. These data are not causal. I suggest they simply report the results and say that pattern reinstatement is significantly higher at 28 days than it is at 1 day. Maybe they can see if pattern reinstatement in posterior hipp differs for correct and incorrect trials? Or to see if it is predictive of overall memory performance. That would bring them close to a potential mechanism, but still a correlational one.

RESPONSE: We agree and are now more careful with our wording to avoid the impression of causal inferences which are obviously not warranted for fMRI data in general. We now changed the statement ‘Memory reinstatement became reliant on the posterior hippocampus with increasing delay after encoding’ in the legend of Figure 4, please see page 18, line 436:

‘Posterior hippocampal memory reinstatement increased over time’

We also changed our subtitle ‘Memory reinstatement becomes reliant on the posterior hippocampus over time’, please see page 17, line 421:

‘Posterior hippocampal memory reinstatement increases over time’

We changed the statement ‘Thus, these findings indicate that, over time, memory reinstatement becomes increasingly reliant on the posterior hippocampus, and that those reinstated, remote memory representations are unspecific in nature reflecting semantically, but not a perceptually transformed memories.’, please see page 20, lines 506-508:

‘Taken together, our memory reinstatement analyses indicated that the posterior hippocampus is associated with the reinstatement of remote memories that may be rather unspecific in nature representing the semantic, but not perceptual, gist of the original memory.’

Furthermore, we changed the statement ‘memory reinstatement became increasingly reliant on the posterior hippocampus over time, which represented the semantic, but not perceptual, gist of the memory’, please see page 21, lines 523-525:

‘the posterior hippocampus was associated with the reinstatement of remote memories that were rather unspecific in nature and likely to be confused with the semantic, but not perceptual gist of the original memory’

As suggested by the reviewer, we further analyzed whether posterior hippocampal pattern reinstatement is linked to the correctness of participants’ responses and report those results on page 19, lines 461-465:

‘Analyzing the probability of a hit by means of a generalized LMM with ERS, emotion (neutral vs. negative), delay (1d vs. 28d) and their interaction as fixed effects and the random intercepts of participants and stimulus sets indicated that with increasing delay, posterior hippocampal memory reinstatement was significantly positively associated with the correct endorsement of an old item as ‘old’ (delay \times ERS: $\beta = 7.31$, $z = 2.17$, $p = 0.030$).’

Moreover, in response to another reviewer comment, we subsequently investigated the specificity of this memory reinstatement, please see page 19, lines 465-472:

‘However, correct memory could be supported by specific, detailed memory representations but also by more abstract, gist-like representations. We therefore further analyzed the specificity of the reinstated memories by taking into account the responses for corresponding related lures. Analyzing the probability of a detailed recognition (correct response for old items without FAs for related lures) by means of a binomial generalized LMM with the factors ERS, delay and emotion did not show an association of ERS with detailed memory (all $p > 0.688$; see Supplementary Figure 4), indicating that posterior hippocampal memory reinstatement after 28d is not associated with more detailed memory.’

It would be informative to know how reinstatement differs for targets and false alarms.

RESPONSE: We investigated the change in memory reinstatement over time by computing the similarity between hippocampal encoding pattern activations at test, i.e. ERS, which per definition is computed for targets only (see e.g. Ritchey et al., 2013; Staresina et al., 2012; Wing et al., 2015; Xiao et al., 2017). Nevertheless, we agree that it might be also interesting to see to what extent the representations of semantically or perceptually related lures resemble the representation of the corresponding old items during encoding. Therefore, we now also analyzed the similarity between encoding-related activation patterns and corresponding related images by means of LMMs with the factors emotion, long axis and delay. We now report these results on page 20, lines 489-498:

‘While ERS is computed by correlating pattern representations of individual items during encoding and memory test, i.e. old items, we furthermore assessed the similarity elicited by perceptually or semantically related items at memory test and corresponding old items during encoding as a possible indicator for a reinstatement of the perceptual or semantic gist of the original memory. This analysis indicated that the posterior hippocampus tended to show a higher reinstatement of the semantic gist of the original memory compared to the anterior hippocampus (main effect long axis: $\beta = 0.004$, $t_{6124} = 1.94$, $p = 0.052$; see Supplementary Figure 5). No effect approached significance when analyzing hippocampal reinstatement of

the perceptual gist (all $p > 0.235$), which further suggests that the posterior hippocampus might be specifically associated with the reinstatement of the semantic, but not perceptual, gist of the original memory.'

We discuss these interesting findings now on page 22, lines 573-576:

'Interestingly, our results suggest that posterior hippocampal encoding patterns are not only reinstated by original items but tend to be also reinstated by semantically, but not perceptually, related lures, additionally linking the posterior hippocampus to a rather unspecific memory reinstatement.'

Other minor points

1. *Because there were three encoding trials during learning, it is important to consider that memory representations may change with repetition. Since the main data in the imaging paper involve an analysis of encoding and retrieval pattern overlap, it would be important to know how the reported effects change when using the different repetition trials.*

RESPONSE: Similar to previous research (e.g. Tomparý & Davachi, 2017) we included the encoding-related activity of individual items across all encoding runs at experimental Day 1 for our ERS analysis. This approach helps to cancel out noise of individual runs and thus allows computing the most reliable indicator of encoding-related activity. We now elaborate on our rationale for using the average across the three encoding runs on Day 1 on page 33, lines 902-905:

'We computed the similarity (Pearson's r) between activation patterns across all encoding runs as a reliable indicator of encoding-related activation patterns on experimental Day 1 and activation patterns of the same item during memory testing at Day 2 (see also²⁰).'

Nevertheless, we now additionally computed ERS based on each individual encoding run, i.e. run 1 (ERS_{run1}), run 2 (ERS_{run2}) and run 3 (ERS_{run3}). Contrasting the reported ERS using the average over all encoding runs with each of those new ERS measures on a trial-by-trial level did not indicate significant differences in anterior (ERS_{run1} : $\beta = 0.0003$, $t_{24846} = 0.25$, $p = 0.803$; ERS_{run2} : $\beta = -0.001$, $t_{24846} = -0.97$, $p = 0.334$; ERS_{run3} : $\beta = 0.0008$, $t_{24846} = 0.73$, $p = 0.464$) nor posterior (ERS_{run1} : $\beta = 0.001$, $t_{24846} = 1.05$, $p = 0.293$; ERS_{run2} : $\beta = 0.0005$, $t_{24846} = 0.40$, $p = 0.686$; ERS_{run3} : $\beta = -0.002$, $t_{24846} = -1.39$, $p = 0.165$) hippocampal ERS and the overall pattern of results was very similar to the pattern reported when using the (more reliable) average across all encoding runs. We now report this on page 33 lines 905-907:

'Note that contrasting this ERS measure with ERS based on each individual encoding run, i.e. run 1, run 2, run 3, on a trial-by-trial level yielded a very similar pattern of results, without differences in anterior (all $p > 0.333$) nor posterior hippocampal ERS (all $p > 0.165$).'

Moreover, re-analyzing the delay-dependent change in hippocampal ERS using those new ERS-measures, our results remained largely unchanged, specifically when using ERS_{run1} and ERS_{run2} . For ERS_{run3} this effect was weaker, which is not surprising since the previous two encoding repetitions may have led to a decrease in actual encoding-related activity.

2. *It is curious that the posterior hippocampal patterns did not fit better with a semantic transformation model in their analyses. They should then not be suggesting that semantic representations are developed in this region, leading to the increased semantic false alarm rate.*

RESPONSE: Our conclusion that the posterior hippocampus reflects more gist-like representations is based on our finding that there was a significant, time-dependent increase in posterior hippocampal ERS, which was directly correlated with the number of FAs to semantically related (but not perceptually related) lures, our main behavioral readout of memory transformation. Furthermore, the interpretation of this specific ERS pattern as a more gist-like posterior hippocampal representation was at least partly motivated by initial previous evidence suggesting such time-dependent changes along the hippocampal long axis (Bonnici et al., 2012; Dandolo & Schwabe, 2018; Tomparý & Davachi, 2017). We agree, however, that we should explicitly discuss that we do not see a delay-dependent

increase in the fit of posterior hippocampal patterns to the model 'old and semantically related items are similar' (model 2). We explicitly address this now on page 23-24, lines 598-616:

'Although the specific increase in posterior hippocampal ERS over time and its direct association with our behavioral indicator of semantic transformation (i.e. FAs to semantically related lures) and the decrease in distinct memory representations in the anterior hippocampus over time supports the idea of a time-dependent transformation along the hippocampal anterior-posterior axis with detailed memory representations in the anterior hippocampus and remote, gist-like representations in the posterior hippocampus, it is important to note that we did not observe a decrease in the anterior hippocampal memory reinstatement over time. Moreover, posterior hippocampal memory representations did not increase in similarity to semantically related material as indicated by our model-RSA analyses. The absence of an anterior hippocampal decrease over time and a time-dependent increase of a posterior hippocampal fit to the 'old and semantically related items are similar' model might be taken as support against the suggested differential memory transformation over time in anterior and posterior hippocampal areas. It is to be noted, however, that these seemingly discrepant findings may be owing to the different methodological approaches. Whereas the ERS measures a change in reinstatement of an individual memory at test, the model-based analysis is directed at representational changes for a specific item category at test, i.e. recognition processes that are either specific to old items (model 1), shared by semantically related (model 2) or perceptually related (model 3) lures. Thus, our pattern of results might point to distinct patterns of changes in anterior vs. posterior hippocampus. Elucidating the distinct contributions of anterior vs. posterior regions of the hippocampus to recent and remote memories remains a challenge for future research.'

Moreover, we are now more careful in our wording when referring to the potential changes along the hippocampal long axis, referring now mainly to the results we directly obtained.

Reviewer #3

We thank the reviewer for the positive and very thoughtful comments. We are glad that he or she considers the topic of our manuscript to be very timely and of highest interest, our experimental design to be creative and well executed, our methodological approach of multivariate imaging to investigate the change of neural representations of memories over time clever and the results interesting.

Major concerns:

1) From figure 2C, supplementary table 1 and the pilot data, it becomes clear that semantically related (SR) pictures are not much less perceptually related than perceptually related (PR) items (both have a perceptual relatedness rating of ~6 while unrelated items have a rating of ~2). This perceptual relatedness in the semantic lures should be clearly mentioned, and the corresponding significance tests should be added.

RESPONSE: We completely agree that items containing the semantic gist of the original image are also to some extent perceptually related to the original image. In fact, this was an important part of the rationale of the present study as previous research on qualitative changes in the course of memory transformation did not allow to distinguish whether an increase in false alarms (FAs) to items carrying the semantic gist of originally encoded items results of a perceptual or semantic transformation (which we stated on pages 3-4, lines 82-85: 'Paradigms used in previous studies on time-dependent memory transformation in humans or rodents involved tests of transformation that were both semantically and perceptually similar to the original event and could thus not distinguish between different mechanisms of transformation.' and page 21, lines 530-532: 'Previous studies could not distinguish between these alternatives as test materials and contexts were typically both perceptually and semantically related to the original episode.'). For this very reason, we did not only include material that is semantically related to original material but also material carrying the perceptual, but not the semantic gist of the

original memory to systematically evaluate whether the nature of memory transformation is characterized by a semantic or perceptual transformation. However, we agree that it could be further highlighted that semantically related material is also perceptually related to originally encoded material which we do now on page 11, lines 282-286:

'As expected, semantically related lures were also rated higher in perceptual relatedness to their corresponding old picture ($M = 5.50$, $SEM = 0.20$) compared to unrelated lures (paired t -test: $t_{50} = -3.76$, $p < 0.001$, $d = 2.22$) but tended to be rated lower in perceptual relatedness compared to perceptually related images (paired t -test: $t_{50} = 2.20$, $p = 0.094$, $d = 0.30$).'

Additionally, we now specifically point out that perceptually related items were rated to be significantly more perceptually related than semantically related to the original items, whereas semantically related items were rated to be more semantically than perceptually related to the old items. Please see page 11, line 286-289:

'Importantly, perceptually related lures were rated as significantly higher in perceptual than in semantic relatedness to their corresponding old image (paired t -test: $t_{51} = -16.38$, $p < 0.001$, $d = 2.71$) while semantically related items were rated as significantly more semantically than perceptually related to their corresponding old image (paired t -test: $t_{51} = 16.67$, $p < 0.001$, $d = 2.31$).'

Furthermore, it is to be noted that semantically related items were rated to be semantically significantly more related to the original (i.e. old) items than perceptually related items ($p < 0.001$; page 10, lines 273-276.

Because SR items show high perceptual similarity to old items, only comparing SR items with PR items allows to draw conclusions about the effects of presence or absence of semantic relationships, and for any conclusions about the effects of presence or absence of a perceptual relationship, a comparison of PR items with unrelated items is necessary. This does not affect the majority of analyses, but the contrasts comparing SR-PR or PR-UR and results of the unrelated items should be included where relevant, and a discussion should be added.

RESPONSE: We agree and, as indicated by the reviewer, the vast majority of our analyses on the influence of semantic and perceptual relatedness focused on the respective contrasts. In particular, we show that the FA rate was significantly increased for semantically related relative to perceptually related (and unrelated) items (please see page 7, lines 185-188). Moreover, there was no difference in the FA rate for unrelated and perceptually related lures at the 28d test. We have added this important finding on page 7, lines 191-196:

'This pattern of results suggests a semantic and not a perceptual memory transformation over time. Perceptually related items showed a trend towards more FAs compared to unrelated items at 1d (paired t -test: $t_{50} = -2.31$, $p = 0.073$, $d = 0.47$) which was not evident in the 28d group (paired t -test: $t_{50} = -0.88$, $p = 0.767$, $d = 0.17$), suggesting that while memory for perceptual features may influence memory performance after a short retention interval, perceptual relatedness had no relevant effect on memory at the later delay.'

Please note that all our analyses for time-dependent changes in memory specificity include contrasts between all lure types (e.g. our ANOVA models are including the factor lure type contrasting the three lure types with each other) and were reported if significant.

Moreover, it is important to note that we included, in addition to the comparisons on items categories, also a parametric analysis in which we analyzed at trial-level the influence of both semantic and perceptual relatedness (based on participants' individual ratings on Day 3). This analysis is in our view ideally suited to dissociate the influences of semantic and perceptual relatedness on time-dependent changes in memory. This analysis clearly shows that the semantic but not perceptual relatedness drives the FA rate in the delayed recognition test (please see page 11, lines 290-298 and Figure 2C, right panel).

A supplementary analysis within the SR lures comparing FAs with high and low perceptual relatedness ratings might further validate the argumentation.

RESPONSE: We agree, have now performed the respective analysis and report the results on page 11, lines 299-308:

‘As semantically related items are usually also high in perceptual relatedness to original stimuli, we additionally analyzed whether the delay-dependent increase in FAs for semantically related items was equally evident in semantically related lures low (≤ 5) vs. high (> 5) in perceptual relatedness. A generalized LMM with the factors perceptual relatedness level (low vs. high), delay (1d vs. 28d) and emotion (neutral vs. negative) and the random intercept of participants and stimuli confirmed our previous finding of an emotionally enhanced increase in the probability for a FA for semantically related lures over time (delay \times emotion: $\beta = 0.93$, $z = 2.19$, $p = 0.029$). Importantly, this analysis did not reveal any main effect ($p = 0.476$) nor interaction (all $p > 0.215$) for the factor perceptual relatedness level, indicating that FAs for semantically related lures did not differ between items low vs. high in perceptual relatedness.’

Please note, that our analyses included also a generalized linear mixed model (LMM), in which we analyzed the impact of both the semantic and perceptual relatedness rating for each item and its relation to subsequent memory specificity (i.e. FAs) and this analysis, taking the complete range of both semantic and perceptual relatedness ratings for all items into account. This analysis showed that time-dependent increases in FAs were driven by semantic but not by perceptual relatedness (please see page 11, lines 290-298 and Figure 2C, right panel).

2) It is highly interesting that perceptually related items do not seem to produce more false alarms than unrelated items. It is mainly semantically related false alarms that are produced. This speaks in favor of the idea that the memories are mostly stored in semantic (‘abstract’) form rather than in perceptual (‘photographic’) form. The lack of PR FAs should be discussed.

RESPONSE: We fully agree and share the reviewers view that this finding is extremely interesting. We are happy to discuss this finding now further on page 21-22, lines 548-550:

‘Interestingly, however, our finding that only semantically related, but not perceptually related, items induce significantly higher FAs compared to unrelated lures indicates that memories are, regardless of temporal delay, mostly stored in a semantic rather than in a perceptual form.’

3) The authors interpret the increase in semantic FAs as a sign of a semantic transformation. It can, however, also mean that previously available perceptual detail, which prevented the false alarm at 1d, has been lost over time, while coarse semantic information was still available at 28d. Instead of a semantic transformation, the results would then rather describe forgetting of identifiable detail. That detail could pertain to the perceptual (photographic representation of blue bikes) domain or to the semantic domain (abstract representation of “blue bikes”).

In fact, the much simpler explanation is that both semantic and detailed perceptual information is encoded from the start and lost over time than that the brain is occupied during the 28 days after encoding random images with their re-coding and semanticization. In that view, the data would not be in line with MTT/TTT and repeated reactivation and re-encoding (line 473).

RESPONSE: Thank you for this interesting perspective, which we have now added as an alternative explanation. Please see page 21, line 542-548:

‘Alternatively, however, it could also be argued that previously available perceptual detail, which prevented the FAs at 1d, has been lost over time, while coarse semantic information was still available at 28d. Instead of a semantic transformation, our findings would then rather suggest forgetting of identifiable detail. This detail could pertain to the perceptual domain or to the semantic domain. In other words, both semantic and perceptual information could be encoded during initial encoding but then being forgotten at different rates over time.’

4) Similarly, the increased false alarm probability and higher ERS for SR items at 28d (figure 3 lower right) can be seen as simply a consequence of the better retention of semantic information compared to perceptual information. Higher ERS would result from a better retention of at least some semantic memory, which is indicated by higher probability of FA.

A similar figure to figure 3 lower right with probability of correct recall as a function of ERS (perhaps separately for neutral and negative items) could bring clarity to this issue. If stronger ERS correlates with higher correct recall as well as more semantic false alarms, then both indicate better memory retention. Without showing that higher ERS explicitly does not correlate with correct ('detailed') recall, the conclusion that "representations are unspecific in nature, reflecting semantically transformed memories" (line 333) is not warranted.

RESPONSE: We thank the reviewer for this very helpful advice. In response to this comment, we additionally analyzed the probability of an item to be correctly recalled ('detailed') by means of a binomial generalized LMM with emotion, delay and ERS and their interaction as fixed effects and the random intercept of stimulus set and participant. This analysis did not show any association of ERS with correct (detailed) memory (all $p > 0.688$). We now report the findings of this analysis on page 19, lines 468-472:

'Analyzing the probability of a detailed recognition (correct response for old items without FAs for related lures) by means of a binomial generalized LMM with the factors ERS, delay and emotion did not show an association of ERS with detailed memory (all $p > 0.688$; see Supplementary Figure 4), indicating that posterior hippocampal memory reinstatement after 28d is not associated with more detailed memory.'

5) I do not concur with the interpretation of the ERS analysis that there is a "change" or "reorganization" along the hippocampal long axis. While they find an increased reinstatement after 28d in the posterior hippocampus, the authors do not observe the (expected) higher reinstatement of recent memories in the anterior hippocampus. While it is an interesting finding to see a change in ERS in the posterior hippocampus from 1d to 28d, it does not permit any conclusions about representational changes along the anterior-posterior axis of the hippocampus. The authors could have used any other region that showed no change in ERS for the same argument. Conclusions of a change or reorganization along the hippocampal axis have to be removed.

RESPONSE: Our hypothesis of a memory transformation from the anterior to the posterior hippocampus was based on previous findings connecting the anterior hippocampus with the recollection of contextual details (Wing et al., 2015), novelty detection (Cowan et al., 2021), source memory specificity (Langnes et al., 2019), constructing autobiographical memories (Audrain et al., 2022) and detailed future event representations (Addis et al., 2011) and the posterior hippocampus with remote, generalized memory representations (Bonnici et al., 2012; Dandolo & Schwabe, 2018; Tomparry & Davachi, 2017). In line with these findings, our model-based Representational Similarity Analysis (RSA) results showed a decrease in specific memory representations in the anterior hippocampus over time while the posterior hippocampus reinstated remote memories that were rather unspecific in nature as they were prone to be confused with semantically related, but not perceptually related material. We agree, however, that our wording here may have been misleading as we did not see a decrease in anterior hippocampal ERS paralleling the posterior hippocampal increase in ERS. We are therefore now more careful in our conclusions, significantly softened our interpretations regarding time-dependent changes along the hippocampal long axis and changed our wording to focus more directly on our actual findings.

For instance, we changed our subtitle 'Memory reinstatement becomes reliant on the posterior hippocampus over time', please see page 17, line 421:

'Posterior hippocampal memory reinstatement increases over time'

Furthermore, we changed the statement 'Memory reinstatement became reliant on the posterior hippocampus with increasing delay after encoding' in the legend of Figure 4, please see page 18, line 436:

'Posterior hippocampal memory reinstatement increased over time'

We further changed the statement 'Thus, these findings indicate that, over time, memory reinstatement becomes increasingly reliant on the posterior hippocampus, and that those reinstated, remote memory representations are unspecific in nature reflecting semantically, but not a perceptually transformed memories.', please see page 20, lines 506-508:

'Taken together, our memory reinstatement analyses indicated that the posterior hippocampus is associated with the reinstatement of remote memories that may be rather unspecific in nature representing the semantic, but not perceptual, gist of the original memory.'

We also changed the statement 'memory reinstatement became increasingly reliant on the posterior hippocampus over time, which represented the semantic, but not perceptual, gist of the memory', please see page 21, lines 523-525:

'the posterior hippocampus was associated with the reinstatement of remote memories that were rather unspecific in nature and likely to be confused with the semantic, but not perceptual gist of the original memory'

Furthermore, we now discuss evidence that would not be in line with a differential transformation in anterior vs. posterior parts of the hippocampus over time more explicitly on pages 23-24, lines 598-616:

'Although the specific increase in posterior hippocampal ERS over time and its direct association with our behavioral indicator of semantic transformation (i.e. FAs to semantically related lures) and the decrease in distinct memory representations in the anterior hippocampus over time supports the idea of a time-dependent transformation along the hippocampal anterior-posterior axis with detailed memory representations in the anterior hippocampus and remote, gist-like representations in the posterior hippocampus, it is important to note that we did not observe a decrease in the anterior hippocampal memory reinstatement over time. Moreover, posterior hippocampal memory representations did not increase in similarity to semantically related material as indicated by our model-RSA analyses. The absence of an anterior hippocampal decrease over time and a time-dependent increase of a posterior hippocampal fit to the 'old and semantically related items are similar' model might be taken as support against the suggested differential memory transformation over time in anterior and posterior hippocampal areas. It is to be noted, however, that these seemingly discrepant findings may be owing to the different methodological approaches. Whereas the ERS measures a change in reinstatement of an individual memory at test, the model-based analysis is directed at representational changes for a specific item category at test, i.e. recognition processes that are either specific to old items (model 1), shared by semantically related (model 2) or perceptually related (model 3) lures. Thus, our pattern of results might point to distinct patterns of changes in anterior vs. posterior hippocampus. Elucidating the distinct contributions of anterior vs. posterior regions of the hippocampus to recent and remote memories remains a challenge for future research.'

We have further removed all interpretations regarding a reorganization along the hippocampal long axis.

6) *The analysis of retrieval similarity between stimulus types is not entirely convincing for several reasons:*

- First and foremost, the four predicted similarity matrices are not plausible. RSM model 1 assumes high correlations between response patterns of all old items. It thus seems to assume that all old items have been learned in the same session and should therefore have some similarity (the context during encoding). It therefore does not reflect item specificity (contrary to line 350), but rather the generalizable aspects of the memories. Model 2 assumes high similarities also between activity evoked by stimuli that are semantically related to the old stimuli. However, there is no similarity that I could think of between SR items, except their link to the old items which in turn have a common context. However, the model assumes the association between SR and old items to be weaker than the association between unrelated SR items. Because their only link is the association with items that were learned in the same context, SR items should be assumed to have a weaker relation between each other than with the associated old item. Model 3 is probably incorrectly depicted in figure 4. PR

items should show similarity to old items, not to SR items. Again, as for SR items, I do not see why one could assume a high similarity between PR items, which are unrelated to each other.

RESPONSE: Thanks for the opportunity to clarify our rationale for the chosen models. Our model-based Representational Similarity Analysis (RSA) approach was not designed to analyze similarities between individual stimuli but similarities in neural pattern representations depending on the category of stimuli and the delay after encoding as an indicator of memory transformation processes. For model 1, as we stated on page 12 lines 320-32 ('similar representations for old pictures that are distinct from patterns for all novel stimuli') we did not expect distinct representations between old items but between old items and lures. We expected old items to be similar, as they – contrary to lures – should equally initiate recognition processes in neural areas relevant for specific memory representations. Thus, this model does not expect item specificity but recognition processes that are specific to old items. The title that we used to paraphrase model 1 ('old items are distinct') in Figure 3A (formerly Figure 4A) and throughout the text may have contributed to this misunderstanding. We therefore changed the model title to 'old items are distinct to all lures' (e.g. Figure 3A and line 321) and further make our rationale for the model clearer now on page 12, lines 325-329:

'Note that for all models we expected old items to be represented more similarly, as they should equally initiate recognition processes in neural areas relevant for memory representations that, in case of recent, specific memory, should be distinct from all lures (model 1), or, in case of transformed memory representations, similar to either semantically (model 2) or perceptually (model 3) related lures.'

And in the methods section on page 32, lines 879-881:

'Note that for all models we expected old items to be represented more similarly, as they should equally initiate recognition processes in neural areas relevant for specific (model 1) or transformed (model 2 and model 3) memory representations.'

As for the other models, we hypothesized, that in the course of a semantic (model 2) or perceptual (model 3) transformation, semantically or perceptually related items should initiate similar activation patterns to old items in regions relevant for processing semantically or perceptually transformed memories, respectively. Moreover, in the case of a semantic transformation (model 2), semantically related lures were expected to equally initiate activation patterns corresponding to retrieving semantic information. In the case of a perceptual transformation (model 3) we expected perceptually related lures to be represented similarly as they equally should initiate activation patterns corresponding to processing perceptual information. Although this rationale is based on a previous account to investigate memory transformation processes (Dandolo & Schwabe, 2018), we agree that the similarity of activation patterns between lures might be less intuitive than the similarity in activation patterns when processing old items vs. lures. We therefore now performed an alternative analysis repeating our model-based RSAs with a focus on the relationship between old items and lures, i.e. excluding model-assumptions regarding the relationship between lures (please see the new Figure 3A). We focus now on this revised model-based RSA in our results section, as this version may be more straightforward, and report a very similar main pattern of results as in our original analysis (please see Figure 3B-C).

- The main effects in the anterior hippocampus stem from a negative fit of the model at the 28d interval. A negative fit would indicate that the model is not well suited to describe the actual activation patterns. Specifically, a negative model fit for Model 1 at 28d could mean that items lose their common, generalizable response pattern (context) and evoke more dissimilar responses, which explicitly suppress each other. Results can therefore not be interpreted as a detailed representation in the anterior hippocampus that is lost over time.

Response: Please note that we adapted and clarified our model-based RSA approach (see Figure 3A) in response to the previous comment of this reviewer. The results of the revised model-based RSA shows a significant positive correlation (i.e. a significant positive model fit) between anterior hippocampal pattern activations and model 1 (one-sample t -test: $p < 0.001$, $t_{25} = 3.82$) in the 1d but no significant fit to model 1 in the 28d group (one-sample t -test: $p = 0.891$, $t_{25} = -0.14$), indicating specific anterior hippocampal memory representations at the 1d interval that decrease over time.

Nevertheless, we explored the possible reason for the negative fit to model 1 for the anterior hippocampal pattern representations at 28d with our previous models (which, again, is not present using the current, adapted approach). Generally, our model-based approach assesses how well neural data fits a conceptual model by correlating the neural Representational Similarity Matrix (RSM) of a specific ROI (here anterior hippocampus) with an a-priori defined RSM (model 1) using Spearman's *rho*. As model 1 predicts that the similarity between old items is higher than the similarity between all other items, a negative model fit, i.e. negative Spearman's *rho*, would indicate that the similarity structure of the neural RSM might be reversed to the predicted similarity structure. Exploring the anterior hippocampal RSM for neutral items at 28d (which showed the negative model fit in the previous model-based analyses) revealed a slightly lower similarity for old items ($r = -0.004$) compared to the similarity between all other items ($r = -0.000008$), therefore indeed reversing the similarity structure predicted by model 1, leading to a negative model fit for this condition. The fact that this negative correlation between remote anterior hippocampal representations and model 1 disappeared after adapting the models (see Figure 3A) indicates that this result was mainly driven by the similarity between novel items, potentially reflecting novelty-detection processes by the anterior hippocampus (Cowan et al., 2021), which may have outweighed distinct memory representations at this remote time point.

7) *Some other additional analyses would be helpful:*

- *Analyses on PR and SR items could be done depending on whether an image evoked a false alarm or not.*

RESPONSE: As expected, the FA rate was low after a 1d retention interval. Furthermore, for perceptually related items there was no increase in FAs over time, thus for these items the FA rate remained low, which is an important finding that relates to the key questions of the present study (i.e. semantic vs. perceptual nature of memory transformation over time). Due to the relatively low overall FA rate, however, a neural analysis depending on whether an image evoked a FA or not was not reasonable. Instead, we decided to perform the fMRI analyses at the category level because this ensured a high statistical power (60 trials per item type) and we reasoned that the neural data may be more sensitive to time-dependent changes in memory representations than the dichotomous ('yes' vs. 'no') behavioral responses. We linked these neural changes to the behavioral level through correlational analyses and theoretically motivated models. Nevertheless, we agree that contrasting the activity associated with FAs directly might be of interest and we discuss this now on page 24, line 617-621:

'Furthermore, it has to be noted that memory performance was overall high in the present study, in particular for the 1d-group, which did not allow an analysis of neural activity associated with FAs to specific types of lures. To enable an analysis focussed on incorrectly endorsed related material, future studies should thus consider increasing task difficulty, for instance by increasing the number of encoded items or extending the retention interval.'

- *The ERS analysis might provide more information, if the ERS of later remembered vs forgotten (or detailed vs transformed vs forgotten) items would be compared separately instead of only looking at group differences.*

RESPONSE: Please note that in addition to the analysis of the delay-dependent change in ERS along the anterior-posterior hippocampal axis, we analyzed whether posterior hippocampal ERS was associated with FAs to semantically or perceptually related lures (see Figure 4, lower right panel). We now additionally analyzed the relationship between posterior hippocampal ERS and correct recognition performance in the respective trial and report these results on page 19, lines 461-465:

'Analyzing the probability of a hit by means of a generalized LMM with ERS, emotion (neutral vs. negative), delay (1d vs. 28d) and their interaction as fixed effects and the random intercepts of participants and stimulus sets indicated that with increasing delay, posterior hippocampal memory reinstatement was significantly positively associated with the correct endorsement of an old item as 'old' (delay \times ERS: $\beta = 7.31$, $z = 2.17$, $p = 0.030$).'

Furthermore, we investigated the specificity for this recognition performance, i.e. whether posterior hippocampal memory reinstatement is associated with detailed recognition. Please see page 19, lines 465-472:

'However, correct memory could be supported by specific, detailed memory representations but also by more abstract, gist-like representations. We therefore further analyzed the specificity of the reinstated memories by taking into account the responses for corresponding related lures. Analyzing the probability of a detailed recognition (correct response for old items without FAs for related lures) by means of a binomial generalized LMM with the factors ERS, delay and emotion did not show an association of ERS with detailed memory (all $p > 0.688$; see Supplementary Figure 4), indicating that posterior hippocampal memory reinstatement after 28d is not associated with more detailed memory.'

Moreover, reducing our analysis of delay-dependent changes of ERS along the hippocampal long axis to correctly recognized items (i.e. hits) did not change our result of a significant increase in ERS from 1d to 28d in the left posterior hippocampus (delay \times long axis: $\beta = 0.01$, $t_{5107.81} = 3.13$, $p_{corr} = 0.004$). We are now reporting this result on page 19, line 456-459:

'Note that repeating this analysis after excluding items that were not correctly recognized (misses) did not change this result of a significant increase in posterior hippocampal ERS from 1d to 28d (delay \times long axis: $\beta = 0.01$, $t_{5107.81} = 3.13$, $p_{corr} = 0.004$).'

As indicated in the previous response, reducing our neuroimaging analysis to incorrect responses would not have allowed a sufficient number of data points and would lead to a very uneven distribution of trial numbers between groups. Nevertheless, we tried fitting a generalized LMM on correlations between neural representation patterns of old items at encoding and old items that were not correctly remembered at recognition testing (misses). However, even models with the simplest random effects structure, i.e. including only a random intercept of participants or stimuli, resulted in a 'singular fit' likely due to the data not being sufficiently informative to estimate a random effects variance greater than zero, thus underlining our view that this type of analysis is not feasible.

- The ERS analysis might also be done on a whole brain level. Although the hippocampus is central for episodic memory reinstatement, with the special interest of the present study in semantic generalization, it suggests itself to include a wider range of regions in the analysis.

RESPONSE: We agree and now performed additional, explorative ERS analyses for neocortical ROIs. We have added these results on page 20, lines 499-505:

'Furthermore, we explored delay-dependent changes in memory reinstatement in neocortical areas that have been previously implicated in long-term memory (IFG, vmPFC, aCC, precuneus, angular gyrus) as well as sensory control ROIs (occipital pole, Heschl's gyrus) by means of LMMs with the factors delay, emotion and their interactions as fixed effects. These analyses did not yield any delay-dependent variations in memory reinstatement, neither by old items (i.e. ERS; all $p > 0.164$) nor by semantically (all $p > 0.203$) or perceptually (all $p > 0.201$) related lures, suggesting that there was no relevant memory reinstatement in these areas.'

- Calculating Encoding-Retrieval-Similarity only for the original stimuli misses out on the important point whether PR or SR items also increase or decrease in similarity with the original items over time. In particular, a "semantic transformation" should increase similarity of activation patterns from 1d to 28d for SR with old items. If similarity does not increase, results would rather speak for forgetting of detail and retaining of coarse semantic information than for a semantic transformation.

RESPONSE: Per definition, ERS is computed between pattern representations during encoding and the same items during retrieval (see e.g. Ritchey et al., 2013; Staresina et al., 2012; Wing et al., 2015; Xiao et al., 2017). However, in response to this reviewer's comment, we now additionally analyzed the similarity between encoding-related activation patterns and corresponding lures by means of LMMs

with the factors emotion, long axis and delay as fixed effects and the random intercept of participants and stimuli and report these results on page 20, lines 489-498:

'While ERS is computed by correlating pattern representations of individual items during encoding and memory test, i.e. old items, we furthermore assessed the similarity elicited by perceptually or semantically related items at memory test and corresponding old items during encoding as a possible indicator for a reinstatement of the perceptual or semantic gist of the original memory. This analysis indicated that the posterior hippocampus tended to show a higher reinstatement of the semantic gist of the original memory compared to the anterior hippocampus (main effect long axis: $\beta = 0.004$, $t_{6124} = 1.94$, $p = 0.052$; see Supplementary Figure 5). No effect approached significance when analyzing hippocampal reinstatement of the perceptual gist (all $p > 0.235$), which further suggests that the posterior hippocampus might be specifically associated with the reinstatement of the semantic, but not perceptual, gist of the original memory.'

- *The RSA analyses show increasing model fit in all ROIs. Were there more ROIs tested or did all selected ROIs show the increase? Are there any cortical regions that do not show this pattern?*

RESPONSE: For our model-based RSA, we specifically included neocortical ROIs which have been previously implicated in long-term memory retrieval, i.e. the ventro-medial prefrontal cortex (vmPFC; Binder et al., 2009; Binder & Desai, 2011; Bowman & Zeithamova, 2018), inferior frontal gyrus (IFG; Sommer, 2016; Takashima et al., 2009), anterior cingulate cortex (aCC; Atucha et al., 2017; Frankland et al., 2004; Frankland & Bontempi, 2005; Sekeres et al., 2020), angular gyrus and the precuneus (Brodt et al., 2016, 2018). Note that we now adapted our model-based RSA approach in response to another comment by this reviewer. For those adapted, reduced models, post-hoc testing indicated a delay-dependent increase in model 2 only for the vmPFC ($p = 0.046$) and right angular gyrus ($p_{corr} = 0.011$), with a trend into the same direction for the precuneus ($p = 0.053$). We agree, however, that it might be interesting to include neocortical ROIs for which no increase in transformed memory representations over time is expected. Thus, similar to a previous approach (Ritchey et al., 2013), we now additionally analyzed pattern activations of sensory control-ROIs (bilateral occipital pole and Heschl's gyrus), please see page 16, lines 405-416 for the results of these analyses:

'Furthermore, we repeated this model-based RSA in the bilateral occipital pole and Heschl's gyrus as neocortical control regions for which we did not expect any increase in transformed memory representations over time. Analyzing activation patterns in those regions by means of delay (1d vs. 28d) \times model (1: 'old items are distinct from all lures' vs. 2: 'old and semantically related items are similar' vs. 3: 'old and perceptually related items are similar') \times emotion (neutral vs. negative) ANOVAs did not indicate any time-dependent change in fit of pattern representations in both, the occipital pole ($p > 0.414$) and Heschl's gyrus (all $p > 0.644$). Interestingly, activation patterns in the occipital pole showed an overall higher fit to model 3 ('old and perceptually related items are similar') compared to model 1 (paired t -test: $t_{50} = -4.87$, $p < 0.001$, $d = 0.67$) as well as model 2 (paired t -test: $t_{50} = -3.85$, $p = 0.001$, $d = 0.54$), regardless of temporal delay (main effect model: $F_{1,83,91.39} = 13.26$, $p < 0.001$, $\eta_p^2 = 0.21$). This finding most likely reflects the processing of overlapping visual features in old and perceptually related images.'

Minor concerns:

1) *Unrelated items were unrelated to the three others in a set, but were they also unrelated to the old images of the other 59 sets? How were relations between image sets assessed?*

RESPONSE: Unrelated images were matched to a respective old item in terms of subjectively perceived visual complexity, the depiction of people or animals by the first author and another independent rater. All unrelated (and perceptually related) images carried a different semantic gist than all other images, i.e. if one original image carried the semantic gist 'rental bikes' no other lure (or old item) besides the corresponding semantically related lure depicted rental bikes. We now further clarify this on pages 25-26, lines 671-676:

'The four pictures belonging to a set were matched to a respective old item in terms of subjectively perceived visual complexity, the depiction of people or animals by the first author and another independent rater. All unrelated (and perceptually related) images carried a different semantic gist than all other images, i.e. if one original image carried the semantic gist 'rental bikes' no other lure (or old item) besides the corresponding semantically related lure depicted rental bikes.'

Moreover, our finding that there were only very few FAs to unrelated lures is generally in line with the view that these did not overlap with old items.

2) It is highly interesting that the activity pattern in posterior hippocampus becomes more similar to encoding after 28d. This is difficult to explain with any current model of memory consolidation. Is it possible that the higher difficulty of retrieval at 28d had participants focus longer and more attentively on the images?

RESPONSE: We agree that this finding is very interesting. This result actually dovetails with previous findings suggesting that the posterior hippocampus might be implicated in remote, generalized memory representations (Bonnici et al., 2012; Dandolo & Schwabe, 2018; Tompary & Davachi, 2017).

We would like to emphasize that the presentation time of the stimuli in the encoding as well as in the recognition test was constant (3s) for all participants. Thus, it is in our view rather unlikely that the increased posterior hippocampal ERS in the 28d group can be explained by longer stimulus viewing at the 28d retention delay. Right after viewing a stimulus at memory testing, participants indicated via a button press whether they recognized the previously presented item or not. We now additionally analyzed the response time during this memory test as this response time may reflect priming and attentive processing effects. An LMM with the fixed effects of delay, emotion, posterior hippocampal ERS and their interaction and the random intercept of participants and stimuli did not show any main effect of delay ($\beta = 32.88$, $t = 0.81$, $p = 0.422$), emotion ($\beta = 10.13$, $t = 0.59$, $p = 0.556$), ERS ($\beta = 33.81$, $t = 0.15$, $p = 0.883$) nor any interaction of the included factors (all $p > 0.186$) on response time during memory testing, suggesting that at least response times, as an indicator of attentive processing, did not differ. We now included this control analysis on pages 19-20, lines 483-488:

'Furthermore, analyzing response times during memory testing, as an indicator of the attentiveness in the respective trial, by means of an LMM with delay, emotion, posterior hippocampal ERS and their interaction as fixed effects and the random intercept of participants and stimuli did not show any main effect ($p > 0.421$) nor interaction (all $p > 0.186$) of the included factors suggesting that the delay-dependent increase in posterior hippocampal ERS was not related to attentional differences during memory testing between groups.'

3) On page 9, the authors describe a mixed model analysis investigating the occurrence of categorized stimulus sets regarding their memory (detailed, transformed, forgotten sets). The authors should state how many sets are included in each category for each group. In general, false alarm rates are very low in the 1d-group and might affect the interpretability of this analysis.

RESPONSE: Thank you for the opportunity to clarify our approach. Here we analyzed the probability of a stimulus set to belong to a specificity category (detailed, perceptually transformed, semantically transformed or forgotten) by means of a trial-wise binomial (1 = belonging to a specific category, 0 = not belonging to a specific category) generalized LMM. Thus, all 60 items per specificity category and participant are included in each analysis except of the very few trials in which participants missed to indicate their memory for the presented item (missed responses), which on average led to only 1.22% (SEM = 0.40%) of missing data points per participant (no difference between delay groups; two-sample t -test: $t_{31,20} = -0.83$, $p = 0.420$). We now make this clearer on page 10 line 255-260:

'Thus, all 60 items per specificity category and participant are included in each analysis except of the very few trials in which participants missed to indicate their memory for the previously presented item (missed responses), which on average led to only 0.95% (SEM = 0.44%) of missing data points per participant (no difference between delay groups; two-sample t -test: $t_{31,20} = -0.83$, $p = 0.420$; see Supplementary Table 3 for an overview of the amount of stimulus sets per category).'

In addition, we now list the number of sets for each specificity category per emotion category and delay group in Supplementary Table 3.

4) *In figure 3C right, the number of stimuli for the extreme ends is also unclear. Here, it would be interesting to see the individual data points that underlie these predicted probabilities.*

RESPONSE: We were not sure whether the reviewer is referring to Figure 2C or the lower right panel in Figure 4 (formerly Figure 3; there was no panel 3C). However, our response applies to both of these panels equally. As these panels depict marginal effects plots, i.e. the probability of a FA (0 = no FA, 1 = FA) predicted by the generalized linear mixed model (LMM) fitted on the whole trial-wise data, the addition of individual data points wouldn't be accurate. Please note that all individual data points for the x-axis (averaged over trials per participant and lure type) are depicted on the left-hand side of the figure (Figure 2C left panel). Furthermore, we now plot the distribution of individual similarity values (averaged over trials per participant) in Supplementary Figure 3.

It is also not entirely clear how this figure was generated. A gLMM is mentioned, which would imply linear predictors, but the relationship between relatedness and FA rate is depicted as a smooth curve.

RESPONSE: Thanks for the opportunity to clarify this. We analyzed the occurrence of the binomial variable FA (0 = no FA, 1 = FA) by means of a generalized LMM with a logit function, i.e. a multi-level binomial logistic regression analysis. Thus, a curved association between the dependent variable and significant predictors is expected. We now clarify this when explaining this model in the methods on page 30, lines 802-805:

'The individual stimulus relatedness ratings on Day 3 further allowed us to analyze FAs by means of a binomial generalized LMM with a logit function and the factor delay (1d vs. 28d), emotion (neutral vs. negative), semantic relatedness, perceptual relatedness and their interactions as fixed effects and the random intercept of participants and stimuli.'

Furthermore, we now explicitly mention how plots have been generated on line page 30, lines 818-820:

'Results were visualized by utilizing bar plots and individual data points with the package ggplot2⁶⁷ and plotting marginal effects of generalized LMMs with the package sjPlot⁶⁸.'

5) *If I understand the analyses correctly, the unpaired t-tests and the main effect of delay (1d/28d) in line 166 should be redundant.*

RESPONSE: Thanks for pointing this out. We now report only the main effect of delay on page 6, line 168.

6) *In line 261 (page 10) the authors should probably refer to figure 2C instead of 2B. They should refer to figure 2B when reporting the results of the mixed model in lines 244-256.*

RESPONSE: Thanks a lot for noticing this mistake, which has now been corrected.

7) *In line 214 and 307 and 395 it should say *p<.050.*

RESPONSE: Thanks for noticing this mistake, which has now been corrected.

8) *It is unclear why the authors state in line 252 that better memory for emotional items comes "at the cost of reduced memory for semantic details." There are more items correctly remembered and there*

is memory for the semantic gist for additional items, indicating at least some residual semantic memory for those items. Both speak for better semantic memory for emotional material.

RESPONSE: We agree that our wording has been misleading and changed it to ‘at the cost of reduced memory specificity’ on page 10, lines 267-272:

‘Even more importantly, emotionally negative pictures were significantly more often semantically transformed over time (z-test: $z = -4.31$, $p < 0.001$) than neutral ones (z-test: $z = -2.17$, $p = 0.030$; delay \times emotion: $\beta = 0.66$, $z = 2.46$, $p = 0.014$), in line with findings suggesting that superior memory for emotional material, indicated here by a slower forgetting rate, may come at the cost of reduced memory specificity^{19,23-25}.’

9) *The authors should also state whether the “main effect delay” for the aCC and IFG is significant (line 431-436) and report statistical parameters.*

RESPONSE: We agree. Please note that this comment refers to a previous analysis which we replaced with an alternative model-RSA approach as a result of a previous comment of this reviewer. In this new, alternative analysis, neither the aCC nor IFG shows significant delay-dependent changes in fit to model 2 (please see page 16 lines 403-404 and Figure 3C, lower panel).

10) *What is meant by “we subtracted run-related pattern variances” in line 780?*

RESPONSE: We thank the reviewer for the opportunity to further clarify our approach to correct for drift-related variances in representational pattern similarities (Alink et al., 2015; Mumford et al., 2014). Specifically, we computed the mean pattern similarity for each run-combination, i.e. between all comparisons of two items presented in the same run (run 1, run 2 or run 3) and the pattern similarity of each between-run comparison (run 1 and run 2, run 2 and run 3 or run 3 and run 1). Those run-related pattern similarity values were subsequently subtracted from each correlation value of the corresponding run-combination. We now further clarify this on page 32, lines 866-870:

‘Next, we computed the mean pattern similarity for comparisons within each of the three runs and for each between-run combination (run 1 and run 2, run 2 and run 3 or run 3 and run 1). Those run-related pattern similarities were then subtracted from each correlation estimate of the corresponding run-combination to account for inflated correlations as a function of temporal proximity between scans^{72,73}.’

11) *There are a number of references that cite pre-prints without this being clear in the text. I do not believe that the non-peer reviewed literature should be put on the same footing as the peer-reviewed literature.*

RESPONSE: We agree that basing statements exclusively on manuscripts which have not undergone peer-review would be problematic. However the three cited preprints (Alink et al., 2015; Audrain et al., 2022; Cox et al., 2021) are listed as additional, never as sole literature for our statements. As those manuscript enable us to include the most recent findings in our manuscript, we would prefer to keep them, if both the reviewer and the editor agree.

12) *Is the 1997 version of the IAPS the same as the 1988 version? [Lang, P., Ohman, A., & Vaitl, D. (1988). The international affective picture system Gainesville, FL: University of Florida, Centre for Research in Psychophysiology]*

RESPONSE: We apologize for the misleading reference. We used the IAPS stimuli from 2008, which to our knowledge are the most recent ones. We have adjusted the reference as indicated on the IAPS website (<https://csea.php.ufl.edu/media.html>), please see page 39, lines 1065-1064:

'56. Lang, P.J., Bradley, M.M. & Cuthbert, B.N., *The International Affective Picture System (IAPS): Affective ratings of pictures and instruction manual. Technical Report A-8.* (University of Florida, 2008)'

13) For the first supplementary result, please provide a figure (or at least means), not just significance tests.

RESPONSE: We agree and are now visualizing these results in Supplementary Figure 6.

14) *Supplementary Table 1: There might be a copy error for perceptual relatedness of semantically related negative and unrelated negative values, as both are identical.*

RESPONSE: Thank you for noticing this mistake, which has now been corrected.

References

- Addis, D. R., Cheng, T., P. Roberts, R., & Schacter, D. L. (2011). Hippocampal contributions to the episodic simulation of specific and general future events. *Hippocampus*, 21(10), 1045–1052.
<https://doi.org/10.1002/hipo.20870>
- Alink, A., Walther, A., Krugliak, A., van den Bosch, J. J. F., & Kriegeskorte, N. (2015). *Mind the drift—Improving sensitivity to fMRI pattern information by accounting for temporal pattern drift.* bioRxiv.
<https://doi.org/10.1101/032391>
- Atucha, E., Vukojevic, V., Fornari, R. V., Ronzoni, G., Demougin, P., Peter, F., Atsak, P., Coolen, M. W., Papassotiropoulos, A., McGaugh, J. L., de Quervain, D. J.-F., & Roozendaal, B. (2017). Noradrenergic activation of the basolateral amygdala maintains hippocampus-dependent accuracy of remote memory. *Proceedings of the National Academy of Sciences*, 114(34), 9176–9181.
- Audrain, S., Gilmore, A. W., Wilson, J. M., Schacter, D. L., & Martin, A. (2022). *A role for the anterior hippocampus in autobiographical memory construction regardless of temporal distance.* bioRxiv.
<https://doi.org/10.1101/2022.05.01.490212>
- Binder, J. R., & Desai, R. H. (2011). The neurobiology of semantic memory. *Trends in Cognitive Sciences*, 15(11), 527–536. <https://doi.org/10.1016/j.tics.2011.10.001>
- Binder, J. R., Desai, R. H., Graves, W. W., & Conant, L. L. (2009). Where is the semantic system? A critical review and meta-analysis of 120 functional neuroimaging studies. *Cerebral Cortex*, 19(12), 2767–2796. <https://doi.org/10.1093/cercor/bhp055>

- Bonnici, H. M., Chadwick, M. J., Lutti, A., Hassabis, D., Weiskopf, N., & Maguire, E. A. (2012). Detecting Representations of Recent and Remote Autobiographical Memories in vmPFC and Hippocampus. *Journal of Neuroscience*, *32*(47), 16982–16991. <https://doi.org/10.1523/JNEUROSCI.2475-12.2012>
- Bowman, C. R., & Zeithamova, D. (2018). Abstract memory representations in the ventromedial prefrontal cortex and hippocampus support concept generalization. *The Journal of Neuroscience*, *38*(10), 2605–2614. <https://doi.org/10.1523/JNEUROSCI.2811-17.2018>
- Brodts, S., Gais, S., Beck, J., Erb, M., Scheffler, K., & Schönauer, M. (2018). Fast track to the neocortex: A memory engram in the posterior parietal cortex. *Science*, *362*(6418), 1045–1048. <https://doi.org/10.1126/science.aau2528>
- Brodts, S., Pöhlchen, D., Flanagin, V. L., Glasauer, S., Gais, S., & Schönauer, M. (2016). Rapid and independent memory formation in the parietal cortex. *Proceedings of the National Academy of Sciences*, *113*(46), 13251–13256. <https://doi.org/10.1073/pnas.1605719113>
- Cowan, E. T., Fain, M., O’Shea, I., Ellman, L. M., & Murty, V. P. (2021). VTA and anterior hippocampus target dissociable neocortical networks for post-novelty enhancements. *The Journal of Neuroscience*, *41*(38), 8040–8050. <https://doi.org/10.1523/JNEUROSCI.0316-21.2021>
- Cox, W., Meeter, M., Kindt, M., & van Ast, V. (2021). *Time-dependent emotional memory transformation: Divergent pathways of item memory and contextual dependency*. PsyArXiv. <https://doi.org/10.31234/osf.io/b3w24>
- Dandolo, L. C., & Schwabe, L. (2018). Time-dependent memory transformation along the hippocampal anterior–posterior axis. *Nature Communications*, *9*(1), 1205. <https://doi.org/10.1038/s41467-018-03661-7>
- Frankland, P. W., & Bontempi, B. (2005). The organization of recent and remote memories. *Nature Reviews Neuroscience*, *6*(2), 119–130. <https://doi.org/10.1038/nrn1607>
- Frankland, P. W., Bontempi, B., Talton, L. E., Kaczmarek, L., & Silva, A. J. (2004). The involvement of the anterior cingulate cortex in remote contextual fear memory. *Science*, *304*(5672), 881–883. <https://doi.org/10.1126/science.1094804>
- Langnes, E., Vidal-Piñeiro, D., Sneve, M. H., Amlien, I. K., Walhovd, K. B., & Fjell, A. M. (2019). Development and decline of the hippocampal long-axis specialization and differentiation during encoding and retrieval of episodic memories. *Cerebral Cortex*, *29*(8), 3398–3414. <https://doi.org/10.1093/cercor/bhy209>

- Mumford, J. A., Davis, T., & Poldrack, R. A. (2014). The impact of study design on pattern estimation for single-trial multivariate pattern analysis. *NeuroImage*, *103*, 130–138.
<https://doi.org/10.1016/j.neuroimage.2014.09.026>
- Poppenk, J., Evensmoen, H. R., Moscovitch, M., & Nadel, L. (2013). Long-axis specialization of the human hippocampus. *Trends in Cognitive Sciences*, *17*(5), 230–240.
<https://doi.org/10.1016/j.tics.2013.03.005>
- Ritchey, M., Wing, E. A., LaBar, K. S., & Cabeza, R. (2013). Neural similarity between encoding and retrieval is related to memory via hippocampal interactions. *Cerebral Cortex*, *23*(12), 2818–2828.
<https://doi.org/10.1093/cercor/bhs258>
- Sekeres, M. J., Moscovitch, M., Grady, C. L., Sullens, D. G., & Winocur, G. (2020). Reminders reinstate context-specificity to generalized remote memories in rats: Relation to activity in the hippocampus and aCC. *Learning & Memory*, *27*(1), 1–5.
- Sekeres, M. J., Winocur, G., & Moscovitch, M. (2018). The hippocampus and related neocortical structures in memory transformation. *Neuroscience Letters*, *680*, 39–53.
<https://doi.org/10.1016/j.neulet.2018.05.006>
- Sommer, T. (2016). The emergence of knowledge and how it supports the memory for novel related information. *Cerebral Cortex*, *27*, 1906–1921. <https://doi.org/10.1093/cercor/bhw031>
- Staresina, B. P., Henson, R. N. A., Kriegeskorte, N., & Alink, A. (2012). Episodic reinstatement in the medial temporal lobe. *Journal of Neuroscience*, *32*(50), 18150–18156.
<https://doi.org/10.1523/JNEUROSCI.4156-12.2012>
- Takashima, A., Nieuwenhuis, I. L. C., Jensen, O., Talamini, L. M., Rijpkema, M., & Fernandez, G. (2009). Shift from hippocampal to neocortical centered retrieval network with consolidation. *The Journal of Neuroscience*, *29*(32), 10087–10093.
- Tompary, A., & Davachi, L. (2017). Consolidation promotes the emergence of representational overlap in the hippocampus and medial prefrontal cortex. *Neuron*, *96*(1), 228–241.
<https://doi.org/10.1016/j.neuron.2017.09.005>
- Wing, E. A., Ritchey, M., & Cabeza, R. (2015). Reinstatement of individual past events revealed by the similarity of distributed activation patterns during encoding and retrieval. *Journal of Cognitive Neuroscience*, *27*(4), 679–691. https://doi.org/10.1162/jocn_a_00740

Xiao, X., Dong, Q., Gao, J., Men, W., Poldrack, R. A., & Xue, G. (2017). Transformed neural pattern reinstatement during episodic memory retrieval. *The Journal of Neuroscience*, *37*(11), 2986–2998.
<https://doi.org/10.1523/JNEUROSCI.2324-16.2017>

Zeidman, P., & Maguire, E. A. (2016). Anterior hippocampus: The anatomy of perception, imagination and episodic memory. *Nature Reviews Neuroscience*, *17*(3), 173–182.
<https://doi.org/10.1038/nrn.2015.24>

REVIEWER COMMENTS

Reviewer #1 (Remarks to the Author):

In my opinion, the authors have thoroughly and successfully addressed all of the points I raised. The implemented changes have considerably improved the manuscript, and I don't have any other suggestions or major concerns. I believe that the work presented includes highly relevant results derived from high-quality research that will have a significant impact in the field of long-term memory. For these reasons, I recommend this manuscript for publication in Nature Communications.

Reviewer #2 (Remarks to the Author):

I appreciate the clarifications the authors provide to my main analytic concern regarding low numbers of trials in their analyses. It is now clear that for these analyses, they are using all trials regardless of memory type/success. However, this now raises another concern which is how to interpret neural activation patterns at a delay (especially as significant one) where episodic memory is absent or so very fragmented. If one is including all trials irrespective of memory, you are no longer looking at memory representations. Prior work on memory consolidation examines how the memory representations may change with time for information that is still remembered. Unfortunately, this changes my interpretation of all of the reported data and raises serious concerns about whether and how the neural results have any bearing on memory representations.

Reviewer #3 (Remarks to the Author):

All my concerns have been perfectly addressed in this revision. I want to thank the authors explicitly for providing all these additional results and explanations, which make this excellent manuscript even more intriguing than it already was.

Responses to Reviewers

Reviewer #1

In my opinion, the authors have thoroughly and successfully addressed all of the points I raised. The implemented changes have considerably improved the manuscript, and I don't have any other suggestions or major concerns. I believe that the work presented includes highly relevant results derived from high-quality research that will have a significant impact in the field of long-term memory. For these reasons, I recommend this manuscript for publication in Nature Communications.

Response: We thank the reviewer for these positive and encouraging comments and are delighted that this reviewer recommends our manuscript for publication in *Nature Communications*.

Reviewer #2

We are glad that this reviewer appreciates our clarifications of the design and power of our neuroimaging analyses and thank this reviewer, again, for the very helpful and constructive feedback on our manuscript.

It is now clear that for these analyses, they are using all trials regardless of memory type/success. However, this now raises another concern which is how to interpret neural activation patterns at a delay (especially as significant one) where episodic memory is absent or so very fragmented. If one is including all trials irrespective of memory, you are no longer looking at memory representations. Prior work on memory consolidation examines how the memory representations may change with time for information that is still remembered. Unfortunately, this changes my interpretation of all of the reported data and raises serious concerns about whether and how the neural results have any bearing on memory representations.

Response: We thank the reviewer for the opportunity to clarify the rationale for our analytic approach. We agree that when reporting the results of our model-based Representational Similarity Analysis (RSA) we should have emphasized that we compared the activation patterns of all encoded stimuli to avoid any misunderstanding that our manuscript might have been limited to trials that were behaviorally identified as “remembered”. Therefore, we have now gone through the manuscript and adjusted relevant sections to emphasize that the similarity between all encoded items and lures was taken into account. In particular, we changed “memory representations” to “representations of encoded events” to more precisely reflect the actual analyses. For example, please see page 2, lines 28-30:

“Model-based MRI analyses revealed time-dependent increases in semantically, but not perceptually, transformed representations of encoded events in prefrontal and parietal cortices, while detailed pattern representations in the anterior hippocampus declined over time.”

Page 4, lines 111-112:

“Within the hippocampus, the anterior hippocampus was associated with distinct representations of encoded events that declined with increasing delay after encoding.”

Page 12, lines 314-315:

“Distinct pattern representations of encoded events in the anterior hippocampus decreased over time”

We acknowledge that several studies investigating memory consolidation have limited their neuroimaging analyses to trials with items that were correctly remembered. However, specifically in long-term memory research, this approach is controversial as it disproportionately omits trials from remote time periods (Tallman et al., 2022). Moreover, in contrast to previous studies, our neural analyses were focused on the neural underpinnings of qualitative changes in the course of memory transformation over time rather than neural activity associated with accurate memories. Reducing our neuroimaging analyses to only remembered trials would most likely lead to an exclusion of memories particularly low in specificity, which would make an analysis of the neural underpinnings of time-dependent memory transformation virtually impossible. Moreover, multivariate analysis of fMRI data is expected to be much more sensitive to fine-grained changes in memory representations over time than the dichotomous (“old” vs. “new”) behavioral data. We now clarify this on page 32, lines 858-867:

“We opted for an analysis at the category level instead of relying on participants’ correct or incorrect responses because (i) we were interested in how the encoding-retrieval delay and lure type affected the similarity between representational patterns as an indicator of the specificity of the neural representational patterns rather than the underlying neural patterns of a specific behavioral response; (ii) (multivariate) neural data are much more sensitive to fine-grained changes in memory representations compared to behavioral data that is merely based on dichotomous ‘yes’ vs. ‘no’ (i.e. ‘old’ vs. ‘new’) responses; (iii) reducing analyses on incorrectly endorsed lures (FAs) would have resulted in an insufficient number of trials for the fMRI analyses while (iv) focusing solely on correctly endorsed items (hits) would exclude items that are particularly low in memory specificity, which are of particular interest when investigating the neural underpinnings of memory transformation over time.”

In the present study, we investigated the neural underpinnings of memory transformation by comparing neural activation pattern similarities between originally encoded items and lures to hypothesis-driven conceptual models that encapsulated unique similarity profiles between originally encoded items and different types of lures, i.e. they expected original items to be either represented distinct to all lures (model 1) or similar to either perceptually related (model 2) or semantically related (model 3) lures. This approach, i.e. comparing multivariate representational patterns of all experimental trials (irrespective of the correctness of the response) to conceptual models, allows inferences about the structure of neural representations (Diedrichsen & Kriegeskorte, 2017; Kriegeskorte et al., 2008; Nili et al., 2014) and has been successfully employed in previous studies to characterize memory representations, even at longer delays after encoding (Blumenthal et al., 2018; Clarke et al., 2016; Huffman & Stark, 2017; Kluein et al., 2019; Dandolo & Schwabe, 2018). Therefore, the model-based RSA approach we employed here is highly suitable for investigating changes in memory quality over time. We now clarify our rationale in more detail on page 32, lines 870-874:

“This approach, i.e. comparing multivariate representational patterns of all experimental trials (irrespective of the correctness of the response) to conceptual models, allows inferences about the structure of neural representations^{29,30,73} and has been successfully employed in previous studies to characterize memory representations, even at longer delays after encoding^{19,74-77} and is thus highly suitable for investigating changes in memory quality over time.”

While our model-based RSA allowed investigating time-dependent changes in representational similarity between (all) encoded and new item categories at memory test, we further investigated the reactivation of individual items during memory test, i.e. Encoding-Retrieval-Similarity (ERS) as a measure of trial-specific memory reinstatement (Xue, 2018). Specifically, we investigated delay-dependent changes in hippocampal memory reinstatement and correlated item-specific pattern similarity measures with behavior on a trial-by-trial-level, an approach that is, again, in accordance

with previous literature (LaRocque et al., 2013; Ritchey et al., 2013; Staresina et al., 2012; Xiao et al., 2017; Xue et al., 2010). Moreover, we showed that results of our ERS-analysis analysis did not change when either all original items or only items that were behaviorally indicated as remembered (hits) were included in this analysis (as mentioned in our manuscript on page 19, lines 458-461). Furthermore, we show that representational changes in hippocampal pattern reinstatement were directly linked to item recognition, and, most importantly, to our behavioral readout of semantic memory transformation, i.e. hits and false alarms to corresponding semantically related items, respectively (see page 19, lines 462-467 and lines 474-485 in our manuscript).

Finally, we would like to emphasize that, although our behavioral data demonstrate the expected memory transformation over time, they also show intact memory at the delayed test (see Figure 1). Even at the 28d delay, the hit rate is above 70 percent while the overall false alarm rate is below 25 percent. Thus, our data do not indicate that memory became absent or fragmented over time but that memory remained largely intact over time.

In sum, our behavioral data demonstrate intact memory even at the delayed test and, using well established analytical approaches, our fMRI data show direct links between neural representations and memory readouts. Thus, any concerns whether we are assessing actual memory representations are in our view not warranted. Our analytic approach is well in line with previous studies on long-term memories and reducing the analysis to only correctly recognized items may have interfered with a sophisticated analysis of qualitative changes of neural memory representations over time. Nevertheless, we have now changed the wording in our manuscript to make our analytical approach more explicit.

Reviewer #3

All my concerns have been perfectly addressed in this revision.

I want to thank the authors explicitly for providing all these additional results and explanations, which make this excellent manuscript even more intriguing than it already was.

Response: We thank the reviewer for the very positive and encouraging comments. We are delighted to read that this reviewer considers our manuscript excellent.

References

- Blumenthal, A., Stojanoski, B., Martin, C. B., Cusack, R., & Köhler, S. (2018). Animacy and real-world size shape object representations in the human medial temporal lobes. *Human Brain Mapping, 39*(9), 3779–3792. <https://doi.org/10.1002/hbm.24212>
- Clarke, A., Pell, P. J., Ranganath, C., & Tyler, L. K. (2016). Learning warps object representations in the ventral temporal cortex. *Journal of Cognitive Neuroscience, 28*(7), 1010–1023. https://doi.org/10.1162/jocn_a_00951
- Dandolo, L. C., & Schwabe, L. (2018). Time-dependent memory transformation along the hippocampal anterior–posterior axis. *Nature Communications, 9*(1), 1205. <https://doi.org/10.1038/s41467-018-03661-7>
- Diedrichsen, J., & Kriegeskorte, N. (2017). Representational models: A common framework for understanding encoding, pattern-component, and representational-similarity analysis. *PLOS Computational Biology, 13*(4), e1005508. <https://doi.org/10.1371/journal.pcbi.1005508>
- Huffman, D. J., & Stark, C. E. L. (2017). The influence of low-level stimulus features on the representation of contexts, items, and their mnemonic associations. *NeuroImage, 155*, 513–529. <https://doi.org/10.1016/j.neuroimage.2017.04.019>
- Klun, L. M., Dandolo, L. C., Jocham, G., & Schwabe, L. (2019). Dorsolateral Prefrontal Cortex Enables Updating of Established Memories. *Cerebral Cortex, 29*(10), 4154–4168. <https://doi.org/10.1093/cercor/bhy298>
- Kriegeskorte, N., Mur, M., & Bandettini, P. (2008). Representational similarity analysis – connecting the branches of systems neuroscience. *Frontiers in Systems Neuroscience, 2*, 4. <https://doi.org/10.3389/neuro.06.004.2008>
- LaRocque, K. F., Smith, M. E., Carr, V. A., Witthoft, N., Grill-Spector, K., & Wagner, A. D. (2013). Global similarity and pattern separation in the human medial temporal lobe predict subsequent memory. *Journal of Neuroscience, 33*(13), 5466–5474. <https://doi.org/10.1523/JNEUROSCI.4293-12.2013>

- Nili, H., Wingfield, C., Walther, A., Su, L., Marslen-Wilson, W., & Kriegeskorte, N. (2014). A toolbox for representational similarity analysis. *PLoS Computational Biology*, *10*(4), e1003553.
<https://doi.org/10.1371/journal.pcbi.1003553>
- Ritchey, M., Wing, E. A., LaBar, K. S., & Cabeza, R. (2013). Neural similarity between encoding and retrieval is related to memory via hippocampal interactions. *Cerebral Cortex*, *23*(12), 2818–2828. <https://doi.org/10.1093/cercor/bhs258>
- Staresina, B. P., Henson, R. N. A., Kriegeskorte, N., & Alink, A. (2012). Episodic reinstatement in the medial temporal lobe. *Journal of Neuroscience*, *32*(50), 18150–18156.
<https://doi.org/10.1523/JNEUROSCI.4156-12.2012>
- Tallman, C. W., Clark, R. E., & Smith, C. N. (2022). Human brain activity and functional connectivity as memories age from one hour to one month. *Cognitive Neuroscience*, *13*(3–4), 1–19.
<https://doi.org/10.1080/17588928.2021.2021164>
- Xiao, X., Dong, Q., Gao, J., Men, W., Poldrack, R. A., & Xue, G. (2017). Transformed neural pattern reinstatement during episodic memory retrieval. *The Journal of Neuroscience*, *37*(11), 2986–2998. <https://doi.org/10.1523/JNEUROSCI.2324-16.2017>
- Xue, G. (2018). The neural representations underlying human episodic memory. *Trends in Cognitive Sciences*, *22*(6), 544–561. <https://doi.org/10.1016/j.tics.2018.03.004>
- Xue, G., Dong, Q., Chen, C., Lu, Z., Mumford, J. A., & Poldrack, R. A. (2010). Greater neural pattern similarity across repetitions is associated with better memory. *Science*, *330*(6000), 97–101.
<https://doi.org/10.1126/science.1193125>

REVIEWERS' COMMENTS

Reviewer #1 (Remarks to the Author):

Although I can understand Reviewer 2's concerns, I think it is difficult to establish the status of a memory based only on being identified as "remembered" in a behavioural manner. However, what I find important is to make clear that all trials were used and why. I believe that the authors properly addressed the points raised and the adjusted changes in the manuscript make clear how analyses were performed and the rationale behind.

Reviewer #2 (Remarks to the Author):

The authors have provided clarification of the analytic approach. Specifically, they clarified that they use all trials, regardless of memory, even for the 28 day brain data. I had assumed they would have only used remembered trials because the focus of the study and the interpretation rests heavily on 'memory representations' and transformations. However, unfortunately, with this new information in hand, there are myriad concerns about the reported results that relate to this. The most important is that it is impossible to know how to interpret the results now. If memories have been forgotten over time, they may show more similarity with other perceptually related items based on so many factors, none of which have anything to do with memory representations. An old 'memory' that is forgotten may look similar to a new perceptually similar memory simply because you are comparing two 'new' memories with similar content. Thus the framing of this paper as if it has anything to do with memory transformations is very misleading. Most (if not all) of the current papers examining memory transformations treat correct trials separately from incorrect trials. Unfortunately, this is a major flaw in the current analyses and approach and renders the result uninterpretable as they are currently presented and written.

Reviewer #3 (Remarks to the Author):

I still think this is an excellent manuscript and agree with the revision of the authors.

Responses to reviewers

Reviewer #1

Although I can understand Reviewer 2's concerns, I think it is difficult to establish the status of a memory based only on being identified as "remembered" in a behavioural manner. However, what I find important is to make clear that all trials were used and why. I believe that the authors properly addressed the points raised and the adjusted changes in the manuscript make clear how analyses were performed and the rationale behind.

RESPONSE: We thank the reviewer for evaluating our manuscript again and are glad that they consider our responses to the concerns raised by reviewer#2 appropriate.

Reviewer #2

The authors have provided clarification of the analytic approach. Specifically, they clarified that they use all trials, regardless of memory, even for the 28 day brain data. I had assumed they would have only used remembered trials because the focus of the study and the interpretation rests heavily on 'memory representations' and transformations. However, unfortunately, with this new information in hand, there are myriad concerns about the reported results that relate to this. The most important is that it is impossible to know how to interpret the results now. If memories have been forgotten over time, they may show more similarity with other perceptually related items based on so many factors, none of which have anything to do with memory representations. An old 'memory' that is forgotten may look similar to a new perceptually similar memory simply because you are comparing two 'new' memories with similar content. Thus the framing of this paper as if it has anything to do with memory transformations is very misleading. Most (if not all) of the current papers examining memory transformations treat correct trials separately from incorrect trials. Unfortunately, this is a major flaw in the current analyses and approach and renders the result uninterpretable as they are currently presented and written.

RESPONSE: We thank the reviewer for evaluating our revised manuscript again. This comment basically repeats the previous comment of the reviewer to which we had already extensively responded. In short, (1) we disagree with the reviewer as a focus exclusively on remembered trials would have led to biased results in our time-dependent analysis (e.g. Tallman et al., 2022), and (2) our approach is well in line with several other publications in the field (Blumenthal et al., 2018; Clarke et al., 2016; Huffman & Stark, 2017; Klüen et al., 2019; Dandolo & Schwabe, 2018). Moreover, it is important to note that our multivariate ERS data remain largely unaffected even when we focus exclusively on remembered trials, as suggested by the reviewer, and that our neural data show overall direct links to the behavioral data. We have addressed all of these aspects in detail in our revised manuscript and made very explicit how we analyzed our data and why we did so. We do not see what else we could do and we were glad to see that reviewers #1 and #3 agreed to our view and approach. For your information, we copy here our previous response to this reviewer comment.

“We thank the reviewer for the opportunity to clarify the rationale for our analytic approach. We agree that when reporting the results of our model-based Representational Similarity Analysis (RSA) we should have emphasized that we compared the activation patterns of all encoded stimuli to avoid any misunderstanding that our manuscript might have been limited to trials that were behaviorally

identified as “remembered”. Therefore, we have now gone through the manuscript and adjusted relevant sections to emphasize that the similarity between all encoded items and lures was taken into account. In particular, we changed “memory representations” to “representations of encoded events” to more precisely reflect the actual analyses. For example, please see page 2, lines 28-30:

“Model-based MRI analyses revealed time-dependent increases in semantically, but not perceptually, transformed representations of encoded events in prefrontal and parietal cortices, while detailed pattern representations in the anterior hippocampus declined over time.”

Page 4, lines 111-112:

“Within the hippocampus, the anterior hippocampus was associated with distinct representations of encoded events that declined with increasing delay after encoding.”

Page 12, lines 314-315:

“Distinct pattern representations of encoded events in the anterior hippocampus decreased over time”

We acknowledge that several studies investigating memory consolidation have limited their neuroimaging analyses to trials with items that were correctly remembered. However, specifically in long-term memory research, this approach is controversial as it disproportionately omits trials from remote time periods (Tallman et al., 2022). Moreover, in contrast to previous studies, our neural analyses were focused on the neural underpinnings of qualitative changes in the course of memory transformation over time rather than neural activity associated with accurate memories. Reducing our neuroimaging analyses to only remembered trials would most likely lead to an exclusion of memories particularly low in specificity, which would make an analysis of the neural underpinnings of time-dependent memory transformation virtually impossible. Moreover, multivariate analysis of fMRI data is expected to be much more sensitive to fine-grained changes in memory representations over time than the dichotomous (“old” vs. “new”) behavioral data. We now clarify this on page 32, lines 858-867:

“We opted for an analysis at the category level instead of relying on participants’ correct or incorrect responses because (i) we were interested in how the encoding-retrieval delay and lure type affected the similarity between representational patterns as an indicator of the specificity of the neural representational patterns rather than the underlying neural patterns of a specific behavioral response; (ii) (multivariate) neural data are much more sensitive to fine-grained changes in memory representations compared to behavioral data that is merely based on dichotomous ‘yes’ vs. ‘no’ (i.e. ‘old’ vs. ‘new’) responses; (iii) reducing analyses on incorrectly endorsed lures (FAs) would have resulted in an insufficient number of trials for the fMRI analyses while (iv) focusing solely on correctly endorsed items (hits) would exclude items that are particularly low in memory specificity, which are of particular interest when investigating the neural underpinnings of memory transformation over time.”

In the present study, we investigated the neural underpinnings of memory transformation by comparing neural activation pattern similarities between originally encoded items and lures to hypothesis-driven conceptual models that encapsulated unique similarity profiles between originally encoded items and different types of lures, i.e. they expected original items to be either represented distinct to all lures (model 1) or similar to either perceptually related (model 2) or semantically related (model 3) lures. This approach, i.e. comparing multivariate representational patterns of all experimental trials (irrespective of the correctness of the response) to conceptual models, allows inferences about the structure of neural representations (Diedrichsen & Kriegeskorte, 2017; Kriegeskorte et al., 2008; Nili et al., 2014) and has been successfully employed in previous studies to characterize memory representations, even at longer delays after encoding (Blumenthal et al., 2018; Clarke et al., 2016; Huffman & Stark, 2017; Kluehn et al., 2019; Dandolo & Schwabe, 2018). Therefore,

the model-based RSA approach we employed here is highly suitable for investigating changes in memory quality over time. We now clarify our rationale in more detail on page 32, lines 870-874:

“This approach, i.e. comparing multivariate representational patterns of all experimental trials (irrespective of the correctness of the response) to conceptual models, allows inferences about the structure of neural representations^{29,30,73} and has been successfully employed in previous studies to characterize memory representations, even at longer delays after encoding^{19,74-77} and is thus highly suitable for investigating changes in memory quality over time.”

While our model-based RSA allowed investigating time-dependent changes in representational similarity between (all) encoded and new item categories at memory test, we further investigated the reactivation of individual items during memory test, i.e. Encoding-Retrieval-Similarity (ERS) as a measure of trial-specific memory reinstatement (Xue, 2018). Specifically, we investigated delay-dependent changes in hippocampal memory reinstatement and correlated item-specific pattern similarity measures with behavior on a trial-by-trial-level, an approach that is, again, in accordance with previous literature (LaRocque et al., 2013; Ritchey et al., 2013; Staresina et al., 2012; Xiao et al., 2017; Xue et al., 2010). Moreover, we showed that results of our ERS-analysis did not change when either all original items or only items that were behaviorally indicated as remembered (hits) were included in this analysis (as mentioned in our manuscript on page 19, lines 458-461). Furthermore, we show that representational changes in hippocampal pattern reinstatement were directly linked to item recognition, and, most importantly, to our behavioral readout of semantic memory transformation, i.e. hits and false alarms to corresponding semantically related items, respectively (see page 19, lines 462-467 and lines 474-485 in our manuscript).

Finally, we would like to emphasize that, although our behavioral data demonstrate the expected memory transformation over time, they also show intact memory at the delayed test (see Figure 1). Even at the 28d delay, the hit rate is above 70 percent while the overall false alarm rate is below 25 percent. Thus, our data do not indicate that memory became absent or fragmented over time but that memory remained largely intact over time.

In sum, our behavioral data demonstrate intact memory even at the delayed test and, using well established analytical approaches, our fMRI data show direct links between neural representations and memory readouts. Thus, any concerns whether we are assessing actual memory representations are in our view not warranted. Our analytic approach is well in line with previous studies on long-term memories and reducing the analysis to only correctly recognized items may have interfered with a sophisticated analysis of qualitative changes of neural memory representations over time. Nevertheless, we have now changed the wording in our manuscript to make our analytical approach more explicit.”

Reviewer #3

I still think this is an excellent manuscript and agree with the revision of the authors.

RESPONSE: We thank the reviewer for evaluating our revised manuscript again and their positive feedback. We are glad that the reviewer agrees with our previous revision.